# DUET: Decentralized Bilevel Optimization without Lower-Level Strong Convexity

**Zhen Qin**[†]**, Zhuqing Liu**[‡]**, Songtao Lu**[◦*]**, Yingbin Liang**[†]**, Jia Liu**[†]

[†]Department of Electrical and Computer Engineering, The Ohio State University
[‡]Department of Computer Science and Engineering, University of North Texas
[◦]Department of Computer Science and Engineering, The Chinese University of Hong Kong

## Abstract

Decentralized bilevel optimization (DBO) provides a powerful framework for multi-agent systems to solve local bilevel tasks in a decentralized fashion without the need for a central server. However, most existing DBO methods rely on lower-level strong convexity (LLSC) to guarantee unique solutions and a well-defined hypergradient for stationarity measure, hindering their applicability in many practical scenarios not satisfying LLSC. To overcome this limitation, we introduce a new single-loop DBO algorithm called diminishing quadratically-regularized bilevel decentralized optimization (DUET), which eliminates the need for LLSC by introducing a diminishing quadratic regularization to the lower-level (LL) objective. We show that DUET achieves an iteration complexity of $O(1/T^{1-5p-\frac{11}{4}\tau})$ for approximate KKT-stationary point convergence under relaxed assumptions, where $p$ and $\tau$ are control parameters for LL learning rate and averaging, respectively. In addition, our DUET algorithm incorporates gradient tracking to address data heterogeneity, a key challenge in DBO settings. To the best of our knowledge, this is the first work to tackle DBO without LLSC under decentralized settings with data heterogeneity. Numerical experiments validate the theoretical findings and demonstrate the practical effectiveness of our proposed algorithms.

## 1 Introduction

In recent years, Decentralized Bilevel Optimization (DBO) over networks has gained significant attention. Consider a DBO problem, where the agents form a peer-to-peer network represented by an undirected connected graph $\mathcal{G} = (\mathcal{N}, \mathcal{L})$. Here $\mathcal{N}$ and $\mathcal{L}$ are the sets of agents (nodes) and edges, respectively, with $|\mathcal{N}| = m$. Each agent $i$ can share information with neighboring agents $\mathcal{N}_i \triangleq \{i' \in \mathcal{N} : (i, i') \in \mathcal{L}\}$ and has access to a local dataset of size $n$. The goal is for all agents to collaboratively solve the following decentralized bilevel optimization problem:

$$\min_{\mathbf{x}_i \in \mathbb{R}^{p_1}, \mathbf{y}_i \in \mathcal{S}(\mathbf{x}_i)} \frac{1}{m} \sum_{i=1}^{m} f_i(\mathbf{x}_i, \mathbf{y}_i) \tag{1}$$

$$\text{s.t. } \mathcal{S}(\mathbf{x}_i) := \arg\min_{\mathbf{y}_i \in \mathbb{R}^{p_2}} g_i(\mathbf{x}_i, \mathbf{y}_i), \forall i; \ \mathbf{x}_i = \mathbf{x}_{i'}, \text{ if } (i, i') \in \mathcal{L}, \tag{2}$$

where $\mathbf{x}_i \in \mathbb{R}^{p_1}$ and $\mathbf{y}_i \in \mathbb{R}^{p_2}$ are parameters to be trained for the UL and LL subproblems at agent $i$, respectively. In this paper, we assume that the UL objective $\frac{1}{m}\sum_{i=1}^{m} f_i(\mathbf{x}_i, \mathbf{y}_i)$ is non-convex and the LL objectives $g_i(\mathbf{x}_i, \mathbf{y}_i), \forall i$, are convex but not strongly convex (i.e., not LLSC), respectively. In the absence of LLSC, the LL solution could be a set-valued map $\mathcal{S}(\mathbf{x}_i)$ (i.e., non-unique optimal solutions to the LL problem). The consensus constraints $\mathbf{x}_i = \mathbf{x}_{i'}$ in (2) ensure that the local copies at neighboring agents $i$ and $i'$ are equal to each other, hence a "consensus" among the agents. The LL variable $\mathbf{y}_i$ is influenced by the UL variable $\mathbf{x}_i$ chosen from the feasible set $\mathbf{X}$ (i.e., $\mathbf{x}_i \in \mathbf{X}$).

DBO provides an effective framework for solving multi-agent, nested optimization problems, where each agent solves a local bilevel task while coordinating with others in a network without relying on a central server. This approach proves particularly beneficial in scenarios such as multi-agent

---

*This work was completed while S. Lu was a senior research scientist at IBM Research in the U.S.

pretraining-finetuning (Rajeswaran et al., 2019; Poon & Peyré, 2021; Liu & Liu, 2021; Hashemi et al., 2024) for Large Language Models (LLMs), which faces significant challenges in private finetuning data environments, thereby making collaboration critical for successful fine-tuning. This framework is also useful in multi-agent meta learning (Rajeswaran et al., 2019; Liu & Liu, 2021), and reinforcement learning (Zhang et al., 2020; Lu et al., 2022), where decentralization reduces communication costs and enhances privacy. DBO problems share the same structure as their centralized counterpart and involve an upper-level (UL) objective function dependent on the optimal parameter values of a lower-level (LL) objective. Even in the centralized case, bilevel optimization is inherently challenging without lower-level strong convexity (LLSC). Several algorithmic approaches have been proposed for centralized bilevel optimization without LLSC. These include using the sequential averaging method (SAM) (Sabach & Shtern, 2017; Liu et al., 2023a; Li et al., 2020), penalty methods (Lu & Mei, 2023), and employing the value function approach (Yao et al., 2024).

Despite the progress in LLSC-less centralized bilevel optimization, designing efficient algorithms for LLSC-less DBO turns out to be *far from* a simple extension of the centralized counterpart. Instead, LLSC-less DBO is a *new area* with a collection of new challenging and important problems, which warrant drastically different algorithmic designs. To date, LLSC-less DBO remains under-explored and this gap in the literature is largely due to the fact that most of the algorithmic ideas for centralized bilevel optimization *cannot* be directly applied to DBO. The first key reason is that, instead of solving a single LL problem, DBO involves *multiple* LL tasks across different agents, making centralized techniques inapplicable. Another major challenge is the data heterogeneity across agents, where each agent works with its own distinct dataset. This further complicates coordination among agents, making it difficult for centralized bilevel optimization approaches to be effective in DBO.

To solve DBO problems without LLSC, a natural starting point is to leverage the decentralized network-consensus approach (Nedic & Ozdaglar, 2009), where agents collaboratively solve a global learning task to reach a consensus. However, two fundamental challenges arise when applying network-consensus methods to DBO: (1) Most existing DBO methods (see, e.g., (Chen et al., 2022; 2023; Lu et al., 2022; Niu et al., 2023; Liu et al., 2022b; Qiu et al., 2023; Liu et al., 2023b),) heavily rely on the assumption of LLSC to guarantee a well-defined Hessian inverse in the upper-level (UL) hypergradient evaluation and the uniqueness of the LL solution, both of which may break down in the absence of LLSC. Exacerbating the situation is the fact that the norm of the UL hypergradient is the most widely used stationarity measure for bilevel optimization. Without a well-defined UL hypergradient in the absence of LLSC, it is not even clear what should be used as a stationarity measure in DBO; (2) Without LLSC, the lack of uniqueness in LL solutions complicates coordination in decentralized network-consensus approaches, where agents must exchange their updates without a central server. Aggregating information from agents becomes more difficult, as the LL solution may shift randomly, resulting in oscillations and poor convergence in DBO.

These challenges motivate us to design new efficient network-consensus-based algorithms for DBO without LLSC. Toward this end, we propose a novel approach called diminishing quadratically-regularized bilevel decentralized optimization (DUET). To our knowledge, none of the existing works has considered solving LLSC-less DBO problems, particularly in decentralized environments with data heterogeneity. Our major contributions are summarized as follows:

- **New Single-Loop Algorithm for LLSC-less DBO:** We propose DUET, a *single-loop* algorithm that integrates gradient tracking and consensus updates to avoid the computational complexity of conventional double-loop structure in bilevel optimization, while ensuring convergence in decentralized settings with data heterogeneity. To our knowledge, this is the first algorithm with provable convergence for DBO without LLSC.

- **New Stationarity Measure for LLSC-less DBO Convergence:** We propose to use the approximate KKT stationarity as the convergence measure of DBO solution quality in our algorithm and provide a detailed convergence rate analysis based on this new measure. We establish *state-of-the-art* finite-time convergence rates of $O(1/T^{1-3p-\frac{11}{4}\tau})$ and $O(1/T^{1-5p-\frac{11}{4}\tau})$ corresponding to the dual variables being bounded and unbounded, respectively. Here, $\tau$ and $p$ control the LL learning rate and averaging, respectively. Moreover, we note that this new approximate KKT-based stationarity measure is general for all DBO problems, which could be of independent theoretical interests. Most notably, the convergence of our DUET algorithms is proved by establishing a *new descent lemma* for the Lyapunov function (cf. Lemma 1), which resolves the difficulty resulting from the inapplicability of using the standard descent lemma in the absence of LLSC.

- **New Augmented LL Objective:** We propose a new augmented LL objective function that allows us to relax several restrictive assumptions made in existing works on LLSC-less bilevel optimization. Notably, our approach does *not* require strong convexity for the UL objective at each agent, Lipschitz continuity of second-order derivatives, or a bounded dataset.
- **Handling Decentralization with Data Heterogeneity:** Without gradient heterogeneity assumptions, our algorithm overcomes the challenges of consensus errors in DBO with data heterogeneity, ensuring agents can synchronize their updates effectively without relying on LLSC.

## 2 RELATED WORK

In this section, we provide an overview of two closely related lines of works: 1) DBO and 2) centralized bilevel optimization without LLSC, thus putting our work into comparative perspectives.

**1) Decentralized Bilevel Optimization (DBO):** Numerous studies have focused on solving decentralized bilevel optimization problems on graphs with LLSC.

One line of work focuses on achieving consensus only for the UL variables, where algorithms are often designed with the hypergradient norm as a stationarity measure. Liu et al. (2022b) introduced a local full-gradient-based algorithm with variance reduction and gradient tracking to achieve $O(n/\epsilon)$ sample complexity and $O(1/\epsilon)$ communication complexity, where $n$ is the size of the dataset at each agent. Furthermore, momentum information is leveraged (Gao et al., 2023; Qiu et al., 2023) to enable single-loop algorithmic architecture by slightly trading off convergence rate performance. To address data heterogeneity, Niu et al. (2023) introduced a single-loop algorithm for nonconvex-strongly-convex bilevel optimization that handles heterogeneity without requiring bounded hypergradients. In collaborative learning, Zhang et al. (2023) proposed COBO, an SGD-based algorithm that scales with clients and outperforms federated learning baselines in heterogeneous settings. Another line of works enforce consensus on the LL variables refer to Appendix B.

Despite these advancements, all aforementioned DBO methods assume LLSC. In contrast, our work departs from this assumption, addressing decentralized bilevel optimization without LLSC under non-i.i.d. data, thus filling a critical gap in the literature.

**2) Centralized Bilevel Optimization without LLSC:** In recent years, centralized bilevel optimization without LLSC has also received increasing attention. For example, Chen et al. (2024a) employed an $\epsilon$-stationary point for the hyper-objective as a convergence metric to quantify algorithmic and proposed a first-order bilevel algorithm with a convergence rate of $O(\ell\kappa^3/\epsilon^2)$, where $\ell$ is a Lipschitz constant and $\kappa$ is the condition number, though it required PL conditions for LL objectives. Liu et al. (2022a) presents a first-order algorithm for non-convex bilevel optimization that avoids Hessian computations, ensures practical efficiency with non-asymptotic convergence guarantees, and introduces a modified KKT condition with a stationarity measure to address bilevel problem challenges. Lu & Mei (2023) reformulated the LL convex bilevel problem as a constrained min-max problem and used the classic penalty method, achieving a convergence rate of $\mathcal{O}(1/\epsilon^4)$ to find $\epsilon$-KKT points. Jiang et al. (2023) tackled a "simple bilevel" problem and proposed a double-loop algorithm utilizing the condition gradient method to approximate nonlinear LL convex functions with linear inequality constraints, achieving a convergence rate of $\mathcal{O}(1/\epsilon^2)$. The stochastic variant in Cao et al. (2023) extends this method to both stochastic and finite-sum settings, with rates matching the standard conditional gradient method. More recently, Yao et al. (2024) introduced a value-function-based proximal Lagrangian approach for constrained LL convex bilevel problems, achieving a convergence rate of $\mathcal{O}(1/T^{(1-2p)/2})$, where $p$ controls the penalty parameter decay. Additional related works are discussed in Appendix B.

The most related work on centralized LLSC-less bilevel optimization is in Liu et al. (2023a), which reformulated the LL convex bilevel problem as a constrained problem using first-order stationarity condition. By employing the KKT condition as the stationarity measure, they proposed a single-loop method that averages the UL and LL objectives, achieving a convergence rate of $O\left(1/T^{1-3p-3\tau}\right)$, where $p$ and $\tau$ control the decreasing LL learning rate and the averaging parameter. While both our work and Liu et al. (2023a) reformulate the bilevel problem as constrained optimization and use aggregation function for solving LL problem, our work differs from Liu et al. (2023a) in the following key aspects: 1) Liu et al. (2023a) required strong convexity of the UL objectives, which limits their approach's applicability in real-world scenarios where UL objectives are often nonconvex. In contrast,

| Algorithm | Setting | Lower Level | Upper Level | Het. Data |
|---|---|---|---|---|
| FOPM (Lu & Mei, 2023) | Centralized | Convex | Nonconvex | NA |
| CG-BiO (Jiang et al., 2023) | Centralized | Convex | Nonconvex | NA |
| F$^2$BA (Chen et al., 2024a) | Centralized | Nonconvex, PL | Nonconvex | NA |
| SBCGF (Cao et al., 2023) | Centralized | Convex | Nonconvex | NA |
| LV-HBA (Yao et al., 2024) | Centralized | Convex | Nonconvex | NA |
| sl-BAMM (Liu et al., 2023a) | Centralized | Convex | Strongly Convex | NA |
| INTERACT (Liu et al., 2022b) | Decentralized | Strongly Convex | Nonconvex | i.i.d |
| DIAMOND (Qiu et al., 2023) | Decentralized | Strongly Convex | Nonconvex | i.i.d |
| Prometheus(Liu et al., 2023b) | Decentralized | Strongly Convex | Nonconvex | i.i.d |
| SLDBO (Dong et al., 2024) | Decentralized | Strongly Convex | Nonconvex | non-i.i.d |
| LoPA (Niu et al., 2023) | Decentralized | Strongly Convex | Nonconvex | non-i.i.d |
| DUET (**Ours**) | Decentralized | Convex | Nonconvex | non-i.i.d |

Table 1: Summary of bilevel optimization algorithms.

our method can be applied to non-convex UL objective by employing a different aggregation function that sequentially averages the LL objective with a diminishing quadratic regularizer. 2) Although both approaches employ the KKT condition as the stationarity measure, our measure applies to the decentralized setting by accounting for consensus errors and aggregating the stationarity measure across subproblems from agents with non-i.i.d. data distributions. In contrast, the KKT-based stationarity measure in Liu et al. (2023a) cannot handle data hetergeneity.

In summary, while the aforementioned existing works addressed centralized bilevel optimization without LLSC, they cannot be generalized to address the decentralized setting with heterogeneous data challenges in a straightforward fashion. In contrast, our work tackles LLSC-less DBO and relaxes several assumptions typically made in the bilevel optimization literature. For easy reference, we summarize the most relevant bilevel optimization algorithms in Table 1.

## 3 THE DIMINISHING QUADRATICALLY-REGULARIZED BILEVEL OPTIMIZATION ALGORITHM (DUET)

**1) Problem Reformulation for a New Stationarity Convergence Metric:** In the LLSC DBO literature, the uniqueness of the LL solution $\mathbf{y}_i^*(\mathbf{x}_i)$ ensures that the hypergradient norm of each agent's UL objective $\Phi^i(\mathbf{x}_i) = f_i(\mathbf{x}_i, \mathbf{y}_i^*(\mathbf{x}_i))$ is well-defined. Consequently, the hypergradient norm of the overall UL objective $\Phi(\mathbf{x}_i) = \frac{1}{m} \sum_{i=1}^{m} \Phi^i(\mathbf{x}_i)$ is also well-defined. This norm has been widely used as a measure for stationarity in previous works (e.g., Ghadimi & Wang (2018); Liu et al. (2022b; 2023b); Dong et al. (2024); Lu & Mei (2023)). However, in the absence of LLSC, the Hessian matrix of the LL problem is *not* full-rank and thus not invertible. In decentralized settings, this problem is further exacerbated by the *inconsistent updates* across agents, which leads to the conventional hypergradient-norm-based stationarity measure being *ill-defined*. This motivates us to develop a *new* stationarity measure that handles both the absence of LLSC and the consensus errors among agents at the same time.

Toward this end, inspired by Wolfe-duality, we first reformulate the LLSC-less DBO problem into an equivalent constrained optimization problem. Instead of directly solving for $\mathbf{y}_i^*(\mathbf{x}_i)$, we replace the LL problem by introducing the LL-stationary condition (i.e., $\nabla_{\mathbf{y}} g(\mathbf{x}, \mathbf{y}) = 0$) as constraints:

$$\min_{\mathbf{x}_i \in \mathbb{R}^{p_1}, \mathbf{y}_i \in \mathbb{R}^{p_2}} \frac{1}{m} \sum_{i=1}^{m} f_i(\mathbf{x}_i, \mathbf{y}_i) \quad \text{s.t. } \nabla_{\mathbf{y}} g_i(\mathbf{x}_i, \mathbf{y}_i) = 0, \forall i; \quad \mathbf{x}_i = \mathbf{x}_{i'}, \text{ if } (i, i') \in \mathcal{L}. \quad (3)$$

The reformulation in Problem (3) is equivalent to the original Problem (1) because the LL-stationarity is both necessary and sufficient for the LL-optimality when the LL problem $g_i(\mathbf{x}_i, \mathbf{y}_i)$ is convex in $\mathbf{y}_i$ for any fixed $\mathbf{x}_i$, which is satisfied in our problem setting. Our key rationale behind converting the original bilevel optimization problem in (1) into an equivalent conventional constrained optimization problem in (3) is to facilitate the use of the KKT conditions, for which the KKT stationary condition can naturally serve as a new stationarity measure, hence resolving the conundrum of lacking a well-defined hypergradient norm as the stationarity measure in the absence of LLSC.

We now state the KKT conditions for Problem (3), for which the Lagrangian function can be written as $\mathcal{L}(\mathbf{x}, \mathbf{y}, \mathbf{v}) := \frac{1}{m} \sum_{i=1}^{m} f_i(\mathbf{x}_i, \mathbf{y}_i) - \sum_{i=1}^{m} \mathbf{v}_i^\top \nabla_{\mathbf{y}} g_i(\mathbf{x}_i, \mathbf{y}_i)$, where $\mathbf{v}_i, \forall i$, are dual variables

associated with the constraints. Then, a KKT solution $(\mathbf{x}_i^*, \mathbf{y}_i^*, \mathbf{v}_i^*)$, if exists, satisfies the following:

$$
\begin{cases}
\frac{1}{m}\nabla_{\mathbf{x}}f_i\left(\mathbf{x}_i^*, \mathbf{y}_i^*\right) - \nabla_{\mathbf{xy}}^2 g_i\left(\mathbf{x}_i^*, \mathbf{y}_i^*\right)\mathbf{v}_i^* = 0, \forall i, & \text{(Stationarity of Problem (3))} \\
\frac{1}{m}\nabla_{\mathbf{y}}f_i\left(\mathbf{x}_i^*, \mathbf{y}_i^*\right) - \nabla_{\mathbf{yy}}^2 g_i\left(\mathbf{x}_i^*, \mathbf{y}_i^*\right)\mathbf{v}_i^* = 0, \forall i, & \text{(Stationarity of Problem (3))} \\
-\nabla_{\mathbf{y}}g_i\left(\mathbf{x}_i^*, \mathbf{y}_i^*\right) = 0, \forall i, & \text{(Primal Feasibility of Problem (3))} \\
\mathbf{x}_i^* - \mathbf{x}_{i'}^* = 0, \forall(i, i') \in \mathcal{L}. & \text{(Primal Feasibility of Problem (3))}
\end{cases}
$$

Note that the dual feasibility and complementary slackness conditions in this KKT system are implied by the primal feasibility condition and hence can be omitted. For convenience, we define the KKT stationarity residual for a primal-dual pair $(\mathbf{x}_i, \mathbf{y}_i, \mathbf{v}_i)$ as $\mathrm{KKT}(\mathbf{x}_i, \mathbf{y}_i, \mathbf{v}_i) := \|\nabla\mathcal{L}(\mathbf{x}_i, \mathbf{y}_i, \mathbf{v}_i)\|^2$, which will be used as a part of our stationarity convergence metric defined later. Note that when LLSC holds, it is not difficult to show that the $\nabla\Phi(\bar{\mathbf{x}}) = 0$ if and only if $\mathrm{KKT}(\mathbf{x}_i, \mathbf{y}_i, \mathbf{v}_i) = 0$ for some $\mathbf{y}, \mathbf{v} \in \mathbb{R}^{p_2}$. This fact will serve as a "bridge" to connect the above KKT staionarity and the hypgradients induced by the diminishing $\mu_t$-quadratic regularization described next.

**2) The Diminishing $\mu_t$-Quadratic Regularization:** To address the challenge of lacking LLSC in our algorithm design, our basic idea is to augment the LL objective function by introducing a quadratic regularizer that is controlled by a sequence of diminishing regularization parameters, thereby reviving the LLSC in each iteration. These regularization parameters are carefully selected to ensure that the augmented problems converge to the original problem, thereby leading to a solution to Problem (3). Specifically, at iteration $t$ we define the augmented LL objective function as follows:

$$
\psi_{\mu_t}^i(\mathbf{x}_{i,t}, \mathbf{y}_{i,t}) := \mu_t h_i(\mathbf{x}_{i,t}, \mathbf{y}_{i,t}) + (1 - \mu_t)g_i(\mathbf{x}_{i,t}, \mathbf{y}_{i,t}), \tag{4}
$$

where $h_i(\mathbf{x}_{i,t}, \mathbf{y}_{i,t}) = \frac{1}{2}\|\mathbf{x}_{i,t}\|^2 + \frac{1}{2}\|\mathbf{y}_{i,t}\|^2$. Here, the norm $\|\cdot\|$ represents the $\ell_2$ norm. $\mathbf{x}_{i,t}$ and $\mathbf{y}_{i,t}$ are the variables corresponding to agent $i$ at iteration $t$. Here $\{\mu_t\}_{t=0}^{\infty}$, where $\mu_t > 0$, $\forall t$, is the diminishing sequence of regularization parameters, which ensures that $\psi_{\mu_t}^i(\mathbf{x}, \cdot)$ is strongly convex for any $\mathbf{x}$. This augmentation also leverages the connection between the KKT condition and the norm of the $\mu_t$-induced hypergradient, allowing us to replace the LL objective $g_i(\mathbf{x}_{i,t}, \mathbf{y}_{i,t})$ by $\psi_{\mu_t}^i(\mathbf{x}_{i,t}, \mathbf{y}_{i,t})$, facilitating the solution to Problem (3). Thanks to the strong convexity of the quadratic regularizor, $\psi_{\mu_t}^i(\mathbf{x}_{i,t}, \cdot)$ has a unique minimizer for any given $\mathbf{x}$-variable, which is denoted as $\mathbf{y}_{i,\mu_t}^*(\mathbf{x}_i)$.

Next, we define the approximate UL objective as $\Phi_{\mu_t}(\mathbf{x}_t) = \frac{1}{m}\sum_{i=1}^m \Phi_{\mu_t}^i(\mathbf{x}_{i,t})$, where $\Phi_{\mu_t}^i(\mathbf{x}_{i,t}) \triangleq f_i(\mathbf{x}_{i,t}, \mathbf{y}_{i,\mu_t}^*(\mathbf{x}_{i,t}))$. Similar to conventional bilevel optimizaiton with LLSC, for differentiable $\Phi_{\mu_t}^i(\mathbf{x}_{i,t})$, the hypergradient $\nabla\Phi_{\mu_t}^i(\mathbf{x}_{i,t})$ can be derived by the chain rule, the implicit function theorem, and the augmented LL function as: $\nabla\Phi_{\mu_t}^i(\mathbf{x}_{i,t}) = \nabla_{\mathbf{x}}f_i\left(\mathbf{x}_{i,t}, \mathbf{y}_{i,\mu_t}^*(\mathbf{x}_{i,t})\right) - \nabla_{\mathbf{xy}}^2\psi_{\mu_t}^i\left(\mathbf{x}_{i,t}, \mathbf{y}_{i,\mu_t}^*(\mathbf{x}_{i,t})\right)\mathbf{v}_{i,\mu_t}^*(\mathbf{x}_{i,t})$, where $\mathbf{v}_{i,\mu_t}^*(\mathbf{x}_{i,t}) \in \mathbb{R}^{p_2}$ is the solution of the linear system: $\mathbf{v}_{i,\mu_t}^*(\mathbf{x}_{i,t}) := [\nabla_{\mathbf{yy}}^2\psi_{\mu_t}^i(\mathbf{x}_{i,t}, \mathbf{y}_{i,\mu_t}^*(\mathbf{x}_{i,t}))]^{-1}\nabla_{\mathbf{y}}f_i(\mathbf{x}_{i,t}, \mathbf{y}_{i,\mu_t}^*(\mathbf{x}_{i,t}))$.

After introducing the regularization in (4), we can now adopt the approximate KKT condition by replacing the LL objective with $\psi_{\mu_t}^i(\mathbf{x}_{i,t}, \mathbf{y}_{i,t})$, and thus consider $\mathrm{KKT}(\mathbf{x}_t, \mathbf{y}_t, \mathbf{v}_t) := \|\nabla\mathcal{L}(\mathbf{x}_t, \mathbf{y}_t, \mathbf{v}_t)\|^2 \leq \epsilon$ at time $t$. Here $\mathbf{x}_t := [\mathbf{x}_{1,t}^\top, \ldots, \mathbf{x}_{m,t}^\top]^\top$, and similarly for $\mathbf{y}_t$ and $\mathbf{v}_t$.

**3) Consensus Mechanism:** To address the consensus constraint $\mathbf{x}_i = \mathbf{x}_{i'}$, $(i, i') \in \mathcal{L}$ in Problem (3), we adopt the network consensus approach (Nedic & Ozdaglar, 2009), where a consensus weight matrix $\mathbf{M} \in \mathbb{R}^{m \times m}$ is used to mix and aggregate information at each iteration. The element $[\mathbf{M}]_{ij}$ represents the weight assigned for the information from the $j$-th agent at the $i$-th agent. Each agent uses the weights in its corresponding row in the $\mathbf{M} \in \mathbb{R}^{m \times m}$ to aggregate the information from its neighbors. For consensus to be reached asymptotically, the matrix $\mathbf{M}$ should satisfy certain properties: (1) *Doubly Stochastic:* $\sum_{i=1}^m[\mathbf{M}]_{ij} = \sum_{j=1}^m[\mathbf{M}]_{ij} = 1$; (2) *Symmetric:* $[\mathbf{M}]_{ij} = [\mathbf{M}]_{ji}$ for all $i, j \in \mathcal{N}$; and (3) *Sparsity Pattern Adhering to the Network Topology:* $[\mathbf{M}]_{ij} > 0$ if $(i, j) \in \mathcal{N}$ and $[\mathbf{M}]_{ij} = 0$ otherwise for all $i, j \in \mathcal{L}$. These properties ensure that the eigenvalues of $\mathbf{M}$ are real and fall within the interval $(-1, 1]$, thus being sortable. Then, we order the eigenvalues of $\mathbf{M}$ as: $-1 < \lambda_m(\mathbf{M}) \leq \cdots \leq \lambda_2(\mathbf{M}) < \lambda_1(\mathbf{M}) = 1$. The second-largest eigenvalue in magnitude of $\mathbf{M}$, denoted as $\lambda \triangleq \max\{|\lambda_2(\mathbf{M})|, |\lambda_m(\mathbf{M})|\}$, will play an important role in our step size selection and thus convergence rate in our proposed DUET algorithm.

**4) The Proposed Algorithm:** With the preliminaries in 1)–3), we are now ready to present our diminishing quadratically-regularized bilevel decentralized optimization (DUET) method. This method is specifically designed to address the challenges of bilevel optimization without

---

**Algorithm 1** The DUET Algorithm at Each Agent $i$.

Set parameter pair $(\mathbf{x}_{i,0}, \mathbf{y}_{i,0}, \mathbf{v}_{i,0}) = (\mathbf{x}_0, \mathbf{y}_0, \mathbf{v}_0)$.
**for** $t = 1, \cdots, T$ **do**
    Update local models $(\mathbf{x}_{i,t+1}, \mathbf{y}_{i,t+1}, \mathbf{v}_{i,t+1})$ as in Eqs. (5);
    Compute the $(\mathbf{d}_{\mathbf{x}}^{i,t}, \mathbf{d}_{\mathbf{y}}^{i,t}, \mathbf{d}_{\mathbf{v}}^{i,t})$ local estimator as in Eq. (6);
    Track global gradients $(\mathbf{h}_{\mathbf{x}}^{i,t}, \mathbf{h}_{\mathbf{y}}^{i,t}, \mathbf{h}_{\mathbf{v}}^{i,t})$ as in Eq. (7);
**end for**

---

LLSC in decentralized environments with data heterogeneity. Our DUET method draws inspiration from the centralized SOBA approach (Dagréou et al., 2022), which features a single-loop structure that is easier to implement and reduces the computational complexity compared to traditional double-loop methods. However, fundamentally different from SOBA, DUET builds on the augmented LL objective function in (4), enabling us to address DBO problems without LLSC. The procedure of our algorithm DUET can be organized into three key steps:

- *Step 1 (Update Local Models):* In each iteration $t$, each agent $i$ updates its local variables as:

$$\mathbf{x}_{i,t+1} = \sum_{i' \in \mathcal{N}_i} [\mathbf{M}]_{ii'} \mathbf{x}_{i',t} - \alpha_t \mathbf{h}_{\mathbf{x}}^{i,t}; \;\; \mathbf{y}_{i,t+1} = P_{r_y^t}[\mathbf{y}_{i,t} - \beta_t \mathbf{h}_{\mathbf{y}}^{i,t}]; \;\; \mathbf{v}_{i,t+1} = P_{r_v^t}[\mathbf{v}_{i,t} + \eta_t \mathbf{h}_{\mathbf{v}}^{i,t}], \quad (5)$$

where $\alpha_t$, $\beta_t$ and $\eta_t$ are step-sizes for updating $\mathbf{x}_i$, $\mathbf{y}_i$ and $\mathbf{v}_i$ variables, respectively, and $P_r[\cdot]$ denotes a projection operator defined as $P_r[q] := \arg\min_{\|q'\| \leq r} \|q' - q\| = \min\{q, r\frac{q}{\|q\|}\}$, where $r > 0$ is the radius. $r_v^t$ and $r_y^t$ are projection parameters of $\mathbf{y}_i$ and $\mathbf{v}_i$ variables, respectively (to be defined in the next subsection). First, the UL variable $\mathbf{x}_{i,t+1}$ is updated by aggregating the UL information from its neighbors and adjusting based on the local gradient $\mathbf{h}_{\mathbf{x}}^{i,t}$, which induces consensus among the agents in the network. The LL variable $\mathbf{y}_{i,t+1}$ is updated through a projected local gradient descent step, reflecting the agent's progress in solving its local optimization problem. Finally, the dual variable $\mathbf{v}_{i,t+1}$ is updated using a projected gradient ascent step to ensure that the necessary optimality conditions of the LL problem are maintained. We also employ the projection steps of $\mathbf{y}_{i,t}$ and $\mathbf{v}_{i,t}$ is to ensure that the sequences $\{\mathbf{y}_{i,t}\}$ and $\{\mathbf{v}_{i,t}\}$ are bounded with radii $r_v^t$ and $r_y^t$ respectively. Later we will show that, based on increasing $r_v^t$ and $r_y^t$ with respect to $t$, the boundedness of $\mathbf{y}_{i,t}$- and $\mathbf{v}_{i,t}$-variables results in the boundedness of $\mathbf{x}_{i,t}$-variables, hence ensuring convergence.

- *Step 2 (Local Gradient Estimate):* In the local gradient estimator step, each agent $i$ computes its local gradients to update its variables:

$$\begin{cases} \mathbf{d}_{\mathbf{y}}^{i,t} = \nabla_{\mathbf{y}} \psi_{\mu_t}^i (\mathbf{x}_{i,t}, \mathbf{y}_{i,t}), \\ \mathbf{d}_{\mathbf{v}}^{i,t} = \nabla_{\mathbf{y}} f_i (\mathbf{x}_{i,t}, \mathbf{y}_{i,t}) - \nabla_{\mathbf{yy}}^2 \psi_{\mu_t}^i (\mathbf{x}_{i,t}, \mathbf{y}_{i,t}) \mathbf{v}_{i,t}, \\ \mathbf{d}_{\mathbf{x}}^{i,t} = \nabla_{\mathbf{x}} f_i (\mathbf{x}_{i,t}, \mathbf{y}_{i,t}) - \nabla_{\mathbf{xy}}^2 \psi_{\mu_t}^i (\mathbf{x}_{i,t}, \mathbf{y}_{i,t}) \mathbf{v}_{i,t}. \end{cases} \quad (6)$$

We update $\mathbf{y}_{i,t}$ using the gradient of the augmented LL objective $\psi_{\mu_t}^i (\mathbf{x}_{i,t}, \mathbf{y}_{i,t})$, while the gradients for $\mathbf{v}_{i,t}$ and $\mathbf{x}_{i,t}$ are derived using the KKT conditions. This ensures that the LL solution meets optimality constraints and that the UL problem is solved efficiently.

- *Step 3 (Gradient Tracking in Upper-Level Parameters):* In this step, each agent $i$ updates its tracked gradient $\mathbf{h}_{\mathbf{x}}^{i,t}$ by averaging the gradients from neighboring agents and correcting the local estimates:

$$\mathbf{h}_{\mathbf{x}}^{i,t} = \sum_{i' \in \mathcal{N}_i} [\mathbf{M}]_{ii'} \mathbf{h}_{\mathbf{x}}^{i',t-1} + \mathbf{d}_{\mathbf{x}}^{i,t} - \mathbf{d}_{\mathbf{x}}^{i,t-1}; \quad \mathbf{h}_{\mathbf{y}}^{i,t} = \mathbf{d}_{\mathbf{y}}^{i,t}; \quad \mathbf{h}_{\mathbf{v}}^{i,t} = \mathbf{d}_{\mathbf{v}}^{i,t}. \quad (7)$$

The purpose of gradient tracking for the UL variables is to further reduce consensus error and accelerate convergence even under non-i.i.d data distributions. On the other hand, since the LL variables $\mathbf{y}_{i,t}$ and $\mathbf{v}_{i,t}$ are updated locally without consensus requirements, gradient tracking is not needed for the LL variables.

To conclude the discussion of the DUET's algorithmic design, we summarize the per-agent algorithm of DUET in Algorithm 1.

## 4 THEORETICAL CONVERGENCE RATE ANALYSIS

In this section, we will establish the theoretical convergence rate for the proposed DUET algorithm. Before we state our main convergence result, we first present several needed assumptions as follows.

**Assumption 1** (Boundedness and Smoothness of the UL Objectives). *The UL objectives $f_i$ satisfies: (a) For any $i \in [m]$, the UL objective $f_i(\mathbf{x}_i, \cdot)$ has a uniform lower bound denoted by $f_{i_0}$; and (b) For any $i \in [m]$, the UL objective $f_i$ is twice differentiable and Lipschitz continuous with a Lipschitz constant of $L_{f_{i_0}}$. The first-order derivatives $\nabla_{\mathbf{x}} f_i(\cdot, \mathbf{y}_i)$, $\nabla_{\mathbf{x}} f_i(\mathbf{x}_i, \cdot)$, $\nabla_{\mathbf{y}} f_i(\cdot, \mathbf{y}_i)$, $\nabla_{\mathbf{y}} f_i(\mathbf{x}_i, \cdot)$ are Lipschitz continuous with respective Lipschitz constants $L_{f_{i\mathbf{x}1}}, L_{f_{i\mathbf{x}2}}, L_{f_{i\mathbf{y}1}}, L_{f_{i\mathbf{y}2}}$.*

**Assumption 2** (Convexity and Smoothness of the LL Objectives). *The LL objective $g_i$ satisfies: (a) for any $i \in [m]$ and any $\mathbf{x}_i$, the LL objective $g_i(\mathbf{x}, \cdot)$ is convex; and (b) for any $i \in [m]$, the LL objective $g_i$ is twice differentiable and the derivatives $\nabla_{\mathbf{y}} g_i$ and $\nabla^2_{\mathbf{xy}} g_i, \nabla^2_{\mathbf{yy}} g_i$ are Lipschitz continuous in $(\mathbf{x}_i, \mathbf{y}_i)$ with respective Lipschitz constants $L_{g_{i\mathbf{y}1}}, L_{g_{i\mathbf{y}2}}$ and $L_{g_{i\mathbf{xy}1}}, L_{g_{i\mathbf{xy}2}}, L_{g_{i\mathbf{yy}1}}, L_{g_{i\mathbf{yy}2}}$.*

The smoothness and boundedness assumptions in Assumptions 1 and 2 are standard in the literature of bilevel optimization (Ghadimi & Wang, 2018; Ji et al., 2021; ji & Liang, 2023; Dagréou et al., 2022; Ji et al., 2024; Kong et al., 2024; He et al., 2024). Unlike many works, however, we do not assume LLSC, which significantly complicates the theoretical analysis. Under the above assumptions, the augmented LL objective $\psi^i_{\mu_t}(\mathbf{x}_i, \cdot)$ is $\sigma_{\psi_{\mu_t}}$- strongly convex with $\sigma_{\psi_{\mu_t}} = \mu_t \sigma_{h_i} = \mu_t$, where $\sigma_{h_i} = 1$ by the definition of function $h_i$. Hence, $\psi^i_{\mu_t}(\mathbf{x}_i, \cdot)$ has a unique minimizer, denoted by $\mathbf{y}^*_{i,\mu_t}(\mathbf{x}_i)$. To this end, we introduce the following convergence metric to help us approach the KKT condition as the $\mu_t$-regularized problem converges to the original problem as $\mu_t$ shrinks to zero. Specifically, for each $\{\mathbf{x}_t, \mathbf{y}_t, \mathbf{v}_t\}$ at time $t$, we define

$$\Pi(\mathbf{x}_t, \mathbf{y}_t, \mathbf{v}_t) := \underbrace{\mathbb{E}\|\nabla\Phi_{\mu_t}(\bar{\mathbf{x}}_t)\|^2}_{\text{Stationarity Error}} + \underbrace{\mathbb{E}\|\mathbf{x}_t - \mathbf{1} \otimes \bar{\mathbf{x}}_t\|^2}_{\text{Consensus Error}} + \underbrace{\mathbb{E}\|\mathbf{y}^*_t - \mathbf{y}_t\|^2}_{\text{Lower-Level Error}} + \underbrace{\mathbb{E}\|\mathbf{v}^*_t - \mathbf{v}_t\|^2}_{\text{Dual Multiplier Error}}, \quad (8)$$

where $\bar{\mathbf{x}}_t \triangleq \frac{1}{m}\sum_{i=1}^m \mathbf{x}_{i,t}$, $\mathbf{y}_t \triangleq [\mathbf{y}_{1,t}^\top, \ldots, \mathbf{y}_{m,t}^\top]^\top$, $\mathbf{y}^*_{\mu_t} \triangleq [\mathbf{y}_{1,\mu_t}^{*\top}, \ldots, \mathbf{y}_{m,\mu_t}^{*\top}]^\top$, $\mathbf{v}_t \triangleq [\mathbf{v}_{1,t}^\top, \ldots, \mathbf{v}_{m,t}^\top]^\top$, and $\mathbf{v}^*_{\mu_t} \triangleq [\mathbf{v}_{1,\mu_t}^{*\top}, \ldots, \mathbf{v}_{m,\mu_t}^{*\top}]^\top$. $\otimes$ is the Kronecker product. Note that the first term in (8) quantifies the convergence of $\bar{\mathbf{x}}_t$ to a stationary point of the global objective. The second term measures the consensus error among local copies of the UL variables. The third and fourth terms quantify the optimality gap in the LL problem's primal variable $\mathbf{y}_t$ and dual variable $\mathbf{v}_t$, respectively, across all agents. Thus, $\Pi(\mathbf{x}_t, \mathbf{y}_t, \mathbf{v}_t) \leq \epsilon$ for a small $\epsilon$-value implies that the algorithm achieves three goals simultaneously: i) approximate KKT stationarity convergence of Problem (3), ii) consensus of UL $\mathbf{x}_t$-variables, and iii) optimal solutions to the LL $\mathbf{y}_t$-variables and dual $\mathbf{v}_t$-variables.

With Assumptions 1 and 2, we also define the following parameters that will be used in our algorithm:

$$r_v^t := \frac{L_{f_{i_0}}}{\sigma_{\psi_{\mu_t}}}, \quad B_1 := L_{f_{i_0}} + (L_{\psi_{i\mathbf{y}1}} r_v^t), \quad \lambda_1 := \frac{1-\lambda^2}{\lambda} + \frac{4}{1-\lambda} + \frac{16}{(1-\lambda)^3},$$

$$r_x^t := \sqrt{2\lambda_1 B_1^2 m \bar{\beta}\bar{\mu} + 2\sum_{i=1}^m \|\mathbf{x}_{i,0}\|^2}, \quad r_y^t := \frac{L_{\psi_{\mathbf{y}1}}}{\sigma_{\psi_{\mu_t}}} r_x^t + \frac{1}{\sigma_{\psi_{\mu_t}}}\|\nabla_{\mathbf{y}}\psi_{\mu_t}(0,0)\|, \quad (9)$$

where $\mathbf{x}_{i,0}$, $i \in [m]$, are the initial points, constants $L_{f_{i_0}}, L_{\psi_{i\mathbf{y}1}}$, and parameter $\sigma_{\psi_{\mu_t}}$ are as defined in Assumptions 1 and 2, and $\lambda$ is the second largest eigenvalue in magnitude of the network graph. $\bar{\beta}$ and $\bar{\mu}$ are the initial LL learning rate and averaging control parameter, respectively. Both $\bar{\beta}$ and $\bar{\mu}$ are constants. With the above notations, we are now ready to state the main convergence rate result of DUET as follows:

**Theorem 1** (Convergence Analysis for DUET). *Under Assumptions 1 and 2, choose $\mu_t = \bar{\mu}(t+1)^{-p}$. Choose the step-sizes as $\beta_t \in [\bar{\beta}(t+1)^{-\tau/4}, 1/L_{f_{i_y2}} + L_{g_{y2}}]$ with $0 < p < 1/6, 0 < \tau < 2/33$ and $\eta_t = (t+1)^{-\tau/2}\beta_t\mu_t$, and $\alpha_t = (t+1)^{-3\tau/2}\beta_t\mu_t^5$. It then holds that*

$$\min_{0 \leq t \leq T} \Pi(\mathbf{x}_t, \mathbf{y}_t, \mathbf{v}_t) = O\left(1/T^{1-5p-\frac{11}{4}\tau}\right).$$

*Further, if $\mathbf{v}_{i,t}$ is bounded, it holds that $\min_{0 \leq t \leq T} \text{KKT}(\mathbf{x}_t, \mathbf{y}_t, \mathbf{v}_t) = O(1/T^{1-5p-\frac{11}{4}\tau} + 1/T^{2p})$.*

The following result immediately follows from Theorem 1:

**Corollary 2.** *Let $p = \frac{1}{7}$ and $\tau \to 0$. Then,* DUET *converges to a KKT point of Problem (3) at rate of $O(1/T^{\frac{2}{7}})$, which implies that the number of communication rounds required to reach $\epsilon$-accuracy for our* DUET *method is $O(1/\epsilon^{\frac{7}{2}})$.*

Moreover, if the problem instance satisfies the stronger condition that $\|\mathbf{v}^*_{\mu_t}(\mathbf{x}_{i,t})\|$ is bounded, we can further improve the convergence result of DUET in Theorem 1 as follows:

**Theorem 3** (Convergence Analysis for DUET with Boundedness Assumption). *Under Assumptions 1 and 2, choose $\mu_t = \bar{\mu}(t+1)^{-p}$. Choose step-sizes as $\beta_t \in [\bar{\beta}(t+1)^{-\tau/4}, 1/L_{f_{iy2}} + L_{g_{y2}}]$ with $0 < p < 1/4, 0 < \tau < 1/11$ and $\eta_t = (t+1)^{-\tau/2}\beta_t$, and $\alpha_t = (t+1)^{-3\tau/2}\beta_t\mu_t^3$. If $\left\|\mathbf{v}_{\mu_t}^*(\mathbf{x}_{i,t})\right\|$ is bounded, it then holds that:*

$$\min_{0 \leq t \leq T} \Pi(\mathbf{x}_t, \mathbf{y}_t, \mathbf{v}_t) = O\left(1/T^{1-3p-\frac{11}{4}\tau}\right).$$

*Further, if $\mathbf{v}_{i,t}$ is bounded, it holds that $\min_{0 \leq t \leq T} \text{KKT}(\mathbf{x}_t, \mathbf{y}_t, \mathbf{v}_t) = O(1/T^{1-3p-\frac{11}{4}\tau} + 1/T^{2p})$.*

Similar to Theorem 1, the following result immediately follows from Theorem 3:

**Corollary 4.** *Let $p = \frac{1}{5}$ and $\tau \to 0$. Then, DUET converges to a KKT point at a rate of $O(1/T^{\frac{2}{5}})$, which implies that the number of communication rounds required to reach $\epsilon$-accuracy for our DUET method is $O(1/\epsilon^{\frac{5}{2}})$.*

Due to space limitation, we relegate the proofs of Theorems 1 and 3 to the Appendix. In here, several important remarks for the proofs of Theorems 1 and 3 are in order:

- It is worth noting that, compared to existing works on decentralized bilevel optimization, the major challenge in proving the convergence results in Theorems 1 and 3 stems from the absence of LLSC, which breaks the standard descent lemma for the LL variable $\mathbf{y}_i$ in convergence analysis. To address this challenge, leveraging our augmented LL objective, we establish a new descent lemma for the implied UL objective function at time $t$, expressed in terms of $\bar{\mathbf{x}}_t$, as follows:

**Lemma 1** (A New Descent Lemma of the Implied UL Objective). *Under Assumptions 1 and 2 and letting $\mu_{t+1} \leq \mu_t \leq \frac{1}{2}$ for all $t$, the sequence $\{\mathbf{x}_{i,t}, \mathbf{y}_{i,t}, \mathbf{v}_{i,t}\}$ generated by DUET satisfy:*

$$
\Phi_{\mu_{t+1}}\left(\bar{\mathbf{x}}_{t+1}\right) - \Phi_{\mu_t}\left(\bar{\mathbf{x}}_t\right)
$$

$$
\leq -\frac{\alpha_t}{2}\left\|\nabla\Phi_{\mu_t}\left(\bar{\mathbf{x}}_t\right)\right\|^2 - \left(\frac{\alpha_t}{2} - \frac{\alpha_t^2 L_{\Phi\mu_t}}{2\sigma_{\psi_{\mu_t}}^2}\right)\|\bar{\mathbf{h}}_{\mathbf{x}}^t\|^2 + \frac{\alpha_t L_{\Phi\mu_t}}{r\sigma_{\psi_{\mu_t}}^2}\frac{1}{m}\sum_{i=1}^m\|\bar{\mathbf{x}}_t - \mathbf{x}_{i,t}\|^2
$$

$$
+ \frac{2\alpha_t r}{m}\sum_{i=1}^m\left(L_{\psi_{\mathbf{xy2}}}\left\|\mathbf{v}_{\mu_t}^*\left(\mathbf{x}_{i,t}\right)\right\| + L_{f_{i\mathbf{x2}}}\right)^2\left\|\mathbf{y}_{i,t} - \mathbf{y}_{\mu_t}^*\left(\mathbf{x}_{i,t}\right)\right\|^2
$$

$$
+ \frac{\alpha_t r L_{\psi_{\mathbf{y1}}}^2}{m}\sum_{i=1}^m\left\|\mathbf{v}_{i,t} - \mathbf{v}_{\mu_t}^*\left(\mathbf{x}_{i,t}\right)\right\|^2 + 2\alpha_t r\left\|\frac{1}{m}\sum_{i=1}^m\mathbf{d}_{\mathbf{x}}^{i,t} - \bar{\mathbf{h}}_{\mathbf{x}}^t\right\|^2
$$

$$
+ \frac{1}{m}\sum_{i=1}^m\left[\frac{2\left\|\nabla_{\mathbf{y}}f_i\left(\bar{\mathbf{x}}_{t+1}, \mathbf{y}_{\mu_{t+1}}^*\left(\bar{\mathbf{x}}_{t+1}\right)\right)\right\|\left\|\mathbf{y}_{\mu_{t+1}}^*\left(\bar{\mathbf{x}}_{t+1}\right)\right\|}{\sigma_{h_i}}\left(\frac{\mu_t - \mu_{t+1}}{\mu_t}\right)\right]
$$

$$
+ \frac{1}{m}\sum_{i=1}^m\frac{2L_{f_{i\mathbf{y2}}}\|\mathbf{y}_{\mu_{t+1}}^*(\bar{\mathbf{x}}_{t+1})\|^2}{\sigma_{h_i}^2}\left(\frac{\mu_t - \mu_{t+1}}{\mu_t}\right)^2,
$$

*where $C_{\Phi 1}$ and $C_{\Phi 2}$ are problem-dependent constants provided in Lemma 12 in the Appendix.*

Lemma 1 characterizes the expected per-iterate descent of the implied UL objective value, which depends on i) the consensus error of the UL parameters $\|\bar{\mathbf{x}}_t - \mathbf{x}_{i,t}\|^2$, ii) the approximation error of the LL optimal parameter $\|\mathbf{y}_{i,t} - \mathbf{y}_{\mu_t}^*(\mathbf{x}_{i,t})\|^2$, iii) the approximation error of the dual parameter $\|\mathbf{v}_{i,t} - \mathbf{v}_{\mu_t}^*(\mathbf{x}_{i,t})\|^2$, and iv) the diminishing speed of the augmented LL objective regularization parameter $\mu_t$.

- The second challenge comes from the fact that DUET employs a decentralized consensus update mechanism for the UL model parameters as shown in (5), which inherently leads to consensus errors. Thanks to our algorithmic design in DUET, the graph topology of the underlying network does *not* theoretically affect the convergence rate order of DUET (i.e., the $T$-dependence in the Big-O convergence rate result in Theorem 1). Also, we achieve the $\mu_t$-convergence rate, which depends on the decay rate $\tau$ of the step-size for LL $y_i$-variables and the decay rate $p$ of regularization parameter $\mu_t$. This is a new result compared to those obtained from LLSC.

- We note that the augmented LL objective (4) for each agent also allows us to relax many restrictive assumptions made in traditional bilevel optimization methods (e.g., Liu et al. (2023a), etc.): i) we do not require strong convexity for the UL objective at every agent; ii) we relax the requirement of

the derivatives $\nabla^2_{\mathbf{xy}} f_i$, $\nabla^2_{\mathbf{yy}} f_i$ being Lipschitz continuous with respect to $\mathbf{x}_i$ and $\mathbf{y}_i$, respectively; and (iii) we also relax the requirement of a bounded dataset of $(\mathbf{x}_i, \mathbf{y}_i)$. These relaxations make our approach more flexible and practical in decentralized settings with data heterogeneity.

**Discussions:** As mentioned earlier, in this paper, we have tried to avoid imposing any extra restrictive assumptions in the absence of LLSC. However, it is interesting and insightful to compare the performance of our DUET algorithm with those who do make extra assumptions. For example, it turns out that the sl-BAMM method (Liu et al., 2023a), which assumes a UL strongly convex (ULSC) objective, can be generalized to the decentralized setting as a baseline for comparisons in our experiments. We name this extension as decentralized sl-BAMM with gradient tracking (DSGT). DSGT adopts the same single-loop framework as DUET but utilizes a different augmented LL objective function, which follows Liu et al. (2023a) to aggregate the UL and LL objectives for every agent as follows: $\psi^i_{\mu_t}(\mathbf{x}_{i,t}, \mathbf{y}_{i,t}) := \mu_t f_i(\mathbf{x}_{i,t}, \mathbf{y}_{i,t}) + (1 - \mu_t) g_i(\mathbf{x}_{i,t}, \mathbf{y}_{i,t})$. For DSGT, we make the following extra ULSC assumption for every agent:

**Assumption 3** (ULSC Assumption for DSGT). *(a) For any $i \in [m]$ and fixed $\mathbf{x}_i$, UL objective $f_i(\mathbf{x}_i, \cdot)$ is $\sigma_{f_i}$ -strongly convex. (b) For any $i \in [m]$, the derivatives $\nabla^2_{\mathbf{xy}} f_i$, $\nabla^2_{\mathbf{yy}} f_i$ are Lipschitz continuous in $(\mathbf{x}_i, \mathbf{y}_i)$ with respective Lipschitz constants $L_{f_{i\mathbf{xy1}}}, L_{f_{i\mathbf{xy2}}}, L_{f_{i\mathbf{yy1}}}, L_{f_{i\mathbf{yy2}}}$.*

With Assumption 3, we can show the following convergence result for sl-BAMM (the proof of Theorem 5 is similar to the proofs of Theorem 1 and 3 and hence omitted for brevity.)

**Theorem 5** (Convergence Analysis for DSGT). *Under Assumptions 1 and 3, choose $\mu_t = \bar{\mu}(t + 1)^{-p}$. Choose the step-sizes as $\beta_t \in [\bar{\beta}(t + 1)^{-\tau/4}, 1/L_{f_{iy2}} + L_{g_{y2}}]$ with $p \in (0, 1/6)$, $\tau \in (0, 2/33)$ and $\eta_t = (t + 1)^{-\tau/2} \beta_t \mu_t$, and $\alpha_t = (t + 1)^{-3\tau/2} \beta_t \mu_t^5$. It then holds that $\min_{0 \le t \le T} \Pi(\mathbf{x}_t, \mathbf{y}_t, \mathbf{v}_t) = O(1/T^{1 - 5p - \frac{11}{4}\tau})$. Further, if $\mathbf{v}_{i,t}$ is bounded, it holds that $\min_{0 \le t \le T} \mathrm{KKT}(\mathbf{x}_t, \mathbf{y}_t, \mathbf{v}_t) = O(1/T^{1 - 5p - \frac{11}{4}\tau} + 1/T^{2p})$.*

We also notice a recent work called LV-HBA (Yao et al., 2024), which considers a more generic case where the lower-level problem includes equality or inequality constraints $g(\mathbf{x}, \mathbf{y})$, and proposes a value function-based proximal Lagrangian method to enforce the constraints with a provable rate of $\mathcal{O}(1/K^{(1-2p)/2})$. However, this convergence rate and our convergence rate are *not comparable* due to *different stationary measure*. Our stationary measure is define in Eq (8). In contrast, the stationary measure $R_k$ of LV-HBA is defined as following: $R_k := \mathrm{dist}\left(0, (\nabla F(\mathbf{x}, \mathbf{y}), 0) + c_k\left((\nabla f(\mathbf{x}, \mathbf{y}), 0) - \nabla v_{\gamma, r}(\mathbf{x}, \mathbf{y}, \mathbf{z})\right) + N_{C \times Z}(\mathbf{x}, \mathbf{y}, \mathbf{z})\right)$, where $F(\mathbf{x}, \mathbf{y})$ is the UL objective, $f(\mathbf{x}, \mathbf{y})$ is the LL objective and $v_{\gamma, r}(\mathbf{x}, \mathbf{y}, \mathbf{z})$ is truncated proximal Lagrangian value function. $c_k$ is penalty parameter. $N_\Omega(s)$ denotes the normal cone to $\Omega$ at $s$.

## 5 NUMERICAL EXPERIMENTS

In this section, we conduct numerical experiments to verify our theoretical results for DUET. Due to the lack of existing algorithms for solving decentralized bilevel optimization problems without LLSC assumption, we compare the convergence performance of DUET and DSGT.

**1) A Pedagogical Example:** We first verify the convergence results under the ULSC and non-LLSC cases using five-agent communication networks, with the network edge connection probability $p_c = 0.5$. The decentralized bilevel optimization problem is defined as the following: $\min_{\mathbf{x} \in \mathbb{R}^n} \frac{1}{2}\|\mathbf{x} - \mathbf{y}_2^*\|^2 + \frac{1}{2}\|\mathbf{y}_1^* - \mathbf{e}\|^2$ s.t $\mathbf{y}^* = (\mathbf{y}_1^*, \mathbf{y}_2^*) \in \arg\min_{(\mathbf{y}_1, \mathbf{y}_2) \in \mathbb{R}^{2n}} \frac{1}{2}\|\mathbf{y}_1\|^2 + \mathbf{x}^\top \mathbf{y}_2$, where $\mathbf{e}$ denotes the all-one vector with dimensionality being clear from the context. As shown in Figs. 1(a) and 1(b), the gradients of $\mathbf{x}$, $\mathbf{y}$, reach zero when using our DUET algorithm, suggesting they can converge to the global optimal solution without the LLSC assumption. Note that we use "GT=1" and "GT=0" to denote the adoption of gradient tracking in the algorithm or otherwise, respectively. As shown in Figs. 1(a) and 1(b), the gradients of $\mathbf{x}$, $\mathbf{y}$ converge more rapidly when gradient tracking is adopted. However, as observed in Fig. 1(d), DSGT tends to select an LL solution $\mathbf{y} \in \mathcal{S}(\mathbf{x})$ that also yields a good value for the UL objective function (i.e., $f_i(\mathbf{x}, \cdot)$), which is due to the ULSC assumption. In contrast, as shown in Fig. 1(c), DUET tends to choose an LL solution $\mathbf{y} \in \mathcal{S}(\mathbf{x})$ that has a good LL objective value, which is more relevant in DBO problems.

**2) Decentralized Meta-learning Problems with Real-World Data:** Next, we evaluate our DUET algorithm on decentralized meta-learning problems with heterogeneous datasets. Following

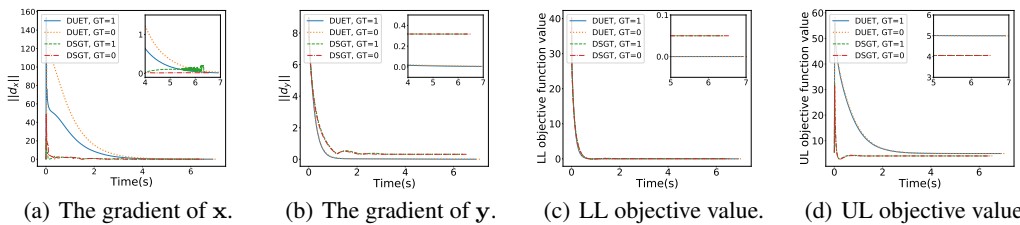

(a) The gradient of **x**.  (b) The gradient of **y**.  (c) LL objective value.  (d) UL objective value.

Figure 1: The gradients of the variables **x** and **y**, and the objective values of the UL and LL problems.

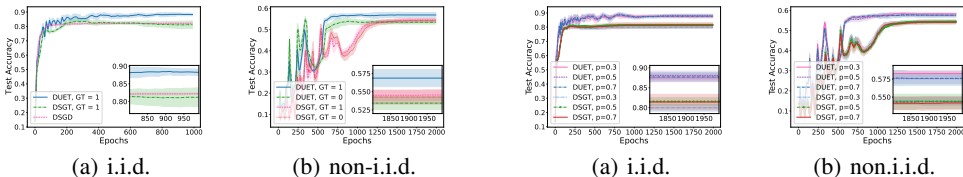

(a) i.i.d.  (b) non-i.i.d.  (a) i.i.d.  (b) non-i.i.d.

Figure 2: Test accuracy on the meta-learning problem with a 5-agent network on MNIST.

Figure 3: Test accuracy on random graphs and the meta-learning problem on MNIST.

the experimental settings in Qiu et al. (2023), we use the MNIST dataset to train $m$ personalized classifiers. In this setup, **x** represents the common parameters shared across all agents, while $\theta$ corresponds to task-specific parameters for each agent. The cross-entropy loss is employed as the main objective function for $f_i$. To prevent overfitting on local samples, we incorporate a quadratic regularization term in the UL objective function $f_i$, affecting the parameters $\theta$. We set $m = 5$ and allocate 12,000 samples to each node in a communication network generated by the random Erdős–Rényi graphs. In the scenario with i.i.d. data, all agents have access to the same global dataset. We compare our proposed algorithms with the baseline DSGD. In Fig. 2(a), the DUET algorithm demonstrates superior performance by achieving the highest testing accuracy, along with fast convergence. Notably, its testing accuracy is 6% higher than that of the standard baseline based on decentralized stochastic gradient descent. This performance highlights the advantages of DUET in collaborative training across multiple learners and in tailored adaptation at each node.

In contrast, in the non-i.i.d. data scenario, we adopt a data partitioning strategy, where each agent accesses data consisting of 95% from two specific labels and 5% randomly selected from other labels. This strategy ensures significantly diverse label distributions and high data heterogeneity across nodes. In the non-i.i.d. data scenario, we compare DUET and DSGT by focusing on their configurations with gradient tracking (GT=1) and without gradient tracking (GT=0). This ablation study examines the benefits of incorporating gradient tracking. As shown in Fig. 2(b), both testing accuracy is higher for all algorithms that include gradient tracking. Specifically, the DUET and DSGT algorithms with gradient tracking significantly outperform their counterparts without gradient tracking, illustrating the effectiveness of gradient tracking in enhancing model performance in environments with heterogeneous data. We further evaluate the impact of edge connection probability $p_c$ on the performance of DUET and DSGT in both i.i.d. and non-i.i.d. settings with a five-agent network. Using $p_c \in \{0.3, 0.5, 0.7\}$ and the same learning rates as before (Fig. 3), we observe only a slight improvement in convergence with higher $p_c$, indicating that our DUET algorithm is robust to different edge connection probabilities and adaptable to various network configurations. To further validate our algorithm, additional experiments on decentralized hyperparameter optimization and decentralized meta learning are presented in Appendix E.3 and E.2.

## 6 CONCLUSION

In this paper, we explored decentralized bilevel optimization (DBO) problems that do not require lower-level strong convexity (LLSC). To address this challenge, we proposed a single-loop DBO algorithm called quadratically regularized bilevel optimization (DUET) and established its theoretical convergence rate performance. Additionally, we adopted a single-loop structure in DUET to more effectively manage the challenges posed by nested structures, and we incorporated gradient tracking to tackle data heterogeneity. We also conducted numerical experiments to validate the effectiveness and efficiency of our proposed DUET algorithm.

ACKNOWLEDGMENTS

The work has been supported in part by NSF grants CAREER CNS-2110259, CNS-2112471, ExpandAI-2324052 RINGS-2148253, ECCS-2413528, DARPA Young Faculty Award D24AP00265, ONR grant N00014-24-1-2729, and AFRL grant PGSC-SC-111374-19s.

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

## A  NOTATION

We summarize our notation in the following table.

| Notation | Definition |
|---|---|
| $\mathcal{N}$ | Set of nodes. |
| $\mathcal{L}$ | Set of edges. |
| $m$ | Number of agents. |
| $i$ | $i$-th agent. |
| $T$ | Total iteration numbers. |
| $t$ | $t$-th iteration. |
| $\alpha$ | Step size for updating $\mathbf{x}$. |
| $\beta$ | Step size for updating $\mathbf{y}$. |
| $\eta$ | Step size for updating $\mathbf{v}$. |
| $p$ | Averaging control parameter. |
| $\tau$ | Learning rate control parameter for updating $\mathbf{y}$. |
| $\mu$ | Regularization parameter for the LL objective. |
| $L_{f_{i_0}}$ | Lipschitz constant of the UL objective $f_i$, see Assumption 1. |
| $L_{f_{i\mathbf{x}1}}, L_{f_{i\mathbf{x}2}}$ | Lipschitz constants for $\nabla_{\mathbf{x}} f_i(\cdot, \mathbf{y}_i)$ and $\nabla_{\mathbf{x}} f_i(\mathbf{x}_i, \cdot)$, see Assumption 1. |
| $L_{f_{i\mathbf{y}1}}, L_{f_{i\mathbf{y}2}}$ | Lipschitz constants for $\nabla_{\mathbf{y}} f_i(\cdot, \mathbf{y}_i)$ and $\nabla_{\mathbf{y}} f_i(\mathbf{x}_i, \cdot)$, see Assumption 1. |
| $f_{i_0}$ | Uniform lower bound of the UL objective $f_i$, see Assumption 1. |
| $L_{g_{i\mathbf{y}1}}, L_{g_{i\mathbf{y}2}}$ | Lipschitz constants for $\nabla_{\mathbf{y}} g_i$, see Assumption 2. |
| $L_{g_{i\mathbf{xy}1}}, L_{g_{i\mathbf{xy}2}}$ | Lipschitz constants for $\nabla^2_{\mathbf{xy}} g_i$, see Assumption 2. |
| $L_{g_{i\mathbf{yy}1}}, L_{g_{i\mathbf{yy}2}}$ | Lipschitz constants for $\nabla^2_{\mathbf{yy}} g_i$, see Assumption 2. |
| $\sigma_{\psi_{\mu_t}}$ | Strongly convex constant of the augmented LL objective $\psi^i_{\mu_t}(\mathbf{x}_i, \cdot)$. |
| $g_i$ | LL objective, convex and smooth, see Assumption 2. |
| $f_i$ | UL objective, bounded and smooth, see Assumption 1. |
| $M$ | Consensus weight matrix $M \in \mathbb{R}^{N \times N}$ for the decentralized network. |
| $\lambda$ | Second largest eigenvalue of the consensus matrix $M$. |
| $\mathbf{x}_{i,t}$ | UL variable for agent $i$ at iteration $t$. |
| $\mathbf{y}_{i,t}$ | LL variable for agent $i$ at iteration $t$. |
| $\mathbf{v}_{i,t}$ | Dual variable for agent $i$ at iteration $t$. |
| $\nabla \mathcal{L}$ | Gradient of the Lagrangian function. |
| $\nabla \Phi$ | Gradient of the total UL objective. |

Table 2: Notation Table

Also, we formally define the communication complexity of a decentralized algorithm as follows:

**Definition 1.** *(Communication Complexity) : The communication complexity is defined as the total number of communication rounds needed to converge to an $\epsilon$-stationary point. In each round, every node can send and receive vector-valued information to and from its neighboring nodes.*

## B  EXPANDED RELATED WORK

**1) Decentralized Bilevel Optimization (DBO):** One line of works enforce consensus on the LL variables. For example, Chen et al. (2022) employed hypergradient estimation with gradient tracking for both deterministic and stochastic algorithms, achieving convergence rates of $\mathcal{O}(1/K)$ and $\mathcal{O}(1/\sqrt{K})$ for nonconvex UL and strongly-convex LL problems. However, enforcing LL consensus adds a costly inner subroutine for decentralized Hessian inverse operation, significantly increasing the communication costs. Lu et al. (2022) explored stochastic bilevel optimization algorithms with primal-dual updates and demonstrated linear convergence speedup with respect to the number of nodes in homogeneous data settings. However, this method does not extend to non-i.i.d. environments, which are common in decentralized systems. For non-i.i.d. data, Yang et al. (2022) introduced a gossip protocol-based decentralized bilevel algorithm that achieves linear speedup while accounting for the communication graph's spectral gap. Additionally, Dong et al. (2024) proposed a single-loop algorithm that does not require gradient heterogeneity assumptions. Similarly, Zhang et al. (2023)

introduced decentralized stochastic bilevel gradient descent algorithms for heterogeneous settings, illustrating how communication for hypergradient estimation affects convergence.

**2) Centralized Bilevel Optimization without LLSC:** Merchav & Sabach (2023) addressed convex bilevel optimization problems with nonsmooth outer objectives, introducing a generalized sub-gradient method to the bilevel setting and achieves sub-linear convergence rates for outer and inner objectives under mild assumptions. They additionally improves to a linear rate for strongly convex outer objectives while allowing for nonsmoothness. In contrast, our work specifically deals with smooth UL objectives. Chen et al. (2024b) proposed a penalty-based algorithm that leverages Hölderian error bounds to achieve convergence rates of $O(1/\epsilon^2)$, assuming a convex UL objective. Cao et al. (2024) introduced an accelerated gradient method specifically designed for bilevel problems with a convex lower-level problem, achieving convergence rates of $O(1/\sqrt{\epsilon})$ for smooth convex objectives. Samadi et al. (2023) achieved optimal complexity guarantees for a class of convex bilevel problems using advanced regularization techniques and strong convexity assumptions. While their works (Merchav & Sabach, 2023; Chen et al., 2024b; Cao et al., 2024; Samadi et al., 2023) assume UL convexity, our work focuses on non-convex UL objectives, broadening the applicability to more general problem settings.

## C  PROOF OF MAIN RESULTS

Before diving into our theoretical analysis, we first introduce the following notations.

Let $\{a_t, b_t, c_t, d_t\}_{t=1}^{\infty}$ be a sequence of decreasing positive constants. The general form of potential function is given by

$$V_t := a_t \left[ f(\bar{\mathbf{x}}_t, \mathbf{y}_{\mu_t}^*(\bar{\mathbf{x}}_t)) - f_0 \right] + d_t \|\mathbf{x}_t - \mathbf{1} \otimes \bar{\mathbf{x}}_t\|^2 + d_t \alpha_t \|\mathbf{h}_{\mathbf{x}}^t - \mathbf{1} \otimes \bar{\mathbf{h}}_{\mathbf{x}}^t\|^2$$
$$+ b_t \|\mathbf{y}_t - \mathbf{y}_{\mu_t}^*(\mathbf{x}_t)\|^2 + c_t \|\mathbf{v}_t - \mathbf{v}_{\mu_t}^*(\mathbf{x}_t)\|^2.$$

We also introduce the following notations:

$$\bar{\mathbf{x}}_t = \frac{1}{m} \sum_{i=1}^{m} \mathbf{x}_{i,t}, \qquad \mathbf{x}_t = [\mathbf{x}_{1,t}^\top, \ldots, \mathbf{x}_{m,t}^\top]^\top,$$
$$\mathbf{y}_t = [\mathbf{y}_{1,t}^\top, \ldots, \mathbf{y}_{m,t}^\top]^\top, \qquad \mathbf{v}_t = [\mathbf{v}_{1,t}^\top, \ldots, \mathbf{v}_{m,t}^\top]^\top.$$

The proofs of the theorem involve four major steps, which include: (1) upper-bounding the descent of the total UL objective; (2) controlling both LL solution and multiplier errors in the descent of the total UL objective; (3) controlling both the consensus steps and the gradient tracking steps; and (4) combining all results from the previous steps and choosing suitable coefficients $\{a_t, b_t, c_t, d_t\}_{t=1}^{\infty}$ for the Lyapunov function to prove the convergence guarantee.

**Step 1: Upper-bounding the descent of the total UL objective.** We begin by analyzing the descent of the total upper-level objective $\Phi_{\mu_t}(\bar{\mathbf{x}}_t)$, which is the average of the individual UL objectives $\Phi_{\mu_t}^i(\bar{\mathbf{x}}_t)$. Our goal is to establish an upper bound on the change in this objective between iterations $t$ and $t+1$. The descent of the approximate overall UL objective is $\Phi_{\mu_t}(\bar{\mathbf{x}}_t) = \frac{1}{m} \sum_{i=1}^{m} \Phi_{\mu_t}^i(\bar{\mathbf{x}}_t)$, where $\Phi_{\mu_t}^i(\bar{\mathbf{x}}_t) = f_i(\bar{\mathbf{x}}_t, \mathbf{y}_{\mu_t}^*(\bar{\mathbf{x}}_t))$.

**Lemma 2.** *Suppose Assumptions holds. Let $\mu_{t+1} \leq \mu_t \leq \frac{1}{2}$ for all $k$. Then the sequence of $\mathbf{x}_{i,t}, \mathbf{y}_{i,t}, \mathbf{v}_{i,t}$ generated by our algorithm satisfy*

$$\Phi_{\mu_{t+1}}(\bar{\mathbf{x}}_{t+1}) - \Phi_{\mu_t}(\bar{\mathbf{x}}_t)$$

$$\leq -\frac{\alpha_t}{2} \|\nabla \Phi_{\mu_t}(\bar{\mathbf{x}}_t)\|^2 - \left( \frac{\alpha_t}{2} - \frac{\alpha_t^2 L_{\Phi \mu_t}}{2\sigma_{\psi \mu_t}^2} \right) \|\bar{\mathbf{h}}_{\mathbf{x}}^t\|^2 + \frac{\alpha_t L_{\Phi \mu_t}}{r \sigma_{\psi \mu_t}^2} \frac{1}{m} \sum_{i=1}^{m} \|\bar{\mathbf{x}}_t - \mathbf{x}_{i,t}\|^2$$

$$+ \frac{2\alpha_t r}{m} \sum_{i=1}^{m} \left( L_{\psi \mathbf{x} \mathbf{y} 2} \|\mathbf{v}_{\mu_t}^*(\mathbf{x}_{i,t})\| + L_{f_i \mathbf{x} 2} \right)^2 \|\mathbf{y}_{i,t} - \mathbf{y}_{\mu_t}^*(\mathbf{x}_{i,t})\|^2$$

$$+ \frac{\alpha_t r L_{\psi \mathbf{y} 1}^2}{m} \sum_{i=1}^{m} \|\mathbf{v}_{i,t} - \mathbf{v}_{\mu_t}^*(\mathbf{x}_{i,t})\|^2$$

$$+ 2\alpha_t r \left\| \frac{1}{m} \sum_{i=1}^{m} \mathbf{d}_{\mathbf{x}}^{i,t} - \bar{\mathbf{h}}_{\mathbf{x}}^t \right\|^2$$

$$+ \frac{1}{m} \sum_{i=1}^{m} \left[ \frac{2 \left\| \nabla_{\mathbf{y}} f_i \left( \bar{\mathbf{x}}_{t+1}, \mathbf{y}^*_{\mu_{t+1}} (\bar{\mathbf{x}}_{t+1}) \right) \right\| \left\| \mathbf{y}^*_{\mu_{t+1}} (\bar{\mathbf{x}}_{t+1}) \right\|}{\sigma_{h_i}} \left( \frac{\mu_t - \mu_{t+1}}{\mu_t} \right) \right]$$

$$+ \frac{1}{m} \sum_{i=1}^{m} \frac{2 L_{f_i \mathbf{y} 2} \| \mathbf{y}^*_{\mu_{t+1}} (\bar{\mathbf{x}}_{t+1}) \|^2}{\sigma_{h_i}^2} \left( \frac{\mu_t - \mu_{t+1}}{\mu_t} \right)^2,$$

*where both $C_{\Phi 1}$ and $C_{\Phi 2}$ are constants given by*

$$C_{\Phi 1} := L_{\psi \mathbf{y} 1}^2 L_{\psi \mathbf{y} \mathbf{y} 2} + L_{\psi \mathbf{y} 1} \left( L_{\psi \mathbf{y} \mathbf{y} 1} + L_{\psi \mathbf{x} \mathbf{y} 1} \right) \sigma_{\psi_\mu} + L_{\psi \mathbf{x} \mathbf{y} 1} \sigma_{\psi_\mu}^2$$

$$C_{\Phi 2} := L_{\psi \mathbf{y} 1}^2 L_{f_i \mathbf{y} 2} + L_{\psi \mathbf{y} 1} \left( L_{f_i \mathbf{y} 1} + L_{f_i \mathbf{x} 2} \right) \sigma_{\psi_\mu} + L_{f_i \mathbf{x} 1} \sigma_{\psi_\mu}^2.$$

*Proof.* We decompose the difference $\Phi^i_{\mu_{t+1}} (\bar{\mathbf{x}}_{t+1}) - \Phi^i_{\mu_t} (\bar{\mathbf{x}}_t)$ into two terms:

$$\Phi^i_{\mu_{t+1}} (\bar{\mathbf{x}}_{t+1}) - \Phi^i_{\mu_t} (\bar{\mathbf{x}}_t)$$

$$= f_i \left( \bar{\mathbf{x}}_{t+1}, \mathbf{y}^*_{\mu_{t+1}} (\bar{\mathbf{x}}_{t+1}) \right) - f_i \left( \bar{\mathbf{x}}_t, \mathbf{y}^*_{\mu_t} (\bar{\mathbf{x}}_t) \right)$$

$$= f_i \left( \bar{\mathbf{x}}_{t+1}, \mathbf{y}^*_{\mu_{t+1}} (\bar{\mathbf{x}}_{t+1}) \right) - f_i \left( \bar{\mathbf{x}}_{t+1}, \mathbf{y}^*_{\mu_t} (\bar{\mathbf{x}}_{t+1}) \right) + f_i \left( \bar{\mathbf{x}}_{t+1}, \mathbf{y}^*_{\mu_t} (\bar{\mathbf{x}}_{t+1}) \right) - f_i \left( \bar{\mathbf{x}}_t, \mathbf{y}^*_{\mu_t} (\bar{\mathbf{x}}_t) \right).$$

By the smoothness of $f_i(\mathbf{x}, \cdot)$ and Cauchy-Schwarz inequality, we have

$$\frac{1}{m} \sum_{i=1}^{m} \left[ f_i \left( \bar{\mathbf{x}}_{t+1}, \mathbf{y}^*_{\mu_{t+1}} (\bar{\mathbf{x}}_{t+1}) \right) - f_i \left( \bar{\mathbf{x}}_{t+1}, \mathbf{y}^*_{\mu_t} (\bar{\mathbf{x}}_{t+1}) \right) \right]$$

$$\leq \frac{1}{m} \sum_{i=1}^{m} \left[ \left\langle \nabla_{\mathbf{y}} f_i \left( \bar{\mathbf{x}}_{t+1}, \mathbf{y}^*_{\mu_{t+1}} (\bar{\mathbf{x}}_{t+1}) \right), \mathbf{y}^*_{\mu_{t+1}} (\bar{\mathbf{x}}_{t+1}) - \mathbf{y}^*_{\mu_t} (\bar{\mathbf{x}}_{t+1}) \right\rangle \right]$$

$$+ \frac{1}{m} \sum_{i=1}^{m} \frac{L_{f_i \mathbf{y} 2}}{2} \| \mathbf{y}^*_{\mu_{t+1}} (\bar{\mathbf{x}}_{t+1}) - \mathbf{y}^*_{\mu_t} (\bar{\mathbf{x}}_{t+1}) \|^2$$

$$\leq \frac{1}{m} \sum_{i=1}^{m} \left[ \left\| \nabla_{\mathbf{y}} f_i \left( \bar{\mathbf{x}}_{t+1}, \mathbf{y}^*_{\mu_{t+1}} (\bar{\mathbf{x}}_{t+1}) \right) \right\| \cdot \left\| \mathbf{y}^*_{\mu_{t+1}} (\bar{\mathbf{x}}_{t+1}) - \mathbf{y}^*_{\mu_t} (\bar{\mathbf{x}}_{t+1}) \right\| \right]$$

$$+ \frac{1}{m} \sum_{i=1}^{m} \frac{L_{f_i \mathbf{y} 2}}{2} \| \mathbf{y}^*_{\mu_{t+1}} (\bar{\mathbf{x}}_{t+1}) - \mathbf{y}^*_{\mu_t} (\bar{\mathbf{x}}_{t+1}) \|^2$$

$$\leq \frac{1}{m} \sum_{i=1}^{m} \left[ \frac{2 \left\| \nabla_{\mathbf{y}} f_i \left( \bar{\mathbf{x}}_{t+1}, \mathbf{y}^*_{\mu_{t+1}} (\bar{\mathbf{x}}_{t+1}) \right) \right\| \left\| \mathbf{y}^*_{\mu_{t+1}} (\bar{\mathbf{x}}_{t+1}) \right\|}{\sigma_{h_i}} \left( \frac{\mu_t - \mu_{t+1}}{\mu_t} \right) \right]$$

$$+ \frac{1}{m} \sum_{i=1}^{m} \frac{2 L_{f_i \mathbf{y} 2} \| \mathbf{y}^*_{\mu_{t+1}} (\bar{\mathbf{x}}_{t+1}) \|^2}{\sigma_{h_i}^2} \left( \frac{\mu_t - \mu_{t+1}}{\mu_t} \right)^2, \tag{10}$$

where the last inequality follows from $\left\| \mathbf{y}^*_\mu(\mathbf{x}) - \mathbf{y}^*_{\mu'}(\mathbf{x}) \right\| \leq \frac{2 \| \mathbf{y}^*_\mu(\mathbf{x}) \|}{\sigma_{h_i}} \cdot \frac{|\mu - \mu'|}{\mu'}$ with $\mu' = \mu_t$ and $\mu = \mu_{t+1}$.

Next, let $L_{\Phi \mu_t} := C_{\Phi 1} \left\| \mathbf{v}^*_{\mu_t} (\bar{\mathbf{x}}_t) \right\| + C_{\Phi 2}$, we have

$$\Phi_{\mu_t} (\bar{\mathbf{x}}_{t+1}) - \Phi_{\mu_t} (\bar{\mathbf{x}}_t) = \frac{1}{m} \sum_{i=1}^{m} \Phi^i_{\mu_t} (\bar{\mathbf{x}}_{t+1}) - \frac{1}{m} \sum_{i=1}^{m} \Phi^i_{\mu_t} (\bar{\mathbf{x}}_t)$$

$$= \frac{1}{m} \sum_{i=1}^{m} \left[ f_i \left( \bar{\mathbf{x}}_{t+1}, \mathbf{y}^*_{\mu_t} (\bar{\mathbf{x}}_{t+1}) \right) - f_i \left( \bar{\mathbf{x}}_t, \mathbf{y}^*_{\mu_t} (\bar{\mathbf{x}}_t) \right) \right]$$

$$\leq \frac{1}{m}\sum_{i=1}^{m}\left\langle \nabla\Phi_{\mu_t}^i\left(\bar{\mathbf{x}}_t\right),\bar{\mathbf{x}}_{t+1}-\bar{\mathbf{x}}_t\right\rangle + \frac{1}{m}\sum_{i=1}^{m}\frac{L_{\Phi\mu_t}}{2\sigma_{\psi_{\mu_t}}^2}\left\|\bar{\mathbf{x}}_{t+1}-\bar{\mathbf{x}}_t\right\|^2. \tag{11}$$

We apply the gradient descent update step

$$\bar{\mathbf{x}}_{t+1}-\bar{\mathbf{x}}_t = -\alpha_t\bar{\mathbf{h}}_{\mathbf{x}}^t. \tag{12}$$

Incorporating (12) into (11), we get

$$\Phi_{\mu_t}\left(\bar{\mathbf{x}}_{t+1}\right)-\Phi_{\mu_t}\left(\bar{\mathbf{x}}_t\right)$$

$$\leq -\alpha_t\left\langle \frac{1}{m}\sum_{i=1}^{m}\nabla\Phi_{\mu_t}^i\left(\bar{\mathbf{x}}_t\right),\bar{\mathbf{h}}_{\mathbf{x}}^t\right\rangle + \frac{\alpha_t^2 L_{\Phi\mu_t}}{2\sigma_{\psi_{\mu_t}}^2}\left\|\bar{\mathbf{h}}_{\mathbf{x}}^t\right\|^2$$

$$= -\frac{\alpha_t}{2}\left\|\nabla\Phi_{\mu_t}\left(\bar{\mathbf{x}}_t\right)\right\|^2 - \left(\frac{\alpha_t}{2}-\frac{\alpha_t^2 L_{\Phi\mu_t}}{2\sigma_{\psi_{\mu_t}}^2}\right)\left\|\bar{\mathbf{h}}_{\mathbf{x}}^t\right\|^2 + \frac{\alpha_t}{2}\left\|\nabla\Phi_{\mu_t}\left(\bar{\mathbf{x}}_t\right)-\bar{\mathbf{h}}_{\mathbf{x}}^t\right\|^2.$$

We could further get

$$\Phi_{\mu_t}\left(\bar{\mathbf{x}}_{t+1}\right)-\Phi_{\mu_t}\left(\bar{\mathbf{x}}_t\right)$$

$$\leq -\frac{\alpha_t}{2}\left\|\nabla\Phi_{\mu_t}\left(\bar{\mathbf{x}}_t\right)\right\|^2 - \left(\frac{\alpha_t}{2}-\frac{\alpha_t^2 L_{\Phi\mu_t}}{2\sigma_{\psi_{\mu_t}}^2}\right)\left\|\bar{\mathbf{h}}_{\mathbf{x}}^t\right\|^2$$

$$+ \frac{\alpha_t}{2}\left\|\nabla\Phi_{\mu_t}\left(\bar{\mathbf{x}}_t\right)-\frac{1}{m}\sum_{i=1}^{m}\nabla\Phi_{\mu_t}^i\left(\mathbf{x}_{i,t}\right)+\frac{1}{m}\sum_{i=1}^{m}\nabla\Phi_{\mu_t}^i\left(\mathbf{x}_{i,t}\right)-\bar{\mathbf{h}}_{\mathbf{x}}^t\right\|^2$$

$$\leq -\frac{\alpha_t}{2}\left\|\nabla\Phi_{\mu_t}\left(\bar{\mathbf{x}}_t\right)\right\|^2 - \left(\frac{\alpha_t}{2}-\frac{\alpha_t^2 L_{\Phi\mu_t}}{2\sigma_{\psi_{\mu_t}}^2}\right)\left\|\bar{\mathbf{h}}_{\mathbf{x}}^t\right\|^2$$

$$+ \frac{\alpha_t}{r}\left\|\nabla\Phi_{\mu_t}\left(\bar{\mathbf{x}}_t\right)-\frac{1}{m}\sum_{i=1}^{m}\nabla\Phi_{\mu_t}^i\left(\mathbf{x}_{i,t}\right)\right\|^2 + \alpha_t r\left\|\frac{1}{m}\sum_{i=1}^{m}\nabla\Phi_{\mu_t}^i\left(\mathbf{x}_{i,t}\right)-\bar{\mathbf{h}}_{\mathbf{x}}^t\right\|^2$$

$$\leq -\frac{\alpha_t}{2}\left\|\nabla\Phi_{\mu_t}\left(\bar{\mathbf{x}}_t\right)\right\|^2 - \left(\frac{\alpha_t}{2}-\frac{\alpha_t^2 L_{\Phi\mu_t}}{2\sigma_{\psi_{\mu_t}}^2}\right)\left\|\bar{\mathbf{h}}_{\mathbf{x}}^t\right\|^2 + \frac{\alpha_t L_{\Phi\mu_t}}{r\sigma_{\psi_{\mu_t}}^2}\frac{1}{m}\sum_{i=1}^{m}\left\|\bar{\mathbf{x}}_t-\mathbf{x}_{i,t}\right\|^2$$

$$+ \alpha_t r\left\|\frac{1}{m}\sum_{i=1}^{m}\nabla\Phi_{\mu_t}^i\left(\mathbf{x}_{i,t}\right)-\bar{\mathbf{h}}_{\mathbf{x}}^t\right\|^2.$$

Now, we consider the last term of this equation,

$$\left\|\frac{1}{m}\sum_{i=1}^{m}\nabla\Phi_{\mu_t}^i\left(\mathbf{x}_{i,t}\right)-\bar{\mathbf{h}}_{\mathbf{x}}^t\right\|^2 = \left\|\frac{1}{m}\sum_{i=1}^{m}\nabla\Phi_{\mu_t}^i\left(\mathbf{x}_{i,t}\right)-\frac{1}{m}\sum_{i=1}^{m}\mathbf{d}_{\mathbf{x}}^{i,t}+\frac{1}{m}\sum_{i=1}^{m}\mathbf{d}_{\mathbf{x}}^{i,t}-\bar{\mathbf{h}}_{\mathbf{x}}^t\right\|^2$$

$$\leq 2\left\|\frac{1}{m}\sum_{i=1}^{m}\nabla\Phi_{\mu_t}^i\left(\mathbf{x}_{i,t}\right)-\frac{1}{m}\sum_{i=1}^{m}\mathbf{d}_{\mathbf{x}}^{i,t}\right\|^2 + 2\left\|\frac{1}{m}\sum_{i=1}^{m}\mathbf{d}_{\mathbf{x}}^{i,t}-\bar{\mathbf{h}}_{\mathbf{x}}^t\right\|^2$$

$$\leq \frac{2}{m}\sum_{i=1}^{m}\left\|\nabla\Phi_{\mu_t}^i\left(\mathbf{x}_{i,t}\right)-\mathbf{d}_{\mathbf{x}}^{i,t}\right\|^2 + 2\left\|\frac{1}{m}\sum_{i=1}^{m}\mathbf{d}_{\mathbf{x}}^{i,t}-\bar{\mathbf{h}}_{\mathbf{x}}^t\right\|^2.$$

The desired result follows from the inequality $\left(\sum_{i=1}^{r}a_i\right)^2 \leq r\sum_{i=1}^{r}a_i^2$. By the definition of $\Phi_{\mu_t}^i\left(\mathbf{x}_i\right)$, we get $\nabla\Phi_{\mu_t}^i\left(\mathbf{x}_{i,t}\right)=\nabla_{\mathbf{x}}f_i\left(\mathbf{x}_{i,t},\mathbf{y}_{\mu_t}^*\left(\mathbf{x}_{i,t}\right)\right)-\nabla_{\mathbf{xy}}^2\psi_{\mu_t}\left(\mathbf{x}_{i,t},\mathbf{y}_{\mu_t}^*\left(\mathbf{x}_{i,t}\right)\right)\mathbf{v}_{\mu_t}^*\left(\mathbf{x}_{i,t}\right)$, which implies that

$$\left\|\nabla\Phi_{\mu_t}^i\left(\mathbf{x}_{i,t}\right)-\mathbf{d}_{\mathbf{x}}^{i,t}\right\|^2$$

$$= \left\|\nabla\Phi_{\mu_t}^i\left(\mathbf{x}_{i,t}\right)-\nabla_{\mathbf{x}}f_i\left(\mathbf{x}_{i,t},\mathbf{y}_{i,t}\right)+\nabla_{\mathbf{xy}}^2\psi_{\mu_t}\left(\mathbf{x}_{i,t},\mathbf{y}_{i,t}\right)\mathbf{v}_{i,t}\right\|$$

$$\leq \left\|\nabla_{\mathbf{x}}f_i\left(\mathbf{x}_{i,t},\mathbf{y}_{\mu_t}^*\left(\mathbf{x}_{i,t}\right)\right)-\nabla_{\mathbf{x}}f_i\left(\mathbf{x}_{i,t},\mathbf{y}_{i,t}\right)\right\|$$

$$+ \left\|\left[\nabla_{\mathbf{xy}}^2\psi_{\mu_t}\left(\mathbf{x}_{i,t},\mathbf{y}_{i,t}\right)-\nabla_{\mathbf{xy}}^2\psi_{\mu_t}\left(\mathbf{x}_{i,t},\mathbf{y}_{\mu_t}^*\left(\mathbf{x}_{i,t}\right)\right)\right]\mathbf{v}_{\mu_t}^*\left(\mathbf{x}_{i,t}\right)\right\|$$

$$+ \left\| \nabla_{\mathbf{xy}}^2 \psi_{\mu_t} \left( \mathbf{x}_{i,t}, \mathbf{y}_{i,t} \right) \left[ \mathbf{v}_{i,t} - \mathbf{v}_{\mu_t}^* \left( \mathbf{x}_{i,t} \right) \right] \right\|$$

$$\leq \left( L_{f_{i\mathbf{x}2}} + L_{\psi_{\mathbf{xy}2}} \left\| \mathbf{v}_{\mu_t}^* \left( \mathbf{x}_{i,t} \right) \right\| \right) \left\| \mathbf{y}_{i,t} - \mathbf{y}_{\mu_t}^* \left( \mathbf{x}_{i,t} \right) \right\| + L_{\psi_{\mathbf{y}1}} \left\| \mathbf{v}_{i,t} - \mathbf{v}_{\mu_t}^* \left( \mathbf{x}_{i,t} \right) \right\|.$$

We now combine all terms and upper-bound the total descent of the UL objective:

$$\Phi_{\mu_{t+1}} \left( \bar{\mathbf{x}}_{t+1} \right) - \Phi_{\mu_t} \left( \bar{\mathbf{x}}_t \right)$$

$$\leq - \frac{\alpha_t}{2} \left\| \nabla \Phi_{\mu_t} \left( \bar{\mathbf{x}}_t \right) \right\|^2 - \left( \frac{\alpha_t}{2} - \frac{\alpha_t^2 L_{\Phi \mu_t}}{2 \sigma_{\psi_{\mu_t}}^2} \right) \| \bar{\mathbf{h}}_{\mathbf{x}}^t \|^2 + \frac{\alpha_t L_{\Phi \mu_t}}{r \sigma_{\psi_{\mu_t}}^2} \frac{1}{m} \sum_{i=1}^m \| \bar{\mathbf{x}}_t - \mathbf{x}_{i,t} \|^2$$

$$+ \frac{2 \alpha_t r}{m} \sum_{i=1}^m \left( L_{\psi_{\mathbf{xy}2}} \left\| \mathbf{v}_{\mu_t}^* \left( \mathbf{x}_{i,t} \right) \right\| + L_{f_{i\mathbf{x}2}} \right)^2 \left\| \mathbf{y}_{i,t} - \mathbf{y}_{\mu_t}^* \left( \mathbf{x}_{i,t} \right) \right\|^2$$

$$+ \frac{\alpha_t r L_{\psi_{\mathbf{y}1}}^2}{m} \sum_{i=1}^m \left\| \mathbf{v}_{i,t} - \mathbf{v}_{\mu_t}^* \left( \mathbf{x}_{i,t} \right) \right\|^2$$

$$+ 2 \alpha_t r \left\| \frac{1}{m} \sum_{i=1}^m \mathbf{d}_{\mathbf{x}}^{i,t} - \bar{\mathbf{h}}_{\mathbf{x}}^t \right\|^2$$

$$+ \frac{1}{m} \sum_{i=1}^m \left[ \frac{2 \left\| \nabla_{\mathbf{y}} f_i \left( \bar{\mathbf{x}}_{t+1}, \mathbf{y}_{\mu_{t+1}}^* \left( \bar{\mathbf{x}}_{t+1} \right) \right) \right\| \left\| \mathbf{y}_{\mu_{t+1}}^* \left( \bar{\mathbf{x}}_{t+1} \right) \right\|}{\sigma_{h_i}} \left( \frac{\mu_t - \mu_{t+1}}{\mu_t} \right) \right]$$

$$+ \frac{1}{m} \sum_{i=1}^m \frac{2 L_{f_{i\mathbf{y}2}} \| \mathbf{y}_{\mu_{t+1}}^* ( \bar{\mathbf{x}}_{t+1} ) \|^2}{\sigma_{h_i}^2} \left( \frac{\mu_t - \mu_{t+1}}{\mu_t} \right)^2.$$

$\square$

## Step 2: Controlling both LL solution and multiplier errors.

In this step, we analyze the errors arising from the LL variables $\mathbf{y}_{\mu_t}(\mathbf{x}_{i,t})$ and the multiplier $\mathbf{v}_{\mu_t}(\mathbf{x}_{i,t})$. These errors contribute to the overall descent of the upper-level (UL) objective function, as discussed in Lemma 2. The boundedness of $\mathbf{y}_{\mu_t}^*(\mathbf{x}_{i,t})$ and $\mathbf{v}_{\mu_t}^*(\mathbf{x}_{i,t})$ will be key to understanding the impact of these errors.

**Lemma 3.** *For $\mathbf{y}_{\mu_t}^*(\mathbf{x}_{i,t})$ and $\mathbf{v}_{\mu_t}^*(\mathbf{x}_{i,t})$, we have (recall that $r_v$ and $r_y$ are defined in (9)):*

*(a)* $\| \mathbf{v}_{\mu_t}^*(\mathbf{x}_{i,t}) \| \leq r_v^t = \frac{L_{f_{i0}}}{\sigma_{\psi_{\mu_t}}} = O(\frac{1}{\sigma_{\psi_{\mu_t}}})$,

*(b)* $\| \mathbf{y}_{\mu_t}^*(\mathbf{x}_{i,t}) \| \leq r_y^t = O(\frac{1}{\sigma_{\psi_{\mu_t}}^2})$, *for any $\mathbf{x}_{i,t}$ satisfying $\|\mathbf{x}_{i,t}\| \leq r_x^t = O(\frac{1}{\sigma_{\psi_{\mu_t}}})$.*

The assumption that $\|\mathbf{x}_{i,t}\| \leq r_x^t = O\left( \frac{1}{\sigma_{\psi_{\mu_t}}} \right)$ will be proven later, but for now, it ensures that $\mathbf{x}_{i,t}$ is bounded relative to $\sigma_{\psi_{\mu_t}}$, further supporting the use of the bounds for $\mathbf{y}_{\mu_t}^*(\mathbf{x}_{i,t})$ and $\mathbf{v}_{\mu_t}^*(\mathbf{x}_{i,t})$.

*Proof.* (a) By the $L_{f_{i0}}$-Lipschitz continuity of $f_i$ and $\sigma_{\psi_\mu}$-strong convexity of $\psi_{\mu_t}^i$ in Assumption 1, we can deduce that

$$\left\| \nabla_{\mathbf{y}} f_i(\mathbf{x}_{i,t}, \mathbf{y}_{\mu_t}^*(\mathbf{x}_{i,t})) \right\| \leq L_{f_{i0}} \text{ and } \left\| \left[ \nabla_{\mathbf{yy}}^2 \psi_{\mu_t}^i(\mathbf{x}_{i,t}, \mathbf{y}_{\mu_t}^*(\mathbf{x}_{i,t})) \right]^{-1} \right\| \leq \frac{1}{\sigma_{\psi_{\mu_t}}}.$$

For $\mathbf{v}_{\mu_t}^*(\mathbf{x}_{i,t}) = \left[ \nabla_{\mathbf{yy}}^2 \psi_{\mu_t} \left( \mathbf{x}_{i,t}, \mathbf{y}_{\mu_t}^*(\mathbf{x}_{i,t}) \right) \right]^{-1} \nabla_{\mathbf{y}} f_i \left( \mathbf{x}_{i,t}, \mathbf{y}_{\mu_t}^*(\mathbf{x}_{i,t}) \right)$, we have

$$\left\| \mathbf{v}_{\mu_t}^*(\mathbf{x}_{i,t}) \right\| \leq \left\| \left[ \nabla_{\mathbf{yy}}^2 \psi_{\mu_t} \left( \mathbf{x}_{i,t}, \mathbf{y}_{\mu_t}^*(\mathbf{x}_{i,t}) \right) \right]^{-1} \right\| \left\| \nabla_{\mathbf{y}} f_i \left( \mathbf{x}_{i,t}, \mathbf{y}_{\mu_t}^*(\mathbf{x}_{i,t}) \right) \right\| \leq \frac{L_{f_{i0}}}{\sigma_{\psi_{\mu_t}}}.$$

(b) Assume $\|\mathbf{x}_{i,t}\| \leq r_x^t$. By the optimality of $\mathbf{y}_{\mu_t}^*(\mathbf{x}_{i,t})$ and Lemme 11 (a), we take $\mathbf{x}_i' = 0$ and have

$$\| \mathbf{y}_{\mu_t}^*(\mathbf{x}_{i,t}) \| \leq \frac{L_{\psi_{\mathbf{y}1}}}{\sigma_{\psi_\mu}} \| \mathbf{x}_{i,t} - 0 \| + \| \mathbf{y}_{\mu_t}^*(0) \|$$

$$= \frac{L_{\psi_{\mathbf{y}1}}}{\sigma_{\psi_\mu}} \|\mathbf{x}_{i,t}\| + \|\mathbf{y}^*_{\mu_t}(0) - 0\|$$

$$\leq \frac{L_{\psi_{\mathbf{y}1}}}{\sigma_{\psi_\mu}} \|\mathbf{x}_{i,t}\| + \frac{1}{\sigma_{\psi_\mu}} \|\nabla_{\mathbf{y}}\psi_\mu(0, \mathbf{y}^*_{\mu_t}(0)) - \nabla_{\mathbf{y}}\psi_\mu(0,0)\|$$

$$\leq \frac{L_{\psi_{\mathbf{y}1}}}{\sigma_{\psi_\mu}} r_x^t + \frac{1}{\sigma_{\psi_{\mu_t}}} \|\nabla_{\mathbf{y}}\psi_{\mu_t}(0,0)\|,$$

where the second inequality follows from the strong convexity of $\psi_{\mu_t}(\mathbf{x}_{i,t}, \cdot)$, the last inequality follows from the optimality of $\mathbf{y}^*_{\mu_t}(0)$. $\qquad\square$

**Lemma 4.** *For any $i \in [m]$, the sequence of $\mathbf{x}_{i,t}, \mathbf{y}_{i,t}, \mathbf{v}_{i,t}$ generated by our algorithm satisfy*

$$\left\|\mathbf{y}_{i,t+1} - \mathbf{y}^*_{\mu_t}(\mathbf{x}_{i,t})\right\|^2$$
$$\leq \left(1 - \sigma_{\psi_{\mu_t}}\beta_t\right)\left\|\mathbf{y}_{i,t} - \mathbf{y}^*_{\mu_t}(\mathbf{x}_{i,t})\right\|^2 - \left(2\beta_t \frac{1}{\sigma_{\psi_{\mu_t}} + L_{\psi_{\mu_t}}} - \beta_t^2\right)\left\|\mathbf{h}^{i,t}_{\mathbf{y}}\right\|^2,$$

*and*

$$\left\|\mathbf{v}_{i,t+1} - \mathbf{v}^*_{\mu_t}(\mathbf{x}_{i,t})\right\|^2 \leq \left(1 - \eta_t\sigma_{\psi_{\mu_t}}\right)\left\|\mathbf{v}_{i,t} - \mathbf{v}^*_{\mu_t}(\mathbf{x}_{i,t})\right\|^2$$
$$+ \frac{2\eta_t}{\sigma_{\psi_{\mu_t}}}\left(L_{\psi_{\mathbf{yy}2}}\left\|\mathbf{v}^*_{\mu_t}(\mathbf{x}_{i,t})\right\| + L_{f_{i\mathbf{y}2}}\right)^2\left\|\mathbf{y}_{i,t} - \mathbf{y}^*_{\mu_t}(\mathbf{x}_{i,t})\right\|^2,$$

*if we choose $\beta_t \leq \frac{2}{\sigma_{\psi_{\mu_t}} + L_{\psi_{\mu_t}}}$ and $\eta_t \leq \frac{1}{L_{\psi_{\mu_t}}}$.*

*Proof.* We begin by analyzing the update for the LL solution, $\mathbf{y}_{i,t+1}$. Recall that the update is performed via a projected gradient step:

$$\mathbf{y}_{i,t+1} = P_{r_y^t}[\mathbf{y}_{i,t} - \beta_t\nabla_{\mathbf{y}}\psi_{\mu_t}(\mathbf{x}_{i,t}, \mathbf{y}_{i,t})],$$

where $P_{r_y^t}[\cdot]$ denotes the projection onto the feasible set with radius $r_y^t$. This projection ensures that $\mathbf{y}_{i,t+1} = \mathbf{y}_{i,t+1}(\mathbf{x}_{i,t+1})$ remains bounded, preventing it from diverging. To analyze the error $\|\mathbf{y}_{i,t+1} - \mathbf{y}^*_{\mu_t}(\mathbf{x}_{i,t})\|$, we decompose the update step:

$$\|\mathbf{y}_{i,t+1} - \mathbf{y}^*_{\mu_t}(\mathbf{x}_{i,t})\|^2 = \|P_{r_y^t}[\mathbf{y}_{i,t} - \beta_t\nabla_{\mathbf{y}}\psi_{\mu_t}(\mathbf{x}_{i,t}, \mathbf{y}_{i,t})] - \mathbf{y}^*_{\mu_t}(\mathbf{x}_{i,t})\|^2.$$

Since the projection is non-expansive, i.e., it does not increase distances, and $\left\|\mathbf{y}^*_{\mu_t}(\mathbf{x}_{i,t})\right\|^2 \leq r_y^t$ by Lemma 3, we have:

$$\|\mathbf{y}_{i,t+1} - \mathbf{y}^*_{\mu_t}(\mathbf{x}_{i,t})\|^2 \leq \|\mathbf{y}_{i,t} - \beta_t\nabla_{\mathbf{y}}\psi_{\mu_t}(\mathbf{x}_{i,t}, \mathbf{y}_{i,t}) - \mathbf{y}^*_{\mu_t}(\mathbf{x}_{i,t})\|^2.$$

Next, we expand the right-hand side:

$$\left\|\mathbf{y}_{i,t+1} - \mathbf{y}^*_{\mu_t}(\mathbf{x}_{i,t})\right\|^2$$
$$\leq \left\|\mathbf{y}_{i,t} - \mathbf{y}^*_{\mu_t}(\mathbf{x}_{i,t})\right\|^2 - 2\beta_t\left\langle\mathbf{y}_{i,t} - \mathbf{y}^*_{\mu_t}(\mathbf{x}_{i,t}), \nabla_{\mathbf{y}}\psi_{\mu_t}(\mathbf{x}_{i,t}, \mathbf{y}_{i,t})\right\rangle + \beta_t^2\left\|\nabla_{\mathbf{y}}\psi_{\mu_t}(\mathbf{x}_{i,t}, \mathbf{y}_{i,t})\right\|^2,$$

To bound the inner product term, by the $\sigma_{\psi_{\mu_t}}$-strong convexity and $L_{\psi_{\mu_t}}$-smoothness of $\psi_{\mu_t}(\mathbf{x}, \cdot)$, since $\nabla_{\mathbf{y}}\psi_{\mu_t}\left(\mathbf{x}_{i,t}, \mathbf{y}^*_{\mu_t}(\mathbf{x}_{i,t})\right) = 0$, Theorem 2.1.12 in (Nesterov, 2018) implies that

$$\left\langle\mathbf{y}_{i,t} - \mathbf{y}^*_{\mu_t}(\mathbf{x}_{i,t}), \nabla_{\mathbf{y}}\psi_{\mu_t}(\mathbf{x}_{i,t}, \mathbf{y}_{i,t})\right\rangle$$
$$= \left\langle\mathbf{y}_{i,t} - \mathbf{y}^*_{\mu_t}(\mathbf{x}_{i,t}), \nabla_{\mathbf{y}}\psi_{\mu_t}(\mathbf{x}_{i,t}, \mathbf{y}_{i,t}) - \nabla_{\mathbf{y}}\psi_{\mu_t}\left(\mathbf{x}_{i,t}, \mathbf{y}^*_{\mu_t}(\mathbf{x}_{i,t})\right)\right\rangle$$
$$\geq \frac{\sigma_{\psi_{\mu_t}}L_{\psi_{\mu_t}}}{\sigma_{\psi_{\mu_t}} + L_{\psi_{\mu_t}}}\left\|\mathbf{y}_{i,t} - \mathbf{y}^*_{\mu_t}(\mathbf{x}_{i,t})\right\|^2 + \frac{1}{\sigma_{\psi_{\mu_t}} + L_{\psi_{\mu_t}}}\left\|\nabla_{\mathbf{y}}\psi_{\mu_t}(\mathbf{x}_{i,t}, \mathbf{y}_{i,t})\right\|^2.$$

Hence, when $0 < \beta_t \leq \frac{2}{\sigma_{\psi_{\mu_t}} + L_{\psi_{\mu_t}}}$, we have

$$\left\|\mathbf{y}_{i,t+1} - \mathbf{y}^*_{\mu_t}(\mathbf{x}_{i,t})\right\|^2$$

$$\leq \left(1 - \frac{2\sigma_{\psi_{\mu_t}} L_{\psi_{\mu_t}}}{\sigma_{\psi_{\mu_t}} + L_{\psi_{\mu_t}}} \beta_t \right) \left\| \mathbf{y}_{i,t} - \mathbf{y}^*_{\mu_t}(\mathbf{x}_{i,t}) \right\|^2 - \left(2\beta_t \frac{1}{\sigma_{\psi_{\mu_t}} + L_{\psi_{\mu_t}}} - \beta_t^2 \right) \left\| \nabla_{\mathbf{y}} \psi_{\mu_t}(\mathbf{x}_{i,t}, \mathbf{y}_{i,t}) \right\|^2$$

$$= \left(1 - \frac{2\sigma_{\psi_{\mu_t}} L_{\psi_{\mu_t}}}{\sigma_{\psi_{\mu_t}} + L_{\psi_{\mu_t}}} \beta_t \right) \left\| \mathbf{y}_{i,t} - \mathbf{y}^*_{\mu_t}(\mathbf{x}_{i,t}) \right\|^2 - \left(2\beta_t \frac{1}{\sigma_{\psi_{\mu_t}} + L_{\psi_{\mu_t}}} - \beta_t^2 \right) \left\| \mathbf{h}^{i,t}_{\mathbf{y}} \right\|^2,$$

which implies the desired result since $\frac{1}{2} \leq \frac{L_{\psi_{\mu_t}}}{\sigma_{\psi_{\mu_t}} + L_{\psi_{\mu_t}}} \leq 1$. This inequality shows that the error in $\mathbf{y}_{i,t}$ contracts as long as the step size $\beta_t$ is chosen appropriately, ensuring the progress of the algorithm toward the optimal solution $\mathbf{y}^*_{\mu_t}(\mathbf{x}_{i,t})$.

Now, we proceed to bound the error in the multiplier $\mathbf{v}_{i,t+1} = \mathbf{v}_{i,t+1}(\mathbf{x}_{i,t+1})$ by using $\left[\nabla^2_{\mathbf{yy}} \psi_{\mu_t}(\mathbf{x}_{i,t}, \mathbf{y}^*_{\mu_t}(\mathbf{x}_{i,t}))\right] \mathbf{v}^*_{\mu_t}(\mathbf{x}_{i,t}) = \nabla_{\mathbf{y}} f_i(\mathbf{x}_{i,t}, \mathbf{y}^*_{\mu_t}(\mathbf{x}_{i,t}))$. As with the previous projection for $\mathbf{y}_{i,t+1}$, the non-expansiveness property of the projection operator $P_{r^t_v}[\cdot]$ gives us:

$$\left\| \mathbf{v}_{i,t+1} - \mathbf{v}^*_{\mu_t}(\mathbf{x}_{i,t}) \right\|^2$$
$$= \left\| P_{r^t_v}[\mathbf{v}_{i,t} - \eta_t \left( \left[\nabla^2_{\mathbf{yy}} \psi_{\mu_t}(\mathbf{x}_{i,t}, \mathbf{y}_{i,t})\right] \mathbf{v}_{i,t} - \nabla_{\mathbf{y}} f_i(\mathbf{x}_{i,t}, \mathbf{y}_{i,t})\right)] - \mathbf{v}^*_{\mu_t}(\mathbf{x}_{i,t}) \right\|^2$$
$$\leq \left\| \mathbf{v}_{i,t} - \mathbf{v}^*_{\mu_t}(\mathbf{x}_{i,t}) - \eta_t \left( \left[\nabla^2_{\mathbf{yy}} \psi_{\mu_t}(\mathbf{x}_{i,t}, \mathbf{y}_{i,t})\right] \mathbf{v}_{i,t} - \nabla_{\mathbf{y}} f_i(\mathbf{x}_{i,t}, \mathbf{y}_{i,t})\right) \right\|^2, \tag{13}$$

where the inequality holds because $\left\| \mathbf{v}^*_{\mu_t}(\mathbf{x}_{i,t}) \right\|^2 \leq r^t_v$ by Lemma 3. Next, we expand the right-hand side, which gives:

$$\mathbf{v}_{i,t} - \mathbf{v}^*_{\mu_t}(\mathbf{x}_{i,t}) - \eta_t \left( \left[\nabla^2_{\mathbf{yy}} \psi_{\mu_t}(\mathbf{x}_{i,t}, \mathbf{y}_{i,t})\right] \mathbf{v}_{i,t} - \nabla_{\mathbf{y}} f_i(\mathbf{x}_{i,t}, \mathbf{y}_{i,t})\right)$$
$$= \mathbf{v}_{i,t} - \mathbf{v}^*_{\mu_t}(\mathbf{x}_{i,t}) - \eta_t \left[\nabla^2_{\mathbf{yy}} \psi_{\mu_t}(\mathbf{x}_{i,t}, \mathbf{y}_{i,t})\right] \left[\mathbf{v}_{i,t} - \mathbf{v}^*_{\mu_t}(\mathbf{x}_{i,t})\right]$$
$$\quad - \eta_t \left[\nabla^2_{\mathbf{yy}} \psi_{\mu_t}(\mathbf{x}_{i,t}, \mathbf{y}_{i,t}) - \nabla^2_{\mathbf{yy}} \psi_{\mu_t}(\mathbf{x}_{i,t}, \mathbf{y}^*_{\mu_t}(\mathbf{x}_{i,t}))\right] \mathbf{v}^*_{\mu_t}(\mathbf{x}_{i,t})$$
$$\quad - \eta_t \left[\nabla_{\mathbf{y}} f_i(\mathbf{x}_{i,t}, \mathbf{y}^*_{\mu_t}(\mathbf{x}_{i,t})) - \nabla_{\mathbf{y}} f_i(\mathbf{x}_{i,t}, \mathbf{y}_{i,t})\right]. \tag{14}$$

Hence, by Cauchy-Schwarz inequality, for all $\epsilon \geq 0$, we incorporate (14) into (13) and have

$$\left\| \mathbf{v}_{i,t+1} - \mathbf{v}^*_{\mu_t}(\mathbf{x}_{i,t}) \right\|^2$$
$$\leq (1+\epsilon) \left\| \left[I - \eta_t \nabla^2_{\mathbf{yy}} \psi_{\mu_t}(\mathbf{x}_{i,t}, \mathbf{y}_{i,t})\right] \left[\mathbf{v}_{i,t} - \mathbf{v}^*_{\mu_t}(\mathbf{x}_{i,t})\right] \right\|^2$$
$$\quad + \left(1 + \frac{1}{\epsilon}\right) \eta_t^2 \times \| \left[\nabla^2_{\mathbf{yy}} \psi_{\mu_t}(\mathbf{x}_{i,t}, \mathbf{y}_{i,t}) - \nabla^2_{\mathbf{yy}} \psi_{\mu_t}(\mathbf{x}_{i,t}, \mathbf{y}^*_{\mu_t}(\mathbf{x}_{i,t}))\right] \mathbf{v}^*_{\mu_t}(\mathbf{x}_{i,t})$$
$$\quad + \left(\nabla_{\mathbf{y}} f_i(\mathbf{x}_{i,t}, \mathbf{y}^*_{\mu_t}(\mathbf{x}_{i,t})) - \nabla_{\mathbf{y}} f_i(\mathbf{x}_{i,t}, \mathbf{y}^*_{i,t})\right) \|^2.$$

When $\eta_t \leq 1/L_{\psi_{\mu_t}}$, by the $\sigma_{\psi_\mu}$- strongly convexity of $\psi_{\mu_t}(\mathbf{x}_i, \cdot)$, since the spectral norm is monotone, we have

$$\left\| \left[I - \eta_t \nabla^2_{\mathbf{yy}} \psi_{\mu_t}(\mathbf{x}_{i,t}, \mathbf{y}_{i,t+1})\right] \left[\mathbf{v}_{i,t} - \mathbf{v}^*_{\mu_t}(\mathbf{x}_{i,t})\right] \right\| \leq \left(1 - \eta_t \sigma_{\psi_{\mu_t}}\right) \left\| \mathbf{v}_{i,t} - \mathbf{v}^*_{\mu_t}(\mathbf{x}_{i,t}) \right\|.$$

Next, by Lipschitz continuity of $\nabla^2_{\mathbf{yy}} \psi_{\mu_t}(\mathbf{x}, \cdot)$ and $\nabla_{\mathbf{y}} f_i(\mathbf{x}, \cdot)$, we get

$$\| \left[\nabla^2_{\mathbf{yy}} \psi_{\mu_t}(\mathbf{x}_{i,t}, \mathbf{y}_{i,t}) - \nabla^2_{\mathbf{yy}} \psi_{\mu_t}(\mathbf{x}_{i,t}, \mathbf{y}^*_{\mu_t}(\mathbf{x}_{i,t}))\right] \mathbf{v}^*_{\mu_t}(\mathbf{x}_{i,t}) + \nabla_{\mathbf{y}} f_i(\mathbf{x}_{i,t}, \mathbf{y}^*_{\mu_t}(\mathbf{x}_{i,t}))$$
$$\quad - \nabla_{\mathbf{y}} f_i(\mathbf{x}_{i,t}, \mathbf{y}_{i,t})\|$$
$$\leq \left(L_{\psi_{\mathbf{yy2}}} \left\| \mathbf{v}^*_{\mu_t}(\mathbf{x}_{i,t}) \right\| + L_{f_{i \mathbf{y2}}}\right) \left\| \mathbf{y}_{i,t} - \mathbf{y}^*_{\mu_t}(\mathbf{x}_{i,t}) \right\|.$$

Taking $\varepsilon = \eta_t \sigma_{\psi_{\mu_t}}$, we have

$$\left\| \mathbf{v}_{i,t+1} - \mathbf{v}^*_{\mu_t}(\mathbf{x}_{i,t}) \right\|^2$$
$$\leq \left(1 + \eta_t \sigma_{\psi_{\mu_t}}\right) \left(1 - \eta_t \sigma_{\psi_{\mu_t}}\right)^2 \left\| \mathbf{v}_{i,t} - \mathbf{v}^*_{\mu_t}(\mathbf{x}_{i,t}) \right\|^2$$
$$\quad + \left(1 + \frac{1}{\eta_t \sigma_{\psi_{\mu_t}}}\right) \eta_t^2 \left(L_{\psi_{\mathbf{yy2}}} \left\| \mathbf{v}^*_{\mu_t}(\mathbf{x}_{i,t}) \right\| + L_{f_{i \mathbf{y2}}}\right)^2 \left\| \mathbf{y}_{i,t} - \mathbf{y}^*_{\mu_t}(\mathbf{x}_{i,t}) \right\|^2,$$

which implies the desired result since $\eta_t^2 \leq \eta_t/L_{\psi_{\mu_t}} \leq \eta_t/\sigma_{\psi_{\mu_t}}$. □

This shows that the error in the multiplier $\mathbf{v}_{i,t}$ also contracts over iterations, provided that the step size $\eta_t$ is chosen appropriately.

By the above lemmas, we can now proceed to analyze the error of LL and multiplier variables.

**Lemma 5.** *Suppose Assumptions hold. For any $i \in [m]$, the the sequence of $\mathbf{x}_{i,t}, \mathbf{y}_{i,t}, \mathbf{v}_{i,t}$ generated by our algorithm satisfy*

$$\left\| \mathbf{y}^*_{\mu_t}(\mathbf{x}_{i,t}) - \mathbf{y}^*_{\mu_{t+1}}(\mathbf{x}_{i,t+1}) \right\|^2 \leq \frac{2L^2_{\psi_{\mathbf{y}1}}}{\sigma^2_{\psi_{\mu_t}}} \left\| \mathbf{x}_{i,t} - \mathbf{x}_{i,t+1} \right\|^2 + \frac{8 \left\| \mathbf{y}^*_{\mu_{t+1}}(\mathbf{x}_{i,t+1}) \right\|^2}{\sigma^2_{h_i}} \left( \frac{\mu_t - \mu_{t+1}}{\mu_t} \right)^2,$$

*and*

$$\left\| \mathbf{v}^*_{\mu_t}(\mathbf{x}_{i,t}) - \mathbf{v}^*_{\mu_{t+1}}(\mathbf{x}_{i,t+1}) \right\|^2 \leq \frac{2 \left( L_{\mathbf{v}1} \left\| \mathbf{v}^*_{\mu_t}(\mathbf{x}_{i,t}) \right\| + L_{\mathbf{v}2} \right)^2}{\sigma^4_{\psi_{\mu_t}}} \left\| \mathbf{x}_{i,t} - \mathbf{x}_{i,t+1} \right\|^2$$

$$+ 2 \left( \left( C_{\mathbf{v}1} \left\| \mathbf{v}^*_{\mu_{t+1}}(\mathbf{x}_{i,t+1}) \right\| + L_{f_i\mathbf{y}2} \right) \left\| \mathbf{y}^*_{\mu_{t+1}}(\mathbf{x}_{i,t+1}) \right\| \right.$$

$$\left. + C_{\mathbf{v}2} \left\| \nabla_{\mathbf{y}} f_i \left( \mathbf{x}_{i,t+1}, \mathbf{y}^*_{\mu_{t+1}}(\mathbf{x}_{i,t+1}) \right) \right\| \right)^2 \left( \frac{\mu_t - \mu_{t+1}}{\mu_t^2} \right)^2,$$

*where both $C_{\mathbf{v}1}$ and $C_{\mathbf{v}2}$ are constants given by*

$$C_{\mathbf{v}1} := 2 \left( L_{h_i\mathbf{yy}2} + L_{g_i\mathbf{yy}2} \right) / \sigma^2_{h_i}, \quad C_{\mathbf{v}2} := 2 \left( L_{f_i\mathbf{y}2} + L_{g_i\mathbf{y}2} \right) / \sigma^2_{h_i}.$$

*Proof.* By the triangle inequality, and Lemma 11(a) and 13(a) with $\mu' = \mu_t$ and $\mu = \mu_{t+1}$,

$$\left\| \mathbf{y}^*_{\mu_t}(\mathbf{x}_{i,t}) - \mathbf{y}^*_{\mu_{t+1}}(\mathbf{x}_{i,t+1}) \right\|$$

$$\leq \left\| \mathbf{y}^*_{\mu_t}(\mathbf{x}_{i,t}) - \mathbf{y}^*_{\mu_t}(\mathbf{x}_{i,t+1}) \right\| + \left\| \mathbf{y}^*_{\mu_t}(\mathbf{x}_{i,t+1}) - \mathbf{y}^*_{\mu_{t+1}}(\mathbf{x}_{i,t+1}) \right\|$$

$$\leq \frac{L_{\psi_{\mathbf{y}1}}}{\sigma_{\psi_{\mu_t}}} \left\| \mathbf{x}_{i,t} - \mathbf{x}_{i,t+1} \right\| + \frac{2 \left\| \mathbf{y}^*_{\mu_{t+1}}(\mathbf{x}_{i,t+1}) \right\|}{\sigma_{h_i}} \left( \frac{\mu_t - \mu_{t+1}}{\mu_t} \right).$$

Then with $\mu' = \mu_t$ and $\mu = \mu_{t+1}$, we have

$$\left\| \mathbf{v}^*_{\mu_t}(\mathbf{x}_{i,t}) - \mathbf{v}^*_{\mu_{t+1}}(\mathbf{x}_{i,t+1}) \right\| \leq \left\| \mathbf{v}^*_{\mu_t}(\mathbf{x}_{i,t}) - \mathbf{v}^*_{\mu_t}(\mathbf{x}_{i,t+1}) \right\| + \left\| \mathbf{v}^*_{\mu_t}(\mathbf{x}_{i,t+1}) - \mathbf{v}^*_{\mu_{t+1}}(\mathbf{x}_{i,t+1}) \right\|$$

$$\leq \frac{\left( L_{\mathbf{v}1} \left\| \mathbf{v}^*_{\mu_t}(\mathbf{x}_{i,t}) \right\| + L_{\mathbf{v}2} \right)}{\sigma^2_{\psi_{\mu_t}}} \left\| \mathbf{x}_{i,t} - \mathbf{x}_{i,t+1} \right\|$$

$$+ \left( \left( C_{\mathbf{v}1} \left\| \mathbf{v}^*_{\mu_{t+1}}(\mathbf{x}_{i,t+1}) \right\| + L_{f_i\mathbf{y}2} \right) \left\| \mathbf{y}^*_{\mu_{t+1}}(\mathbf{x}_{i,t+1}) \right\| \right.$$

$$\left. + C_{\mathbf{v}2} \left\| \nabla_{\mathbf{y}} f_i \left( \mathbf{x}_{i,t+1}, \mathbf{y}^*_{\mu_{t+1}}(\mathbf{x}_{i,t+1}) \right) \right\| \right) \left( \frac{\mu_t - \mu_{t+1}}{\mu_t^2} \right).$$

$\square$

Building upon the aforementioned lemmas, we can now proceed to analyze the iteration-wise differences in the errors of LL and multiplier variables.

**Lemma 6.** *Then for any $i \in [m]$, the sequence of $\mathbf{x}_{i,t}, \mathbf{y}_{i,t}, \mathbf{v}_{i,t}$ generated by our algorithm satisfy*

$$\left\| \mathbf{y}_{i,t+1} - \mathbf{y}^*_{\mu_{t+1}}(\mathbf{x}_{i,t+1}) \right\|^2 - \left\| \mathbf{y}_{i,t} - \mathbf{y}^*_{\mu_t}(\mathbf{x}_{i,t}) \right\|^2$$

$$\leq -\frac{1}{2} \beta_t \sigma_{\psi_{\mu_t}} \left\| \mathbf{y}_{i,t} - \mathbf{y}^*_{\mu_t}(\mathbf{x}_{i,t}) \right\|^2 - \left( 1 + \frac{1}{2} \beta_t \sigma_{\psi_{\mu_t}} \right) \left( 2\beta_t \frac{1}{\sigma_{\psi_{\mu_t}} + L_{\psi_{\mu_t}}} - \beta_t^2 \right) \left\| \mathbf{h}^{i,t}_{\mathbf{y}} \right\|^2$$

$$+ \frac{6L^2_{\psi_{\mathbf{y}1}}}{\beta_t \sigma^3_{\psi_{\mu_t}}} \left\| \mathbf{x}_{i,t} - \mathbf{x}_{i,t+1} \right\|^2 + \frac{24 \left\| \mathbf{y}^*_{\mu_{t+1}}(\mathbf{x}_{i,t+1}) \right\|^2}{\sigma^2_{h_i} \beta_t \sigma_{\psi_{\mu_t}}} \left( \frac{\mu_t - \mu_{t+1}}{\mu_t} \right)^2,$$

*and*

$$\left\| \mathbf{v}_{i,t+1} - \mathbf{v}^*_{\mu_{t+1}}\left(\mathbf{x}_{i,t+1}\right) \right\|^2 - \left\| \mathbf{v}_{i,t} - \mathbf{v}^*_{\mu_t}\left(\mathbf{x}_{i,t}\right) \right\|^2$$

$$\leq -\frac{1}{2}\eta_t \sigma_{\psi_{\mu_t}} \left\| \mathbf{v}_{i,t} - \mathbf{v}^*_{\mu_t}\left(\mathbf{x}_{i,t}\right) \right\|^2 + \frac{6\left(L_{\mathbf{v}1}\left\| \mathbf{v}^*_{\mu_t}\left(\mathbf{x}_{i,t}\right) \right\| + L_{\mathbf{v}2}\right)^2}{\eta_t \sigma^5_{\psi_{\mu_t}}} \left\| \mathbf{x}_{i,t} - \mathbf{x}_{i,t+1} \right\|^2$$

$$+ \frac{3\eta_t}{\sigma_{\psi_{\mu_t}}}\left(L_{\psi_{\mathbf{yy}2}}\left\| \mathbf{v}^*_{\mu_t}\left(\mathbf{x}_{i,t}\right) \right\| + L_{f_i\mathbf{y}2}\right)^2 \left\| \mathbf{y}_{i,t} - \mathbf{y}^*_{\mu_t}\left(\mathbf{x}_{i,t}\right) \right\|^2$$

$$+ \left(\frac{\mu_t - \mu_{t+1}}{\mu_t^2}\right)^2 \frac{6}{\eta_t \sigma_{\psi_{\mu_t}}}\left(\left(C_{\mathbf{v}1}\left\| \mathbf{v}^*_{\mu_{t+1}}\left(\mathbf{x}_{i,t+1}\right) \right\| + L_{f_i\mathbf{y}2}\right)\left\| \mathbf{y}^*_{\mu_{t+1}}\left(\mathbf{x}_{i,t+1}\right) \right\|\right.$$

$$\left. + C_{\mathbf{v}2}\left\| \nabla_{\mathbf{y}} f_i\left(\mathbf{x}_{i,t+1}, \mathbf{y}^*_{\mu_{t+1}}\left(\mathbf{x}_{i,t+1}\right)\right) \right\|\right)^2.$$

*Proof.* By Cauchy-Schwarz inequality, it is easy to check that for any $\epsilon > 0$, we have

$$\|\mathbf{y}_{i,t+1} - \mathbf{y}^*_{\mu_{t+1}}\left(\mathbf{x}_{i,t+1}\right)\|^2$$

$$= \|\mathbf{y}_{i,t+1} - \mathbf{y}^*_{\mu_t}\left(\mathbf{x}_{i,t}\right) + \mathbf{y}^*_{\mu_t}\left(\mathbf{x}_{i,t}\right) - \mathbf{y}^*_{\mu_{t+1}}\left(\mathbf{x}_{i,t+1}\right)\|^2$$

$$\leq (1+\epsilon)\|\mathbf{y}_{i,t+1} - \mathbf{y}^*_{\mu_t}\left(\mathbf{x}_{i,t}\right)\|^2 + (1+\frac{1}{\epsilon})\|\mathbf{y}^*_{\mu_t}\left(\mathbf{x}_{i,t}\right) - \mathbf{y}^*_{\mu_{t+1}}\left(\mathbf{x}_{i,t+1}\right)\|^2.$$

Taking $\epsilon = \frac{1}{2}\beta_t \sigma_{\psi_{\mu_t}}$, by Lemma 4 and Lemma 5, we have

$$\|\mathbf{y}_{i,t+1} - \mathbf{y}^*_{\mu_{t+1}}\left(\mathbf{x}_{i,t+1}\right)\|^2$$

$$\leq \left(1 - \frac{1}{2}\beta_t \sigma_{\psi_{\mu_t}}\right)\left\| \mathbf{y}_{i,t} - \mathbf{y}^*_{\mu_t}\left(\mathbf{x}_{i,t}\right) \right\|^2 - \left(1 + \frac{1}{2}\beta_t \sigma_{\psi_{\mu_t}}\right)\left(2\beta_t \frac{1}{\sigma_{\psi_{\mu_t}} + L_{\psi_{\mu_t}}} - \beta_t^2\right)\|\mathbf{h}_{\mathbf{y}}^{i,t}\|^2$$

$$+ \frac{2L^2_{\psi_{\mathbf{y}1}}}{\sigma^2_{\psi_{\mu_t}}}\left(1 + \frac{2}{\beta_t \sigma_{\psi_{\mu_t}}}\right)\|\mathbf{x}_{i,t} - \mathbf{x}_{i,t+1}\|^2$$

$$+ \frac{8\left\| \mathbf{y}^*_{\mu_{t+1}}\left(\mathbf{x}_{i,t+1}\right) \right\|^2}{\sigma^2_{h_i}}\left(1 + \frac{2}{\beta_t \sigma_{\psi_{\mu_t}}}\right)\left(\frac{\mu_t - \mu_{t+1}}{\mu_t}\right)^2,$$

which implies the desired result since $\beta_t \sigma_{\psi_{\mu_t}} \leq 1$ when $\beta_t \leq 2/(\sigma_{\psi_{\mu_t}} + L_{\psi_{\mu_t}})$.

Similarly, for any $\delta > 0$, by Cauchy-Schware inequality,

$$\|\mathbf{v}_{i,t+1} - \mathbf{v}^*_{\mu_{t+1}}\left(\mathbf{x}_{i,t+1}\right)\|^2$$

$$= \|\mathbf{v}_{i,t+1} - \mathbf{v}^*_{\mu_t}\left(\mathbf{x}_{i,t}\right) + \mathbf{v}^*_{\mu_t}\left(\mathbf{x}_{i,t}\right) - \mathbf{v}^*_{\mu_{t+1}}\left(\mathbf{x}_{i,t+1}\right)\|^2$$

$$\leq (1+\delta)\|\mathbf{v}_{i,t+1} - \mathbf{v}^*_{\mu_t}\left(\mathbf{x}_{i,t}\right)\|^2 + (1+\frac{1}{\delta})\|\mathbf{v}^*_{\mu_t}\left(\mathbf{x}_{i,t}\right) - \mathbf{v}^*_{\mu_{t+1}}\left(\mathbf{x}_{i,t+1}\right)\|^2.$$

Taking $\delta = \frac{1}{2}\eta_t \sigma_{\psi_{\mu_t}}$, by Lemma 4 and Lemma 5, we have

$$\left\| \mathbf{v}_{i,t+1} - \mathbf{v}^*_{\mu_{t+1}}\left(\mathbf{x}_{i,t+1}\right) \right\|^2$$

$$\leq \left(1 - \frac{1}{2}\eta_t \sigma_{\psi_{\mu_t}}\right)\left\| \mathbf{v}_{i,t} - \mathbf{v}^*_{\mu_t}\left(\mathbf{x}_{i,t}\right) \right\|^2$$

$$+ \frac{2\left(L_{\mathbf{v}1}\left\| \mathbf{v}^*_{\mu_t}\left(\mathbf{x}_{i,t}\right) \right\| + L_{\mathbf{v}2}\right)^2}{\sigma^4_{\psi_{\mu_t}}}\left(1 + \frac{2}{\eta_t \sigma_{\psi_{\mu_t}}}\right)\|\mathbf{x}_{i,t} - \mathbf{x}_{i,t+1}\|^2$$

$$+ \frac{2\eta_t}{\sigma_{\psi_{\mu_t}}}\left(1 + \frac{1}{2}\eta_t \sigma_{\psi_{\mu_t}}\right)\left(L_{\psi_{\mathbf{yy}2}}\left\| \mathbf{v}^*_{\mu_t}\left(\mathbf{x}_{i,t}\right) \right\| + L_{f_i\mathbf{y}2}\right)^2 \left\| \mathbf{y}_{i,t} - \mathbf{y}^*_{\mu_t}\left(\mathbf{x}_{i,t}\right) \right\|^2$$

$$+ 2\left(\left(C_{\mathbf{v}1}\left\| \mathbf{v}^*_{\mu_{t+1}}\left(\mathbf{x}_{i,t+1}\right) \right\| + L_{f_i\mathbf{y}2}\right)\left\| \mathbf{y}^*_{\mu_{t+1}}\left(\mathbf{x}_{i,t+1}\right) \right\|\right.$$

$$+ C_{\mathbf{v}2} \left\| \nabla_{\mathbf{y}} f_i \left( \mathbf{x}_{i,t+1}, \mathbf{y}^*_{\mu_{t+1}} \left( \mathbf{x}_{i,t+1} \right) \right) \right\| \right)^2 \times \left( 1 + \frac{2}{\eta_t \sigma_{\psi_{\mu_t}}} \right) \left( \frac{\mu_t - \mu_{t+1}}{\mu_t^2} \right)^2.$$

This desired result follows since $\eta_t \sigma_{\psi_{\mu_t}} \leq 1$ when $\eta_t \leq 1/L_{\psi_{\mu_t}}$. $\qquad\square$

**Step 3: Controlling both the consensus steps and the gradient tracking steps.**
In this step, we are focusing on controlling both the consensus steps and the gradient tracking steps within our decentralized algorithm. The goal is to show that the iterates contract over time, which means that both the decision variables $\mathbf{x}_t$ and the gradient tracking variables $\mathbf{h}_{\mathbf{x}}^t$ converge as the algorithm progresses. This lemma proves that the discrepancy between these variables across agents reduces over iterations, thus guaranteeing convergence.

**Lemma 7** (Iterates Contraction). *The following contraction properties of the iterates hold:*

$$\|\mathbf{x}_{t+1} - \mathbf{1} \otimes \bar{\mathbf{x}}_{t+1}\|^2 \leq (1 + l_1)\lambda^2 \|\mathbf{x}_t - \mathbf{1} \otimes \bar{\mathbf{x}}_t\|^2 + (1 + \frac{1}{l_1})\alpha_t^2 \|\mathbf{h}_{\mathbf{x}}^t - \mathbf{1} \otimes \bar{\mathbf{h}}_{\mathbf{x}}^t\|^2,$$

*and*

$$\|\mathbf{h}_{\mathbf{x}}^{t+1} - \mathbf{1} \otimes \bar{\mathbf{h}}_{\mathbf{x}}^{t+1}\|^2 \leq (1 + l_2)\lambda^2 \|\mathbf{h}_{\mathbf{x}}^t - \mathbf{1} \otimes \bar{\mathbf{h}}_{\mathbf{x}}^t\|^2 + (1 + \frac{1}{l_2})\|\mathbf{d}_{\mathbf{x}}^{t+1} - \mathbf{d}_{\mathbf{x}}^t\|^2,$$

*where $l_1$ and $l_2$ are arbitrary positive constants.*

*Additionally, we have*

$$\|\mathbf{x}_{t+1} - \mathbf{x}_t\|^2 \leq 8\|(\mathbf{x}_t - \mathbf{1} \otimes \bar{\mathbf{x}}_t)\|^2 + 4\alpha_t^2 \|\mathbf{h}_{\mathbf{x}}^t - \mathbf{1} \otimes \bar{\mathbf{h}}_{\mathbf{x}}^t\|^2 + 4\alpha_t^2 m \|\bar{\mathbf{h}}_{\mathbf{x}}^t\|^2.$$

*And also,*

$$\|\mathbf{y}_{t+1} - \mathbf{y}_t\|^2 \leq \beta_t^2 \|\mathbf{h}_{\mathbf{y}}^t\|^2.$$

The first inequality states that the deviation of the iterates $\mathbf{x}_{t+1}$ from their consensus value $\mathbf{1} \otimes \bar{\mathbf{x}}_{t+1}$ (where $\mathbf{1}\otimes$ represents the averaging operator) is bounded by a factor that depends on the previous deviation, scaled by the contraction factor $\lambda$, and the gradient tracking error. The second inequality bounds the change in the gradient tracking variables $\mathbf{h}_{\mathbf{x}}^t$ between iterations $t$ and $t+1$, showing that it contracts over time, depending on the contraction factor $\lambda$ and the difference in the gradient correction steps $\mathbf{d}_{\mathbf{x}}^{t+1} - \mathbf{d}_{\mathbf{x}}^t$. These two contraction results help ensure that the iterates (both the decision variables and the gradient tracking variables) converge towards their consensus values over time.

*Proof.* We begin by analyzing the iterates of $\mathbf{x}_{t+1}$. The idea is to show that the deviation of $\mathbf{x}_{t+1}$ from its average consensus value $\mathbf{1} \otimes \bar{\mathbf{x}}_{t+1}$ contracts over iterations. Define $\widetilde{\mathbf{M}} = \mathbf{M} \otimes \mathbf{I}_m$. First for the iterates $\mathbf{x}_t$, we have the following contraction:

$$\|\widetilde{\mathbf{M}}\mathbf{x}_t - \mathbf{1} \otimes \bar{\mathbf{x}}_t\|^2 = \|\widetilde{\mathbf{M}}(\mathbf{x}_t - \mathbf{1} \otimes \bar{\mathbf{x}}_t)\|^2 \leq \lambda^2 \|\mathbf{x}_t - \mathbf{1} \otimes \bar{\mathbf{x}}_t\|^2. \tag{15}$$

This is because $\mathbf{x}_t - \mathbf{1} \otimes \mathbf{x}_t$ is orthogonal $\mathbf{1}$, which is the eigenvector corresponding to the largest eigenvalue of $\widetilde{\mathbf{M}}$, and $\lambda = \max\{|\lambda_2|, |\lambda_m|\}$. Recall that $\bar{\mathbf{x}}_t = \bar{\mathbf{x}}_{t-1} - \alpha_{t-1}\bar{\mathbf{h}}_{\mathbf{x}}^{t-1}$, hence,

$$\|\mathbf{x}_t - \mathbf{1} \otimes \bar{\mathbf{x}}_t\|^2 = \|\widetilde{\mathbf{M}}\mathbf{x}_{t-1} - \alpha_{t-1}\mathbf{h}_{\mathbf{x}}^{t-1} - \mathbf{1} \otimes (\bar{\mathbf{x}}_{t-1} - \alpha_{t-1}\bar{\mathbf{h}}_{\mathbf{x}}^{t-1})\|^2$$

$$\overset{(a)}{\leq} (1 + l_1)\|\widetilde{\mathbf{M}}\mathbf{x}_{t-1} - \mathbf{1} \otimes \bar{\mathbf{x}}_{t-1}\|^2 + (1 + \frac{1}{l_1})\alpha_{t-1}^2 \|\mathbf{h}_{\mathbf{x}}^{t-1} - \mathbf{1} \otimes \bar{\mathbf{h}}_{\mathbf{x}}^{t-1}\|^2$$

$$\overset{(b)}{\leq} (1 + l_1)\lambda^2 \|\mathbf{x}_{t-1} - \mathbf{1} \otimes \bar{\mathbf{x}}_{t-1}\|^2 + (1 + \frac{1}{l_1})\alpha_{t-1}^2 \|\mathbf{h}_{\mathbf{x}}^{t-1} - \mathbf{1} \otimes \bar{\mathbf{h}}_{\mathbf{x}}^{t-1}\|^2,$$

where (a) is because of triangle inequality and (b) is from (15). Next, we analyze the gradient tracking steps. The goal here is to show that the deviation of the gradient tracking variables from their consensus value also contracts over time. Using the update rule for $\mathbf{h}_{\mathbf{x}}^t$, we have

$$\|\mathbf{h}_{\mathbf{x}}^t - \mathbf{1} \otimes \bar{\mathbf{h}}_{\mathbf{x}}^t\|^2$$

$$=\|\widetilde{\mathbf{M}}\mathbf{h}_{\mathbf{x}}^{t-1} + \mathbf{d}_{\mathbf{x}}^t - \mathbf{d}_{\mathbf{x}}^{t-1} - \mathbf{1} \otimes (\bar{\mathbf{h}}_{\mathbf{x}}^{t-1} + \bar{\mathbf{d}}_{\mathbf{x}}^t - \bar{\mathbf{d}}_{\mathbf{x}}^{t-1})\|^2$$

$$\leq (1+l_2)\lambda^2 \|\mathbf{h}_{\mathbf{x}}^{t-1} - \mathbf{1} \otimes \bar{\mathbf{h}}_{\mathbf{x}}^{t-1}\|^2 + (1+\frac{1}{l_2})\|\mathbf{d}_{\mathbf{x}}^t - \mathbf{d}_{\mathbf{x}}^{t-1} - \mathbf{1} \otimes (\bar{\mathbf{d}}_{\mathbf{x}}^t - \bar{\mathbf{d}}_{\mathbf{x}}^{t-1}))\|^2$$

$$\leq (1+l_2)\lambda^2 \|\mathbf{h}_{\mathbf{x}}^{t-1} - \mathbf{1} \otimes \bar{\mathbf{h}}_{\mathbf{x}}^{t-1}\|^2 + (1+\frac{1}{l_2})\|(\mathbf{I} - \frac{1}{n}(\mathbf{1}\mathbf{1}^\top) \otimes \mathbf{I})(\mathbf{d}_{\mathbf{x}}^t - \mathbf{d}_{\mathbf{x}}^{t-1})\|^2$$

$$\overset{(a)}{\leq} (1+l_2)\lambda^2 \|\mathbf{h}_{\mathbf{x}}^{t-1} - \mathbf{1} \otimes \bar{\mathbf{h}}_{\mathbf{x}}^{t-1}\|^2 + (1+\frac{1}{l_2})\|\mathbf{d}_{\mathbf{x}}^t - \mathbf{d}_{\mathbf{x}}^{t-1}\|^2, \tag{16}$$

where (a) is due to $\|\mathbf{I} - \frac{1}{m}(\mathbf{1}\mathbf{1}^\top) \otimes \mathbf{I}\| \leq 1$.

From (16), letting $l_2 = \frac{1-\lambda}{\lambda}$, we can further have

$$(1+l_2)\lambda^2 \|\mathbf{h}_{\mathbf{x}}^{t-1} - \mathbf{1} \otimes \bar{\mathbf{h}}_{\mathbf{x}}^{t-1}\|^2 + (1+\frac{1}{l_2})\|\mathbf{d}_{\mathbf{x}}^t - \mathbf{d}_{\mathbf{x}}^{t-1}\|^2$$

$$\leq \lambda \|\mathbf{h}_{\mathbf{x}}^{t-1} - \mathbf{1} \otimes \bar{\mathbf{h}}_{\mathbf{x}}^{t-1}\|^2 + \frac{1}{1-\lambda}\|\mathbf{d}_{\mathbf{x}}^t - \mathbf{d}_{\mathbf{x}}^{t-1}\|^2.$$

Then we have

$$\|\mathbf{h}_{\mathbf{x}}^t - \mathbf{1} \otimes \bar{\mathbf{h}}_{\mathbf{x}}^t\|^2 \leq \frac{1}{1-\lambda} \sum_{s=0}^{t+1} \left(\lambda^{t+1-s}\|\mathbf{d}_{\mathbf{x}}^s - \mathbf{d}_{\mathbf{x}}^{s-1}\|^2\right). \tag{17}$$

By using $\|\mathbf{h}_{\mathbf{x}}^t\|^2 = \|\mathbf{h}_{\mathbf{x}}^t - \mathbf{1} \otimes \bar{\mathbf{h}}_{\mathbf{x}}^t\|^2 + n\|\bar{\mathbf{h}}_{\mathbf{x}}^t\|^2$ and (17), we can immediately deduce that

$$\|\mathbf{h}_{\mathbf{x}}^t\|^2 \leq \frac{1}{1-\lambda} \sum_{s=0}^{t+1} \left(\lambda^{t+1-s}\|\mathbf{d}_{\mathbf{x}}^s - \mathbf{d}_{\mathbf{x}}^{s-1}\|^2\right) + n\|\bar{\mathbf{d}}_{\mathbf{x}}^t\|^2.$$

According to the updating mechanism detailed in (5), we have

$$\|\mathbf{x}_t - \mathbf{x}_{t-1}\|^2$$

$$=\|\widetilde{\mathbf{M}}\mathbf{x}_{t-1} - \alpha_{t-1}\mathbf{h}_{\mathbf{x}}^{t-1} - \mathbf{x}_{t-1}\|^2$$

$$=\|(\widetilde{\mathbf{M}} - \mathbf{I})\mathbf{x}_{t-1} - \alpha_{t-1}\mathbf{h}_{\mathbf{x}}^{t-1}\|^2$$

$$\leq 2\|(\widetilde{\mathbf{M}} - \mathbf{I})\mathbf{x}_{t-1}\|^2 + 2\alpha_{t-1}^2\|\mathbf{h}_{\mathbf{x}}^{t-1}\|^2$$

$$=2\|(\widetilde{\mathbf{M}} - \mathbf{I})(\mathbf{x}_{t-1} - \mathbf{1} \otimes \bar{\mathbf{x}}_{t-1})\|^2 + 2\alpha_{t-1}^2\|\mathbf{h}_{\mathbf{x}}^{t-1}\|^2$$

$$\leq 8\|(\mathbf{x}_{t-1} - \mathbf{1} \otimes \bar{\mathbf{x}}_{t-1})\|^2 + 4\alpha_{t-1}^2\|\mathbf{h}_{\mathbf{x}}^{t-1} - \mathbf{1} \otimes \bar{\mathbf{h}}_{\mathbf{x}}^{t-1}\|^2 + 4\alpha_{t-1}^2 m\|\bar{\mathbf{h}}_{\mathbf{x}}^{t-1}\|^2.$$

Also,

$$\|\mathbf{y}_t - \mathbf{y}_{t-1}\|^2 = \|Pr_y^t[\mathbf{y}_{t-1} - \beta_t^2 \mathbf{h}_{t-1}] - \mathbf{y}_{t-1}\|^2 \leq \|\mathbf{y}_{t-1} - \beta_t^2 \mathbf{h}_{t-1} - \mathbf{y}_{t-1}\|^2 \leq \beta_t^2 \|\mathbf{h}_{\mathbf{y}}^{t-1}\|^2,$$

where the first inequality holds because $\|\mathbf{y}_{t-1}\|^2 \leq r_y^{t-1} \leq r_y^t$ by the algorithm setting, implying the non-expensive property. $\qquad\square$

**Lemma 8.** *Let $r_x$ be defined as in* (9). *Then, the following inequality holds:*

$$\|\mathbf{x}_{i,t}\|^2 \leq r_x^2 = O\left(\frac{1}{\sigma_{\psi_{\mu_t}}^2}\right).$$

This lemma states that the squared norm of the vector $\mathbf{x}_{i,t}$ is bounded by $r_x^2$, which is of the order $O\left(\frac{1}{\sigma_{\psi_{\mu_t}}^2}\right)$.

*Proof.* By Assumption 2, we have

$$\left\|\mathbf{d}_{\mathbf{x}}^{i,t}\right\| = \|\nabla_{\mathbf{x}} f_i(\mathbf{x}_{i,t}, \mathbf{y}_{i,t}) - \nabla_{\mathbf{x}\mathbf{y}}\psi_i(\mathbf{x}_{i,t}, \mathbf{y}_{i,t})\mathbf{v}_{i,t}\|$$

$$\leq \|\nabla_{\mathbf{x}} f_i(\mathbf{x}_{i,t}, \mathbf{y}_{i,t})\| + \|\nabla_{\mathbf{x}\mathbf{y}}\psi_i(\mathbf{x}_{i,t}, \mathbf{y}_{i,t})\| \|\mathbf{v}_{i,t}\|$$

$$\leq L_{f_{i_0}} + (L_{\psi_{i\mathbf{y}1}} r_v) = B_1 = O(\frac{1}{\sigma_{\psi_{\mu_t}}}),$$

and

$$\left\|\bar{\mathbf{d}}_{\mathbf{x}}^{i,t}\right\| \leq \frac{1}{m} \sum_{i=1}^{m} \left\|\mathbf{d}_{\mathbf{x}}^{i,t}\right\| \leq B_1 = O(\frac{1}{\sigma_{\psi_{\mu_t}}}).$$

Here we also prove that

$$\left\|\mathbf{d}_{\mathbf{y}}^{i,t}\right\| = \|\nabla_{\mathbf{y}}\psi_i(\mathbf{x}_{i,t}, \mathbf{y}_{i,t})\| \leq \|\nabla_{\mathbf{y}}\psi_i(\mathbf{x}_{i,t}, \mathbf{y}_{i,t}) - \nabla_{\mathbf{y}}\psi_i(0,0)\| + \|\nabla_{\mathbf{y}}\psi_i(0,0)\|$$

$$\leq L_{\psi_{\mathbf{y}}}(\|\mathbf{x}_{i,t}\| + \|\mathbf{y}_{i,t}\|) \leq L_{\psi_{\mathbf{y}}}(r_x^t + r_y^t) + \|\nabla_{\mathbf{y}}\psi_i(0,0)\| \leq B_2 = \tilde{O}(\frac{1}{\sigma_{\psi_{\mu_t}}^2}),$$

and

$$\left\|\mathbf{d}_{\mathbf{v}}^{i,t}\right\| = \|\nabla_{\mathbf{y}}f_i(\mathbf{x}_{i,t}, \mathbf{y}_{i,t}) - \nabla_{\mathbf{y}\mathbf{y}}\psi_i(\mathbf{x}_{i,t}, \mathbf{y}_{i,t})\mathbf{v}_{i,t}\| \leq L_{f_{i_0}} + (L_{\psi_{i\mathbf{y}2}} r_v) = B_3 = O(\frac{1}{\sigma_{\psi_{\mu_t}}}).$$

We can deduce that

$$\sum_{i=1}^{m} \left\|\mathbf{h}_{\mathbf{x}}^{i,t+1}\right\|^2 \leq \frac{4mB_1^2}{1-\lambda}\left(\sum_{s=0}^{t+1}\lambda^{t+1-s}\right) + mB_1^2 \leq \left(1 + \frac{4}{(1-\lambda)^2}\right)B_1^2 m.$$

From $\mathbf{x}_{i,t+1} = \sum_{i'\in\mathcal{N}_i}[\mathbf{M}]_{ii'}\mathbf{x}_{i',t} - \alpha_t\mathbf{h}_{\mathbf{x}}^{i,t}$, we have

$$\sum_{i=1}^{m} \|\mathbf{x}_{i,t+1} - \mathbf{x}_{i,t}\|^2$$

$$\leq \sum_{i=1}^{m} \left\|\sum_{i'\in\mathcal{N}_i}[\mathbf{M}]_{ii'}\mathbf{x}_{i',t} - \alpha_t\mathbf{h}_{\mathbf{x}}^{i,t} - \sum_{i'\in\mathcal{N}_i}[\mathbf{M}]_{ii'}\mathbf{x}_{i',t-1} - \alpha_{t-1}\mathbf{h}_{\mathbf{x}}^{i,t-1}\right\|^2$$

$$\leq \frac{1}{\lambda}\sum_{i=1}^{m} \left\|\sum_{i'\in\mathcal{N}_i}[\mathbf{M}]_{ii'}\mathbf{x}_{i',t} - \sum_{i'\in\mathcal{N}_i}[\mathbf{M}]_{ii'}\mathbf{x}_{i',t-1}\right\|^2 + \frac{1}{1-\lambda}\sum_{i=1}^{m}\left\|\alpha_t\mathbf{h}_{\mathbf{x}}^{i,t} - \alpha_{t-1}\mathbf{h}_{\mathbf{x}}^{i,t-1}\right\|^2$$

$$\leq \frac{1}{\lambda}\sum_{i=1}^{m} \left\|\sum_{i'\in\mathcal{N}_i}[\mathbf{M}]_{ii'}(\mathbf{x}_{i',t} - \mathbf{x}_{i',t-1})\right\|^2 + \frac{1}{1-\lambda}\alpha_{t-1}^2\sum_{i=1}^{m}\left\|\mathbf{h}_{\mathbf{x}}^{i,t} - \mathbf{h}_{\mathbf{x}}^{i,t-1}\right\|^2,$$

where the second inequality is due to the Cauchy-Schwarz inequality. Now we have

$$\sum_{i=1}^{m} \left\|\sum_{i'\in\mathcal{N}_i}[\mathbf{M}]_{ii'}\mathbf{x}_{i',t} - \sum_{i'\in\mathcal{N}_i}[\mathbf{M}]_{ii'}\mathbf{x}_{i',t-1} - (\bar{\mathbf{x}}_t - \bar{\mathbf{x}}_{t-1})\right\|^2$$

$$\leq \lambda^2 \sum_{i=1}^{m} \|\mathbf{x}_{i,t} - \mathbf{x}_{i,t-1} - (\bar{\mathbf{x}}_t - \bar{\mathbf{x}}_{t-1})\|^2.$$

Moreover, we have

$$\sum_{i=1}^{m} \left\|\sum_{i'\in\mathcal{N}_i}[\mathbf{M}]_{ii'}\mathbf{x}_{i',t} - \sum_{i'\in\mathcal{N}_i}[\mathbf{M}]_{ii'}\mathbf{x}_{i',t-1} - (\bar{\mathbf{x}}_t - \bar{\mathbf{x}}_{t-1})\right\|^2$$

$$= \sum_{i=1}^{m} \left\|\sum_{i'\in\mathcal{N}_i}[\mathbf{M}]_{ii'}\mathbf{x}_{i',t} - \sum_{i'\in\mathcal{N}_i}[\mathbf{M}]_{ii'}\mathbf{x}_{i',t-1}\right\|^2 + m\|\bar{\mathbf{x}}_t - \bar{\mathbf{x}}_{t-1}\|^2$$

$$- 2\sum_{i=1}^{m}\left\langle\sum_{i'\in\mathcal{N}_i}[\mathbf{M}]_{ii'}\mathbf{x}_{i',t} - \sum_{i'\in\mathcal{N}_i}[\mathbf{M}]_{ii'}\mathbf{x}_{i',t-1}, \bar{\mathbf{x}}_t - \bar{\mathbf{x}}_{t-1}\right\rangle$$

$$= \sum_{i=1}^{m} \left\|\sum_{i'\in\mathcal{N}_i}[\mathbf{M}]_{ii'}\mathbf{x}_{i',t} - \sum_{i'\in\mathcal{N}_i}[\mathbf{M}]_{ii'}\mathbf{x}_{i',t-1}\right\|^2 - m\|\bar{\mathbf{x}}_t - \bar{\mathbf{x}}_{t-1}\|^2,$$

and

$$\sum_{i=1}^{m} \|\mathbf{x}_{i,t} - \mathbf{x}_{i,t-1} - (\bar{\mathbf{x}}_t - \bar{\mathbf{x}}_{t-1})\|^2$$

$$= \sum_{i=1}^{m} \|\mathbf{x}_{i,t} - \mathbf{x}_{i,t-1}\|^2 + m \|\bar{\mathbf{x}}_t - \bar{\mathbf{x}}_{t-1}\|^2 - 2 \sum_{i=1}^{m} \langle \mathbf{x}_{i,t} - \mathbf{x}_{i,t-1}, \bar{\mathbf{x}}_t - \bar{\mathbf{x}}_{t-1} \rangle$$

$$= \sum_{i=1}^{m} \|\mathbf{x}_{i,t} - \mathbf{x}_{i,t-1}\|^2 - m \|\bar{\mathbf{x}}_t - \bar{\mathbf{x}}_{t-1}\|^2 .$$

Then we obtain

$$\sum_{i=1}^{m} \left\| \sum_{i' \in \mathcal{N}_i} [\mathbf{M}]_{ii'} \mathbf{x}_{i',t} - \sum_{i' \in \mathcal{N}_i} [\mathbf{M}]_{ii'} \mathbf{x}_{i',t-1} \right\|^2$$

$$\leq \lambda^2 \sum_{i=1}^{m} \|\mathbf{x}_{i,t} - \mathbf{x}_{i,t-1}\|^2 + (1 - \lambda^2) m \|\bar{\mathbf{x}}_t - \bar{\mathbf{x}}_{t-1}\|^2$$

$$= \lambda^2 \sum_{i=1}^{m} \|\mathbf{x}_{i,t} - \mathbf{x}_{i,t-1}\|^2 + (1 - \lambda^2) m \alpha_{t-1}^2 \|\bar{\mathbf{d}}_{\mathbf{x}}^{t-1}\|^2 .$$

Now we have that

$$\sum_{i=1}^{m} \|\mathbf{x}_{i,t+1} - \mathbf{x}_{i,t}\|^2$$

$$\leq \lambda \sum_{i=1}^{m} \|\mathbf{x}_{i,t} - \mathbf{x}_{i,t-1}\|^2 + \frac{1 - \lambda^2}{\lambda} m \alpha_{t-1}^2 \|\bar{\mathbf{d}}_{\mathbf{x}}^{t-1}\|^2 + \frac{1}{1 - \lambda} \alpha_{t-1}^2 \sum_{i=1}^{m} \|\mathbf{h}_{\mathbf{x}}^{i,t} - \mathbf{h}_{\mathbf{x}}^{i,t-1}\|^2$$

$$\leq \lambda \sum_{i=1}^{m} \|\mathbf{x}_{i,t} - \mathbf{x}_{i,t-1}\|^2 + \frac{1 - \lambda^2}{\lambda} m \alpha_{t-1}^2 \|\bar{\mathbf{d}}_{\mathbf{x}}^{t-1}\|^2$$

$$+ \frac{2}{1 - \lambda} \alpha_{t-1}^2 \left( \sum_{i=1}^{m} \|\mathbf{h}_{\mathbf{x}}^{i,t}\|^2 + \sum_{i=1}^{m} \|\mathbf{h}_{\mathbf{x}}^{i,t-1}\|^2 \right) .$$

We set $\lambda_1 = \frac{1 - \lambda^2}{\lambda} + \frac{4}{1 - \lambda} + \frac{16}{(1 - \lambda)^3}$ and have

$$\sum_{i=1}^{m} \|\mathbf{x}_{i,t+1} - \mathbf{x}_{i,t}\|^2 \leq \lambda \sum_{i=1}^{m} \|\mathbf{x}_{i,t} - \mathbf{x}_{i,t-1}\|^2 + \lambda_1 B_1^2 m \alpha_{t-1}^2$$

$$\leq \lambda^t \sum_{i=1}^{m} \|\mathbf{x}_{i,0} - \mathbf{x}_{i,-1}\|^2 + \lambda_1 B_1^2 m \sum_{s=1}^{t} \alpha_{s-1}^2$$

$$\leq \lambda_1 B_1^2 m \sum_{s=1}^{t} \alpha_{s-1}^2$$

$$\leq \lambda_1 B_1^2 m \bar{\beta} \bar{\mu} \frac{1}{(t+1)^{2 \left( 1 + \frac{7}{4} \tau + p \right)}},$$

where $\alpha_t = (t+1)^{-3\tau/2} \beta_t \mu_t^3$, and $\sum_{s=1}^{t} \alpha_{s-1}^2 \leq \bar{\beta} \bar{\mu} \frac{1}{(t+1)^{2 \left( 1 + \frac{7}{4} \tau + p \right)}} \log(t+1)$. Then we have

$$\sum_{i=1}^{m} \|\mathbf{x}_{i,t}\|^2 \leq 2 \sum_{i=1}^{m} \|\mathbf{x}_{i,t} - \mathbf{x}_{i,0}\|^2 + 2 \sum_{i=1}^{m} \|\mathbf{x}_{i,0}\|^2$$

$$\leq 2t \sum_{s=0}^{t-1} \sum_{i=1}^{m} \|\mathbf{x}_{i,s+1} - \mathbf{x}_{i,s}\|^2 + 2 \sum_{i=1}^{m} \|\mathbf{x}_{i,0}\|^2$$

$$\leq 2\lambda_1 B_1^2 m\bar{\beta}\bar{\mu}\frac{t^2}{(t+1)^{2\left(1+\frac{7}{4}\tau+p\right)}} + 2\sum_{i=1}^{m}\|\mathbf{x}_i^0\|^2$$

$$\leq 2\lambda_1 B_1^2 m\bar{\beta}\bar{\mu} + 2\sum_{i=1}^{m}\|\mathbf{x}_i^0\|^2$$

which implies $\|\mathbf{x}_{i,t}\|^2 \leq \sum_{i=1}^{m}\|\mathbf{x}_{i,t}\|^2 \leq r_x^2 = O(\frac{1}{\sigma_{\psi_{\mu_t}}^2})$, where $r_x$ is defined in (9). $\qquad\square$

We now analyze the contraction properties of the iterates $\mathbf{x}_{t+1}$ and $\mathbf{h}_{\mathbf{x}}^{t+1}$, respectively, as generated by the algorithm. Specifically, we aim to track how the differences between successive iterations decrease over time.

**Lemma 9.**

$$\|\mathbf{x}_{t+1} - \mathbf{1}\otimes\bar{\mathbf{x}}_{t+1}\|^2 - \|\mathbf{x}_t - \mathbf{1}\otimes\bar{\mathbf{x}}_t\|^2$$
$$\leq ((1+l_1)\lambda^2 - 1)\|\mathbf{x}_t - \mathbf{1}\otimes\bar{\mathbf{x}}_t\|^2 + (1+\frac{1}{l_1})\alpha_t^2\|\mathbf{h}_{\mathbf{x}}^t - \mathbf{1}\otimes\bar{\mathbf{h}}_{\mathbf{x}}^t\|^2,$$

*where $l_1$ is an arbitrary positive constant.*

$$\|\mathbf{h}_{\mathbf{x}}^{t+1} - \mathbf{1}\otimes\bar{\mathbf{h}}_{\mathbf{x}}^{t+1}\|^2 - \|\mathbf{h}_{\mathbf{x}}^t - \mathbf{1}\otimes\bar{\mathbf{h}}_{\mathbf{x}}^t\|^2$$
$$\leq ((1+l_2)\lambda^2 - 1)\|\mathbf{h}_{\mathbf{x}}^t - \mathbf{1}\otimes\bar{\mathbf{h}}_{\mathbf{x}}^t\|^2 + (1+\frac{1}{l_2})(L_{f_{i_1}}^2\|\mathbf{x}_{t+1} - \mathbf{x}_t\|^2 + L_{f_{i_2}}^2\beta_t^2\|\mathbf{h}_{\mathbf{y}}^t\|^2),$$

*where $l_2$ is also an arbitrary positive constant.*

*Proof.* We start by examining the difference in the updates between iterations for both $\mathbf{x}_t$ and $\mathbf{h}_{\mathbf{x}}^t$.

Using the smoothness assumptions (Assumption 2), we first bound the difference between successive updates of $\mathbf{d}_{\mathbf{x}}$, which are defined as the gradient of the objective function:

$$\|\mathbf{d}_{\mathbf{x}}^{t+1} - \mathbf{d}_{\mathbf{x}}^t\|^2 \leq L_{f_{i_1}}^2\|\mathbf{x}_{t+1} - \mathbf{x}_t\|^2 + L_{f_{i_2}}^2\|\mathbf{y}_{t+1} - \mathbf{y}_t\|^2 \leq L_{f_{i_1}}^2\|\mathbf{x}_{t+1} - \mathbf{x}_t\|^2 + L_{f_{i_2}}^2\beta_t^2\|\mathbf{h}_{\mathbf{y}}^t\|^2.$$

Applying the results from Lemma 7, we have:

$$\|\mathbf{x}_{t+1} - \mathbf{1}\otimes\bar{\mathbf{x}}_{t+1}\|^2$$
$$\leq (1+l_1)\lambda^2\|\mathbf{x}_t - \mathbf{1}\otimes\bar{\mathbf{x}}_t\|^2 + (1+\frac{1}{l_1})\alpha_t^2\|\mathbf{h}_{\mathbf{x}}^t - \mathbf{1}\otimes\bar{\mathbf{h}}_{\mathbf{x}}^t\|^2,$$

which reflects the contraction behavior of the consensus steps for the decision variable $\mathbf{x}_t$. Similarly, the contraction property for $\mathbf{h}_{\mathbf{x}}^t$, the gradient tracking variable, follows:

$$\|\mathbf{h}_{\mathbf{x}}^{t+1} - \mathbf{1}\otimes\bar{\mathbf{h}}_{\mathbf{x}}^{t+1}\|^2$$
$$\leq (1+l_2)\lambda^2\|\mathbf{h}_{\mathbf{x}}^t - \mathbf{1}\otimes\bar{\mathbf{h}}_{\mathbf{x}}^t\|^2 + (1+\frac{1}{l_2})(L_{f_{i_1}}^2\|\mathbf{x}_i - \mathbf{x}_i'\|^2 + L_{f_{i_2}}^2\beta_t^2\|\mathbf{h}_{\mathbf{y}}^t\|^2).$$

Finally, we combine the two bounds to obtain the iterative contraction properties for both $\mathbf{x}_{t+1}$ and $\mathbf{h}_{\mathbf{x}}^{t+1}$:

$$\|\mathbf{x}_{t+1} - \mathbf{1}\otimes\bar{\mathbf{x}}_{t+1}\|^2 - \|\mathbf{x}_t - \mathbf{1}\otimes\bar{\mathbf{x}}_t\|^2$$
$$\leq ((1+l_1)\lambda^2 - 1)\|\mathbf{x}_t - \mathbf{1}\otimes\bar{\mathbf{x}}_t\|^2 + (1+\frac{1}{l_1})\alpha_t^2\|\mathbf{h}_{\mathbf{x}}^t - \mathbf{1}\otimes\bar{\mathbf{h}}_{\mathbf{x}}^t\|^2$$
$$\|\mathbf{h}_{\mathbf{x}}^{t+1} - \mathbf{1}\otimes\bar{\mathbf{h}}_{\mathbf{x}}^{t+1}\|^2 - \|\mathbf{h}_{\mathbf{x}}^t - \mathbf{1}\otimes\bar{\mathbf{h}}_{\mathbf{x}}^t\|^2$$
$$\leq ((1+l_2)\lambda^2 - 1)\|\mathbf{h}_{\mathbf{x}}^t - \mathbf{1}\otimes\bar{\mathbf{h}}_{\mathbf{x}}^t\|^2 + (1+\frac{1}{l_2})(L_{f_{i_1}}^2\|\mathbf{x}_{t+1} - \mathbf{x}_t\|^2 + L_{f_{i_2}}^2\beta_t^2\|\mathbf{h}_{\mathbf{y}}^t\|^2).$$

$\qquad\square$

**Step 4: Choosing suitable coefficients such that the descent of Lyapunov function is well controlled.**

The purpose of Step 4 is to identify suitable decreasing coefficients $\{a_t, b_t, c_t, d_t\}_{t=1}^{\infty}$ and a well-chosen averaging parameter $\mu_t$ such that the descent of the Lyapunov function is well-controlled and can be bounded by a summable series. This ensures the algorithm converges under the framework provided by the potential function.

In this lemma, we define a potential function $V_t$ that tracks the overall progress of the algorithm and analyze its behavior over iterations. We then establish an upper bound on the difference between $V_{t+1}$ and $V_t$, showing that the Lyapunov function $V_t$ decreases at each step, guided by well-chosen coefficients that control the descent rate.

**Lemma 10** (Potential function). *We assume that $\{a_t, b_t, c_t, d_t\}_{t=1}^{\infty}$ be a sequence of decreasing positive constants, where $V_t := a_t \left[ f(\bar{\mathbf{x}}_t, \mathbf{y}_{\mu_t}^*(\bar{\mathbf{x}}_t)) - f_0 \right] + d_t \|\mathbf{x}_t - \mathbf{1} \otimes \bar{\mathbf{x}}_t\|^2 + d_t \alpha_t \|\mathbf{h}_{\mathbf{x}}^t - \mathbf{1} \otimes \bar{\mathbf{h}}_{\mathbf{x}}^t\|^2 + b_t \|\mathbf{y}_t - \mathbf{y}_{\mu_t}^*(\mathbf{x}_t)\|^2 + c_t \|\mathbf{v}_t - \mathbf{v}_{\mu_t}^*(\mathbf{x}_t)\|^2$.*

$$
\begin{aligned}
V_{t+1} - V_t \\
\leq & -\frac{a_{t+1}\alpha_t}{2} \|\nabla\Phi_{\mu_t}(\bar{\mathbf{x}}_t)\|^2 + C_1 \|\bar{\mathbf{h}}_{\mathbf{x}}^t\|^2 + C_2 \|\mathbf{x}_t - \mathbf{1} \otimes \bar{\mathbf{x}}_t\|^2 + C_3 \left\| \mathbf{y}_t - \mathbf{y}_{\mu_t}^*(\mathbf{x}_t) \right\|^2 \\
& + C_4 \left\| \mathbf{v}_t - \mathbf{v}_{\mu_t}^*(\mathbf{x}_t) \right\|^2 + C_5 \|\mathbf{h}_{\mathbf{x}}^t - \mathbf{1} \otimes \bar{\mathbf{h}}_{\mathbf{x}}^t\|^2 + C_6 \|\mathbf{h}_{\mathbf{y}}^t\|^2 \\
& + \frac{1}{m} \sum_{i=1}^{m} \frac{2a_{t+1} \left\| \nabla_{\mathbf{y}} f_i\left( \bar{\mathbf{x}}_{t+1}, \mathbf{y}_{\mu_{t+1}}^*(\bar{\mathbf{x}}_{t+1}) \right) \right\| \left\| \mathbf{y}_{\mu_{t+1}}^*(\mathbf{x}_{i,t+1}) \right\|}{\sigma_{h_i}} \left( \frac{\mu_t - \mu_{t+1}}{\mu_t} \right) \\
& + \frac{1}{m} \sum_{i=1}^{m} \frac{2a_{t+1} L_{f_{i\mathbf{y}2}} \|\mathbf{y}_{\mu_{t+1}}^*(\bar{\mathbf{x}}_{t+1})\|^2}{\sigma_{h_i}^2} \left( \frac{\mu_t - \mu_{t+1}}{\mu_t} \right)^2 \\
& + \sum_{i=1}^{m} \frac{24 b_{t+1} \left\| \mathbf{y}_{\mu_{t+1}}^*(\mathbf{x}_{i,t+1}) \right\|^2}{\sigma_{h_i}^2 \beta_t \sigma_{\psi_{\mu_t}}} \left( \frac{\mu_t - \mu_{t+1}}{\mu_t} \right)^2 \\
& + \sum_{i=1}^{m} \frac{6 c_{t+1}}{\eta_t \sigma_{\psi_{\mu_t}}} \left( \frac{\mu_t - \mu_{t+1}}{\mu_t^2} \right)^2 \left( \left( C_{\mathbf{v}1} \left\| \mathbf{v}_{\mu_{t+1}}^*(\mathbf{x}_{i,t+1}) \right\| + L_{f_{i\mathbf{y}2}} \right) \left\| \mathbf{y}_{\mu_{t+1}}^*(\mathbf{x}_{i,t+1}) \right\| \right. \\
& \left. + C_{\mathbf{v}2} \left\| \nabla_{\mathbf{y}} f_i\left( \mathbf{x}_{i,t+1}, \mathbf{y}_{\mu_{t+1}}^*(\mathbf{x}_{i,t+1}) \right) \right\| \right)^2,
\end{aligned}
$$

*where*

$$
\begin{aligned}
C_1 = & -a_{t+1} \left( \frac{\alpha_t}{2} - \frac{\alpha_t^2 L_{\Phi\mu_t}}{2\sigma_{\psi_{\mu_t}}^2} \right) + 4\alpha_t^2 m \frac{6 b_{t+1} L_{\psi\mathbf{y}1}^2}{\beta_t \sigma_{\psi_{\mu_t}}^3} + 4\alpha_t^2 m \frac{6 c_{t+1} \left( L_{\mathbf{v}1} \left\| \mathbf{v}_{\mu_t}^*(\mathbf{x}_{i,t}) \right\| + L_{\mathbf{v}2} \right)^2}{\eta_t \sigma_{\psi_{\mu_t}}^5} \\
& + \frac{1}{1-\lambda} d_{t+1} \alpha_{t+1} L_{f_{i_1}}^2 4\alpha_t^2 m, \\
C_2 = & a_{t+1} \frac{\alpha_t L_{\Phi\mu_t}}{r\sigma_{\psi_{\mu_t}}^2 m} + 8 \left( \frac{6 b_{t+1} L_{\psi\mathbf{y}1}^2}{\beta_t \sigma_{\psi_{\mu_t}}^3} + \frac{6 c_{t+1} \left( L_{\mathbf{v}1} \left\| \mathbf{v}_{\mu_t}^*(\mathbf{x}_{i,t}) \right\| + L_{\mathbf{v}2} \right)^2}{\eta_t \sigma_{\psi_{\mu_t}}^5} \right) \\
& + d_{t+1} (\lambda - 1) + 8 \frac{1}{1-\lambda} d_{t+1} \alpha_{t+1} L_{f_{i_1}}^2, \\
C_3 = & a_{t+1} \alpha_t r \frac{1}{m} \left( L_{\psi\mathbf{xy}2} \left\| \mathbf{v}_{\mu_t}^*(\mathbf{x}_{i,t}) \right\| + L_{f_{i\mathbf{x}2}} \right)^2 \\
& - \left( \frac{b_{t+1}}{2} \beta_t \sigma_{\psi_{\mu_t}} - \frac{3 c_{t+1} \eta_t}{\sigma_{\psi_{\mu_t}}} \left( L_{\psi\mathbf{yy}2} \left\| \mathbf{v}_{\mu_t}^*(\mathbf{x}_{i,t}) \right\| + L_{f_{i\mathbf{y}2}} \right)^2 \right), \\
C_4 = & a_{t+1} \frac{\alpha_t r}{m} L_{\psi\mathbf{y}1}^2 - \frac{c_{t+1}}{2} \eta_t \sigma_{\psi_{\mu_t}}, \\
C_5 = & 4\alpha_t^2 \left( \frac{6 b_{t+1} L_{\psi\mathbf{y}1}^2}{\beta_t \sigma_{\psi_{\mu_t}}^3} + \frac{6 c_{t+1} \left( L_{\mathbf{v}1} \left\| \mathbf{v}_{\mu_t}^*(\mathbf{x}_{i,t}) \right\| + L_{\mathbf{v}2} \right)^2}{\eta_t \sigma_{\psi_{\mu_t}}^5} \right) \\
& + \frac{1}{1-\lambda} d_{t+1} \alpha_t^2 + d_{t+1} \alpha_{t+1} (\lambda - 1) + \frac{1}{1-\lambda} d_{t+1} \alpha_{t+1} L_{f_{i_1}}^2 4\alpha_t^2, \\
C_6 = & d_{t+1} \alpha_{t+1} \frac{1}{1-\lambda} L_{f_{i_2}}^2 \beta_t^2 - b_{t+1} \left( 1 + \frac{1}{2} \beta_t \sigma_{\psi_{\mu_t}} \right) \left( 2\beta_t \frac{1}{\sigma_{\psi_{\mu_t}} + L_{\psi_{\mu_t}}} - \beta_t^2 \right).
\end{aligned}
$$

*Proof.*

$$V_{t+1} - V_t$$

$$
= a_{t+1}\left[f(\bar{\mathbf{x}}_{t+1}, \mathbf{y}^*_{\mu_{t+1}}(\bar{\mathbf{x}}_{t+1})) - f(\bar{\mathbf{x}}_t, \mathbf{y}^*_{\mu_t}(\bar{\mathbf{x}}_t))\right] + (a_{t+1} - a_t)f(\bar{\mathbf{x}}_t, \mathbf{y}^*_{\mu_t}(\bar{\mathbf{x}}_t))
$$

$$
+ d_{t+1}\left[\|\mathbf{x}_{t+1} - \mathbf{1}\otimes\bar{\mathbf{x}}_{t+1}\|^2 - \|\mathbf{x}_t - \mathbf{1}\otimes\bar{\mathbf{x}}_t\|^2\right] + (d_{t+1} - d_t)\|\mathbf{x}_t - \mathbf{1}\otimes\bar{\mathbf{x}}_t\|^2
$$

$$
+ d_{t+1}\alpha_{t+1}\left[\|\mathbf{h}^{t+1}_{\mathbf{x}} - \mathbf{1}\otimes\bar{\mathbf{h}}^{t+1}_{\mathbf{x}}\|^2 - \|\mathbf{h}^t_{\mathbf{x}} - \mathbf{1}\otimes\bar{\mathbf{h}}^t_{\mathbf{x}}\|^2\right]
$$

$$
+ (d_{t+1}\alpha_{t+1} - d_t\alpha_t)\|\mathbf{h}^t_{\mathbf{x}} - \mathbf{1}\otimes\bar{\mathbf{h}}^t_{\mathbf{x}}\|^2
$$

$$
+ b_{t+1}\left[\left\|\mathbf{y}_{t+1} - \mathbf{y}^*_{\mu_{t+1}}(\mathbf{x}_{t+1})\right\|^2 - \left\|\mathbf{y}_t - \mathbf{y}^*_{\mu_t}(\mathbf{x}_t)\right\|^2\right] + (b_{t+1} - b_t)\left\|\mathbf{y}_t - \mathbf{y}^*_{\mu_t}(\mathbf{x}_t)\right\|^2
$$

$$
+ c_{t+1}\left[\left\|\mathbf{v}_{t+1} - \mathbf{v}^*_{\mu_{t+1}}(\mathbf{x}_{t+1})\right\|^2 - \left\|\mathbf{v}_t - \mathbf{v}^*_{\mu_t}(\mathbf{x}_t)\right\|^2\right] + (c_{t+1} - c_t)\left\|\mathbf{v}_t - \mathbf{v}^*_{\mu_t}(\mathbf{x}_t)\right\|^2
$$

$$
\leq a_{t+1}\left[f(\bar{\mathbf{x}}_{t+1}, \mathbf{y}^*_{\mu_{t+1}}(\bar{\mathbf{x}}_{t+1})) - f(\bar{\mathbf{x}}_t, \mathbf{y}^*_{\mu_t}(\bar{\mathbf{x}}_t))\right]
$$

$$
+ d_{t+1}\left[\|\mathbf{x}_{t+1} - \mathbf{1}\otimes\bar{\mathbf{x}}_{t+1}\|^2 - \|\mathbf{x}_t - \mathbf{1}\otimes\bar{\mathbf{x}}_t\|^2\right]
$$

$$
+ d_{t+1}\alpha_{t+1}\left[\|\mathbf{h}^{t+1}_{\mathbf{x}} - \mathbf{1}\otimes\bar{\mathbf{h}}^{t+1}_{\mathbf{x}}\|^2 - \|\mathbf{h}^t_{\mathbf{x}} - \mathbf{1}\otimes\bar{\mathbf{h}}^t_{\mathbf{x}}\|^2\right]
$$

$$
+ b_{t+1}\left[\sum_{i=1}^m \left\|\mathbf{y}_{i,t+1} - \mathbf{y}^*_{\mu_{t+1}}(\mathbf{x}_{i,t+1})\right\|^2 - \sum_{i=1}^m \left\|\mathbf{y}_{i,t} - \mathbf{y}^*_{\mu_t}(\mathbf{x}_{i,t})\right\|^2\right]
$$

$$
+ c_{t+1}\left[\sum_{i=1}^m \left\|\mathbf{v}_{i,t+1} - \mathbf{v}^*_{\mu_{t+1}}(\mathbf{x}_{i,t+1})\right\|^2 - \sum_{i=1}^m \left\|\mathbf{v}_{i,t} - \mathbf{v}^*_{\mu_t}(\mathbf{x}_{i,t})\right\|^2\right]
$$

By combining the estimates from the above lemmas, we have

$$V_{t+1} - V_t$$

$$
\leq a_{t+1}\left[-\frac{\alpha_t}{2}\left\|\nabla\Phi_{\mu_t}(\bar{\mathbf{x}}_t)\right\|^2 - \left(\frac{\alpha_t}{2} - \frac{\alpha_t^2 L_{\Phi\mu_t}}{2\sigma^2_{\psi_{\mu_t}}}\right)\|\bar{\mathbf{h}}^t_{\mathbf{x}}\|^2 + \frac{\alpha_t L_{\Phi\mu_t}}{r\sigma^2_{\psi_{\mu_t}}}\frac{1}{m}\sum_{i=1}^m\|\bar{\mathbf{x}}_t - \mathbf{x}_{i,t}\|^2\right.
$$

$$
+ \frac{\alpha_t r}{m}\sum_{i=1}^m\left(L_{\psi_{\mathbf{xy}2}}\left\|\mathbf{v}^*_{\mu_t}(\mathbf{x}_{i,t})\right\| + L_{f_i\mathbf{x}2}\right)^2\left\|\mathbf{y}_{i,t} - \mathbf{y}^*_{\mu_t}(\mathbf{x}_{i,t})\right\|^2
$$

$$
+ \frac{\alpha_t r L^2_{\psi_{\mathbf{y}1}}}{m}\sum_{i=1}^m\left\|\mathbf{v}_{i,t} - \mathbf{v}^*_{\mu_t}(\mathbf{x}_{i,t})\right\|^2 + 2\alpha_t r\left\|\frac{1}{m}\sum_{i=1}^m \mathbf{d}^{i,t}_{\mathbf{x}} - \bar{\mathbf{h}}^t_{\mathbf{x}}\right\|^2
$$

$$
+ \frac{1}{m}\sum_{i=1}^m\left[\frac{2\left\|\nabla_{\mathbf{y}}f_i\left(\bar{\mathbf{x}}_{t+1}, \mathbf{y}^*_{\mu_{t+1}}(\bar{\mathbf{x}}_{t+1})\right)\right\|\left\|\mathbf{y}^*_{\mu_{t+1}}(\bar{\mathbf{x}}_{t+1})\right\|}{\sigma_{h_i}}\left(\frac{\mu_t - \mu_{t+1}}{\mu_t}\right)\right]
$$

$$
+ \frac{1}{m}\sum_{i=1}^m\frac{2L_{f_i\mathbf{y}2}\|\mathbf{y}^*_{\mu_{t+1}}(\bar{\mathbf{x}}_{t+1})\|^2}{\sigma^2_{h_i}}\left(\frac{\mu_t - \mu_{t+1}}{\mu_t}\right)^2\Bigg]
$$

$$
+ b_{t+1}\sum_{i=1}^m\left[-\frac{1}{2}\beta_t\sigma_{\psi_{\mu_t}}\left\|\mathbf{y}_{i,t} - \mathbf{y}^*_{\mu_t}(\mathbf{x}_{i,t})\right\|^2 + \frac{6L^2_{\psi_{\mathbf{y}1}}}{\beta_t\sigma^3_{\psi_{\mu_t}}}\|\mathbf{x}_{i,t} - \mathbf{x}_{i,t+1}\|^2\right.
$$

$$
- \left(1 + \frac{1}{2}\beta_t\sigma_{\psi_{\mu_t}}\right)\left(2\beta_t\frac{1}{\sigma_{\psi_{\mu_t}} + L_{\psi_{\mu_t}}} - \beta_t^2\right)\|\mathbf{h}^{i,t}_{\mathbf{y}}\|^2
$$

$$
\left.+ \frac{24\left\|\mathbf{y}^*_{\mu_{t+1}}(\mathbf{x}_{i,t+1})\right\|^2}{\sigma^2_{h_i}\beta_t\sigma_{\psi_{\mu_t}}}\left(\frac{\mu_t - \mu_{t+1}}{\mu_t}\right)^2\right]
$$

$$
+ c_{t+1}\sum_{i=1}^m\left[-\frac{1}{2}\eta_t\sigma_{\psi_{\mu_t}}\left\|\mathbf{v}_{i,t} - \mathbf{v}^*_{\mu_t}(\mathbf{x}_{i,t})\right\|^2\right.
$$

$$+ \frac{6 \left( L_{\mathbf{v}1} \left\| \mathbf{v}_{\mu_t}^* \left( \mathbf{x}_{i,t} \right) \right\| + L_{\mathbf{v}2} \right)^2}{\eta_t \sigma_{\psi_{\mu_t}}^5} \left\| \mathbf{x}_{i,t} - \mathbf{x}_{i,t+1} \right\|^2$$

$$+ \frac{3\eta_t}{\sigma_{\psi_{\mu_t}}} \left( L_{\psi_{\mathbf{yy}2}} \left\| \mathbf{v}_{\mu_t}^* \left( \mathbf{x}_{i,t} \right) \right\| + L_{f_{i\mathbf{y}2}} \right)^2 \left\| \mathbf{y}_{i,t} - \mathbf{y}_{\mu_t}^* \left( \mathbf{x}_{i,t} \right) \right\|^2$$

$$+ \frac{6}{\eta_t \sigma_{\psi_{\mu_t}}} \left( \frac{\mu_t - \mu_{t+1}}{\mu_t^2} \right)^2 \left( \left( C_{\mathbf{v}1} \left\| \mathbf{v}_{\mu_{t+1}}^* \left( \mathbf{x}_{i,t+1} \right) \right\| + L_{f_{i\mathbf{y}2}} \right) \left\| \mathbf{y}_{\mu_{t+1}}^* \left( \mathbf{x}_{i,t+1} \right) \right\| \right.$$

$$\left. + C_{\mathbf{v}2} \left\| \nabla_{\mathbf{y}} f_i \left( \mathbf{x}_{i,t+1}, \mathbf{y}_{\mu_{t+1}}^* \left( \mathbf{x}_{i,t+1} \right) \right) \right\| \right)^2 \right]$$

$$+ d_{t+1} \left[ ((1 + l_1)\lambda^2 - 1) \| \mathbf{x}_t - \mathbf{1} \otimes \bar{\mathbf{x}}_t \|^2 + (1 + \frac{1}{l_1}) \alpha_t^2 \| \mathbf{h}_{\mathbf{x}}^t - \mathbf{1} \otimes \bar{\mathbf{h}}_{\mathbf{x}}^t \|^2 \right]$$

$$+ d_{t+1} \alpha_{t+1} ((1 + l_2)\lambda^2 - 1) \| \mathbf{h}_{\mathbf{x}}^t - \mathbf{1} \otimes \bar{\mathbf{h}}_{\mathbf{x}}^t \|^2$$

$$+ d_{t+1} \alpha_{t+1} (1 + \frac{1}{l_2}) (L_{f_{i_1}}^2 \| \mathbf{x}_{t+1} - \mathbf{x}_t \|^2 + L_{f_{i_2}}^2 \beta_t^2 \| \mathbf{h}_{\mathbf{y}}^t \|^2)$$

Rearranging it, we have

$$V_{t+1} - V_t$$

$$\leq - \frac{a_{t+1}\alpha_t}{2} \left\| \nabla \Phi_{\mu_t}(\bar{\mathbf{x}}_t) \right\|^2 - a_{t+1} \left( \frac{\alpha_t}{2} - \frac{\alpha_t^2 L_{\Phi\mu_t}}{2\sigma_{\psi_{\mu_t}}^2} \right) \| \bar{\mathbf{h}}_{\mathbf{x}}^t \|^2$$

$$+ a_{t+1} \frac{\alpha_t L_{\Phi\mu_t}}{r \sigma_{\psi_{\mu_t}}^2 m} \| \mathbf{x}_t - \mathbf{1} \otimes \bar{\mathbf{x}}_t \|^2$$

$$+ a_{t+1} \frac{\alpha_t r}{m} \sum_{i=1}^m \left( L_{\psi_{\mathbf{xy}2}} \left\| \mathbf{v}_{\mu_t}^* \left( \mathbf{x}_{i,t} \right) \right\| + L_{f_{i\mathbf{x}2}} \right)^2 \left\| \mathbf{y}_{i,t} - \mathbf{y}_{\mu_t}^* \left( \mathbf{x}_{i,t} \right) \right\|^2$$

$$+ a_{t+1} \frac{\alpha_t r L_{\psi_{\mathbf{y}1}}^2}{m} \sum_{i=1}^m \left\| \mathbf{v}_{i,t} - \mathbf{v}_{\mu_t}^* \left( \mathbf{x}_{i,t} \right) \right\|^2 + 2a_{t+1} \alpha_t r \left\| \frac{1}{m} \sum_{i=1}^m \mathbf{d}_{\mathbf{x}}^{i,t} - \bar{\mathbf{h}}_{\mathbf{x}}^t \right\|^2$$

$$+ a_{t+1} \frac{1}{m} \sum_{i=1}^m \left[ \frac{2 \left\| \nabla_{\mathbf{y}} f_i \left( \bar{\mathbf{x}}_{t+1}, \mathbf{y}_{\mu_{t+1}}^* (\bar{\mathbf{x}}_{t+1}) \right) \right\| \left\| \mathbf{y}_{\mu_{t+1}}^* (\bar{\mathbf{x}}_{t+1}) \right\|}{\sigma_{h_i}} \left( \frac{\mu_t - \mu_{t+1}}{\mu_t} \right) \right]$$

$$+ a_{t+1} \frac{1}{m} \sum_{i=1}^m \frac{2 L_{f_{i\mathbf{y}2}} \| \mathbf{y}_{\mu_{t+1}}^* (\bar{\mathbf{x}}_{t+1}) \|^2}{\sigma_{h_i}^2} \left( \frac{\mu_t - \mu_{t+1}}{\mu_t} \right)^2$$

$$- \frac{b_{t+1}}{2} \beta_t \sigma_{\psi_{\mu_t}} \sum_{i=1}^m \left\| \mathbf{y}_{i,t} - \mathbf{y}_{\mu_t}^* \left( \mathbf{x}_{i,t} \right) \right\|^2 + \frac{6 b_{t+1} L_{\psi_{\mathbf{y}1}}^2}{\beta_t \sigma_{\psi_{\mu_t}}^3} \sum_{i=1}^m \left\| \mathbf{x}_{i,t} - \mathbf{x}_{i,t+1} \right\|^2$$

$$- b_{t+1} \left( 1 + \frac{1}{2} \beta_t \sigma_{\psi_{\mu_t}} \right) \left( 2\beta_t \frac{1}{\sigma_{\psi_{\mu_t}} + L_{\psi_{\mu_t}}} - \beta_t^2 \right) \sum_{i=1}^m \| \mathbf{h}_{\mathbf{y}}^{i,t} \|^2$$

$$+ \sum_{i=1}^m \frac{24 b_{t+1} \left\| \mathbf{y}_{\mu_{t+1}}^* \left( \mathbf{x}_{i,t+1} \right) \right\|^2}{\sigma_{h_i}^2 \beta_t \sigma_{\psi_{\mu_t}}} \left( \frac{\mu_t - \mu_{t+1}}{\mu_t} \right)^2$$

$$- \frac{c_{t+1}}{2} \eta_t \sigma_{\psi_{\mu_t}} \sum_{i=1}^m \left\| \mathbf{v}_{i,t} - \mathbf{v}_{\mu_t}^* \left( \mathbf{x}_{i,t} \right) \right\|^2$$

$$+ \sum_{i=1}^m \frac{6 c_{t+1} \left( L_{\mathbf{v}1} \left\| \mathbf{v}_{\mu_t}^* \left( \mathbf{x}_{i,t} \right) \right\| + L_{\mathbf{v}2} \right)^2}{\eta_t \sigma_{\psi_{\mu_t}}^5} \left\| \mathbf{x}_{i,t} - \mathbf{x}_{i,t+1} \right\|^2$$

$$+ \sum_{i=1}^m \frac{3 c_{t+1} \eta_t}{\sigma_{\psi_{\mu_t}}} \left( L_{\psi_{\mathbf{yy}2}} \left\| \mathbf{v}_{\mu_t}^* \left( \mathbf{x}_{i,t} \right) \right\| + L_{f_{i\mathbf{y}2}} \right)^2 \left\| \mathbf{y}_{i,t} - \mathbf{y}_{\mu_t}^* \left( \mathbf{x}_{i,t} \right) \right\|^2$$

$$+ \sum_{i=1}^{m} \frac{6c_{t+1}}{\eta_t \sigma_{\psi_{\mu_t}}} \left( \frac{\mu_t - \mu_{t+1}}{\mu_t^2} \right)^2 \left( \left( C_{\mathbf{v}1} \left\| \mathbf{v}^*_{\mu_{t+1}} (\mathbf{x}_{i,t+1}) \right\| + L_{f_{i\mathbf{y}2}} \right) \left\| \mathbf{y}^*_{\mu_{t+1}} (\mathbf{x}_{i,t+1}) \right\| \right.$$

$$\left. + C_{\mathbf{v}2} \left\| \nabla_{\mathbf{y}} f_i \left( \mathbf{x}_{i,t+1}, \mathbf{y}^*_{\mu_{t+1}} (\mathbf{x}_{i,t+1}) \right) \right\| \right)^2$$

$$+ d_{t+1} \left[ ((1+l_1)\lambda^2 - 1)\|\mathbf{x}_t - \mathbf{1} \otimes \bar{\mathbf{x}}_t\|^2 + (1 + \frac{1}{l_1})\alpha_t^2 \|\mathbf{h}_{\mathbf{x}}^t - \mathbf{1} \otimes \bar{\mathbf{h}}_{\mathbf{x}}^t\|^2 \right]$$

$$+ d_{t+1}\alpha_{t+1}((1+l_2)\lambda^2 - 1)\|\mathbf{h}_{\mathbf{x}}^t - \mathbf{1} \otimes \bar{\mathbf{h}}_{\mathbf{x}}^t\|^2$$

$$+ d_{t+1}\alpha_{t+1}(1 + \frac{1}{l_2})(L_{f_{i_1}}^2 \|\mathbf{x}_{t+1} - \mathbf{x}_t\|^2 + L_{f_{i_2}}^2 \beta_t^2 \|\mathbf{h}_{\mathbf{y}}^t\|^2)$$

Rearranging it, we have

$$V_{t+1} - V_t$$

$$\leq - \frac{a_{t+1}\alpha_t}{2}\|\nabla\Phi_{\mu_t}(\bar{\mathbf{x}}_t)\|^2 - a_{t+1}\left(\frac{\alpha_t}{2} - \frac{\alpha_t^2 L_{\Phi\mu_t}}{2\sigma_{\psi_{\mu_t}}^2}\right)\|\bar{\mathbf{h}}_{\mathbf{x}}^t\|^2$$

$$+ a_{t+1}\frac{\alpha_t L_{\Phi\mu_t}}{r\sigma_{\psi_{\mu_t}}^2 m}\|\mathbf{x}_t - \mathbf{1} \otimes \bar{\mathbf{x}}_t\|^2$$

$$+ 2a_{t+1}\alpha_t r \left\| \frac{1}{m}\sum_{i=1}^{m} \mathbf{d}_{\mathbf{x}}^{i,t} - \bar{\mathbf{h}}_{\mathbf{x}}^t \right\|^2 + a_{t+1}\frac{\alpha_t r L_{\psi\mathbf{y}1}^2}{m}\sum_{i=1}^{m}\left\| \mathbf{v}_{i,t} - \mathbf{v}^*_{\mu_t}(\mathbf{x}_{i,t}) \right\|^2$$

$$+ a_{t+1}\frac{\alpha_t r}{m}\sum_{i=1}^{m}\left(L_{\psi\mathbf{xy}2}\left\|\mathbf{v}^*_{\mu_t}(\mathbf{x}_{i,t})\right\| + L_{f_{i\mathbf{x}2}}\right)^2 \left\|\mathbf{y}_{i,t} - \mathbf{y}^*_{\mu_t}(\mathbf{x}_{i,t})\right\|^2$$

$$+ \sum_{i=1}^{m}\left(\frac{6b_{t+1}L_{\psi\mathbf{y}1}^2}{\beta_t \sigma_{\psi_{\mu_t}}^3} + \frac{6c_{t+1}\left(L_{\mathbf{v}1}\left\|\mathbf{v}^*_{\mu_t}(\mathbf{x}_{i,t})\right\| + L_{\mathbf{v}2}\right)^2}{\eta_t \sigma_{\psi_{\mu_t}}^5}\right)\|\mathbf{x}_{i,t} - \mathbf{x}_{i,t+1}\|^2$$

$$- \sum_{i=1}^{m}\left(\frac{b_{t+1}}{2}\beta_t \sigma_{\psi_{\mu_t}} - \frac{3c_{t+1}\eta_t}{\sigma_{\psi_{\mu_t}}}\left(L_{\psi\mathbf{yy}2}\left\|\mathbf{v}^*_{\mu_t}(\mathbf{x}_{i,t})\right\| + L_{f_{i\mathbf{y}2}}\right)^2\right)\left\|\mathbf{y}_{i,t} - \mathbf{y}^*_{\mu_t}(\mathbf{x}_{i,t})\right\|^2$$

$$- b_{t+1}\left(1 + \frac{1}{2}\beta_t \sigma_{\psi_{\mu_t}}\right)\left(2\beta_t \frac{1}{\sigma_{\psi_{\mu_t}} + L_{\psi_{\mu_t}}} - \beta_t^2\right)\sum_{i=1}^{m}\|\mathbf{h}_{\mathbf{y}}^{i,t}\|^2$$

$$- \frac{c_{t+1}}{2}\eta_t \sigma_{\psi_{\mu_t}}\sum_{i=1}^{m}\left\|\mathbf{v}_{i,t} - \mathbf{v}^*_{\mu_t}(\mathbf{x}_{i,t})\right\|^2$$

$$+ a_{t+1}\frac{1}{m}\sum_{i=1}^{m}\left[\frac{2\left\|\nabla_{\mathbf{y}}f_i\left(\bar{\mathbf{x}}_{t+1}, \mathbf{y}^*_{\mu_{t+1}}(\bar{\mathbf{x}}_{t+1})\right)\right\|\left\|\mathbf{y}^*_{\mu_{t+1}}(\bar{\mathbf{x}}_{t+1})\right\|}{\sigma_{h_i}}\left(\frac{\mu_t - \mu_{t+1}}{\mu_t}\right)\right]$$

$$+ a_{t+1}\frac{1}{m}\sum_{i=1}^{m}\frac{2L_{f_{i\mathbf{y}2}}\|\mathbf{y}^*_{\mu_{t+1}}(\bar{\mathbf{x}}_{t+1})\|^2}{\sigma_{h_i}^2}\left(\frac{\mu_t - \mu_{t+1}}{\mu_t}\right)^2$$

$$+ \sum_{i=1}^{m}\frac{24b_{t+1}\left\|\mathbf{y}^*_{\mu_{t+1}}(\mathbf{x}_{i,t+1})\right\|^2}{\sigma_{h_i}^2 \beta_t \sigma_{\psi_{\mu_t}}}\left(\frac{\mu_t - \mu_{t+1}}{\mu_t}\right)^2$$

$$+ \sum_{i=1}^{m}\frac{6c_{t+1}}{\eta_t \sigma_{\psi_{\mu_t}}}\left(\frac{\mu_t - \mu_{t+1}}{\mu_t^2}\right)^2\left(\left(C_{\mathbf{v}1}\left\|\mathbf{v}^*_{\mu_{t+1}}(\mathbf{x}_{i,t+1})\right\| + L_{f_{i\mathbf{y}2}}\right)\left\|\mathbf{y}^*_{\mu_{t+1}}(\mathbf{x}_{i,t+1})\right\|\right.$$

$$\left. + C_{\mathbf{v}2}\left\|\nabla_{\mathbf{y}}f_i\left(\mathbf{x}_{i,t+1}, \mathbf{y}^*_{\mu_{t+1}}(\mathbf{x}_{i,t+1})\right)\right\|\right)^2$$

$$+ d_{t+1}\left[((1+l_1)\lambda^2 - 1)\|\mathbf{x}_t - \mathbf{1} \otimes \bar{\mathbf{x}}_t\|^2 + (1 + \frac{1}{l_1})\alpha_t^2 \|\mathbf{h}_{\mathbf{x}}^t - \mathbf{1} \otimes \bar{\mathbf{h}}_{\mathbf{x}}^t\|^2\right]$$

$$+ d_{t+1}\alpha_{t+1}((1+l_2)\lambda^2 - 1)\|\mathbf{h}_{\mathbf{x}}^t - \mathbf{1} \otimes \bar{\mathbf{h}}_{\mathbf{x}}^t\|^2$$

$$+ d_{t+1}\alpha_{t+1}(1 + \frac{1}{l_2})(L_{f_{i_1}}^2 \|\mathbf{x}_{t+1} - \mathbf{x}_t\|^2 + L_{f_{i_2}}^2 \beta_t^2 \|\mathbf{h}_{\mathbf{y}}^t\|^2).$$

Since $\left\|\mathbf{v}_{\mu_t}^* (\mathbf{x}_{i,t})\right\|$ is of order $O(\frac{1}{\sigma_{\psi_{\mu_t}}})$, to simplify the proof, we take $\left\|\mathbf{v}_{\mu_t}^* (\mathbf{x}_{i,t})\right\|$ as its maximal value over all $m$ agents at iteration $t$. By further calculating the terms from the previous result, we have

$$V_{t+1} - V_t$$

$$\leq - \frac{a_{t+1}\alpha_t}{2} \|\nabla\Phi_{\mu_t}(\bar{\mathbf{x}}_t)\|^2 - a_{t+1} \left( \frac{\alpha_t}{2} - \frac{\alpha_t^2 L_{\Phi\mu_t}}{2\sigma_{\psi_{\mu_t}}^2} \right) \|\bar{\mathbf{h}}_{\mathbf{x}}^t\|^2$$

$$+ a_{t+1} \frac{\alpha_t L_{\Phi\mu_t}}{r\sigma_{\psi_{\mu_t}}^2 m} \|\mathbf{x}_t - \mathbf{1}\otimes\bar{\mathbf{x}}_t\|^2 + 2a_{t+1}\alpha_t r \left\| \frac{1}{m}\sum_{i=1}^m \mathbf{d}_{\mathbf{x}}^{i,t} - \bar{\mathbf{h}}_{\mathbf{x}}^t \right\|^2$$

$$+ a_{t+1}\alpha_t r \frac{1}{m} \left( L_{\psi\mathbf{xy2}} \left\|\mathbf{v}_{\mu_t}^* (\mathbf{x}_{i,t})\right\| + L_{f_{i\mathbf{x2}}} \right)^2 \left\|\mathbf{y}_t - \mathbf{y}_{\mu_t}^* (\mathbf{x}_t)\right\|^2$$

$$- \left( \frac{b_{t+1}}{2}\beta_t\sigma_{\psi_{\mu_t}} - \frac{3c_{t+1}\eta_t}{\sigma_{\psi_{\mu_t}}} \left( L_{\psi\mathbf{yy2}} \left\|\mathbf{v}_{\mu_t}^* (\mathbf{x}_{i,t})\right\| + L_{f_{i\mathbf{y2}}} \right)^2 \right) \left\|\mathbf{y}_t - \mathbf{y}_{\mu_t}^* (\mathbf{x}_t)\right\|^2$$

$$+ a_{t+1} \frac{\alpha_t r}{m} L_{\psi\mathbf{y1}}^2 \left\|\mathbf{v}_t - \mathbf{v}_{\mu_t}^* (\mathbf{x}_t)\right\|^2 - \frac{c_{t+1}}{2}\eta_t\sigma_{\psi_{\mu_t}} \left\|\mathbf{v}_t - \mathbf{v}_{\mu_t}^* (\mathbf{x}_t)\right\|^2$$

$$- b_{t+1} \left( 1 + \frac{1}{2}\beta_t\sigma_{\psi_{\mu_t}} \right) \left( 2\beta_t \frac{1}{\sigma_{\psi_{\mu_t}} + L_{\psi_{\mu_t}}} - \beta_t^2 \right) \sum_{i=1}^m \|\mathbf{h}_{\mathbf{y}}^{i,t}\|^2$$

$$+ \left( \frac{6b_{t+1}L_{\psi\mathbf{y1}}^2}{\beta_t\sigma_{\psi_{\mu_t}}^3} + \frac{6c_{t+1} \left( L_{\mathbf{v1}} \left\|\mathbf{v}_{\mu_t}^* (\mathbf{x}_{i,t})\right\| + L_{\mathbf{v2}} \right)^2}{\eta_t\sigma_{\psi_{\mu_t}}^5} \right) \|\mathbf{x}_t - \mathbf{x}_{t+1}\|^2$$

$$+ \frac{1}{m}\sum_{i=1}^m \frac{2a_{t+1} \left\|\nabla_{\mathbf{y}}f_i \left(\bar{\mathbf{x}}_{t+1}, \mathbf{y}_{\mu_{t+1}}^* (\bar{\mathbf{x}}_{t+1})\right)\right\| \left\|\mathbf{y}_{\mu_{t+1}}^* (\mathbf{x}_{i,t+1})\right\|}{\sigma_{h_i}} \left( \frac{\mu_t - \mu_{t+1}}{\mu_t} \right)$$

$$+ \frac{1}{m}\sum_{i=1}^m \frac{2a_{t+1}L_{f_{i\mathbf{y2}}} \|\mathbf{y}_{\mu_{t+1}}^* (\bar{\mathbf{x}}_{t+1})\|^2}{\sigma_{h_i}^2} \left( \frac{\mu_t - \mu_{t+1}}{\mu_t} \right)^2$$

$$+ \sum_{i=1}^m \frac{24b_{t+1} \left\|\mathbf{y}_{\mu_{t+1}}^* (\mathbf{x}_{i,t+1})\right\|^2}{\sigma_{h_i}^2 \beta_t\sigma_{\psi_{\mu_t}}} \left( \frac{\mu_t - \mu_{t+1}}{\mu_t} \right)^2$$

$$+ \sum_{i=1}^m \frac{6c_{t+1}}{\eta_t\sigma_{\psi_{\mu_t}}} \left( \frac{\mu_t - \mu_{t+1}}{\mu_t^2} \right)^2 \left( \left( C_{\mathbf{v1}} \left\|\mathbf{v}_{\mu_{t+1}}^* (\mathbf{x}_{i,t+1})\right\| + L_{f_{i\mathbf{y2}}} \right) \left\|\mathbf{y}_{\mu_{t+1}}^* (\mathbf{x}_{i,t+1})\right\| \right.$$

$$\left. + C_{\mathbf{v2}} \left\|\nabla_{\mathbf{y}}f_i \left(\mathbf{x}_{i,t+1}, \mathbf{y}_{\mu_{t+1}}^* (\mathbf{x}_{i,t+1})\right)\right\| \right)^2$$

$$+ d_{t+1} \left[ ((1 + l_1)\lambda^2 - 1)\|\mathbf{x}_t - \mathbf{1}\otimes\bar{\mathbf{x}}_t\|^2 + (1 + \frac{1}{l_1})\alpha_t^2 \|\mathbf{h}_{\mathbf{x}}^t - \mathbf{1}\otimes\bar{\mathbf{h}}_{\mathbf{x}}^t\|^2 \right]$$

$$+ d_{t+1}\alpha_{t+1}((1 + l_2)\lambda^2 - 1)\|\mathbf{h}_{\mathbf{x}}^t - \mathbf{1}\otimes\bar{\mathbf{h}}_{\mathbf{x}}^t\|^2$$

$$+ d_{t+1}\alpha_{t+1}(1 + \frac{1}{l_2})(L_{f_{i_1}}^2 \|\mathbf{x}_{t+1} - \mathbf{x}_t\|^2 + L_{f_{i_2}}^2 \beta_t^2 \|\mathbf{h}_{\mathbf{y}}^t\|^2).$$

Then, we obtain the following expression:

$$V_{t+1} - V_t$$

$$\leq - \frac{a_{t+1}\alpha_t}{2} \|\nabla\Phi_{\mu_t}(\bar{\mathbf{x}}_t)\|^2 - a_{t+1} \left( \frac{\alpha_t}{2} - \frac{\alpha_t^2 L_{\Phi\mu_t}}{2\sigma_{\psi_{\mu_t}}^2} \right) \|\bar{\mathbf{h}}_{\mathbf{x}}^t\|^2$$

$$+ a_{t+1} \frac{\alpha_t L_{\Phi\mu_t}}{r\sigma_{\psi_{\mu_t}}^2 m} \|\mathbf{x}_t - \mathbf{1}\otimes\bar{\mathbf{x}}_t\|^2 + 2a_{t+1}\alpha_t r \left\| \frac{1}{m}\sum_{i=1}^m \mathbf{d}_{\mathbf{x}}^{i,t} - \bar{\mathbf{h}}_{\mathbf{x}}^t \right\|^2$$

$$+ a_{t+1}\alpha_t r \frac{1}{m} \left( L_{\psi\mathbf{xy2}} \left\|\mathbf{v}_{\mu_t}^* (\mathbf{x}_{i,t})\right\| + L_{f_{i\mathbf{x2}}} \right)^2 \left\|\mathbf{y}_t - \mathbf{y}_{\mu_t}^* (\mathbf{x}_t)\right\|^2$$

$$- \left( \frac{b_{t+1}}{2} \beta_t \sigma_{\psi_{\mu_t}} - \frac{3c_{t+1}\eta_t}{\sigma_{\psi_{\mu_t}}} \left( L_{\psi_{\mathbf{yy2}}} \left\| \mathbf{v}^*_{\mu_t}(\mathbf{x}_{i,t}) \right\| + L_{f_{i\mathbf{y2}}} \right)^2 \right) \left\| \mathbf{y}_t - \mathbf{y}^*_{\mu_t}(\mathbf{x}_t) \right\|^2$$

$$- b_{t+1} \left( 1 + \frac{1}{2} \beta_t \sigma_{\psi_{\mu_t}} \right) \left( 2\beta_t \frac{1}{\sigma_{\psi_{\mu_t}} + L_{\psi_{\mu_t}}} - \beta_t^2 \right) \sum_{i=1}^m \|\mathbf{h}^{i,t}_{\mathbf{y}}\|^2$$

$$+ a_{t+1} \frac{\alpha_t r}{m} L^2_{\psi_{\mathbf{y1}}} \left\| \mathbf{v}_t - \mathbf{v}^*_{\mu_t}(\mathbf{x}_t) \right\|^2 - \frac{c_{t+1}}{2} \eta_t \sigma_{\psi_{\mu_t}} \left\| \mathbf{v}_t - \mathbf{v}^*_{\mu_t}(\mathbf{x}_t) \right\|^2$$

$$+ \left( \frac{6b_{t+1} L^2_{\psi_{\mathbf{y1}}}}{\beta_t \sigma^3_{\psi_{\mu_t}}} + \frac{6c_{t+1} \left( L_{\mathbf{v1}} \left\| \mathbf{v}^*_{\mu_t}(\mathbf{x}_{i,t}) \right\| + L_{\mathbf{v2}} \right)^2}{\eta_t \sigma^5_{\psi_{\mu_t}}} \right) \times$$

$$\left( 8\|(\mathbf{x}_t - \mathbf{1} \otimes \bar{\mathbf{x}}_t)\|^2 + 4\alpha_t^2 \|\mathbf{h}^t_{\mathbf{x}} - \mathbf{1} \otimes \bar{\mathbf{h}}^t_{\mathbf{x}}\|^2 + 4\alpha_t^2 m \|\bar{\mathbf{h}}^t_{\mathbf{x}}\|^2 \right)$$

$$+ \frac{1}{m} \sum_{i=1}^m \frac{2a_{t+1} \left\| \nabla_{\mathbf{y}} f_i \left( \bar{\mathbf{x}}_{t+1}, \mathbf{y}^*_{\mu_{t+1}}(\bar{\mathbf{x}}_{t+1}) \right) \right\| \left\| \mathbf{y}^*_{\mu_{t+1}}(\mathbf{x}_{i,t+1}) \right\|}{\sigma_{h_i}} \left( \frac{\mu_t - \mu_{t+1}}{\mu_t} \right)$$

$$+ \frac{1}{m} \sum_{i=1}^m \frac{2a_{t+1} L_{f_{i\mathbf{y2}}} \|\mathbf{y}^*_{\mu_{t+1}}(\bar{\mathbf{x}}_{t+1})\|^2}{\sigma^2_{h_i}} \left( \frac{\mu_t - \mu_{t+1}}{\mu_t} \right)^2$$

$$+ \sum_{i=1}^m \frac{24b_{t+1} \left\| \mathbf{y}^*_{\mu_{t+1}}(\mathbf{x}_{i,t+1}) \right\|^2}{\sigma^2_{h_i} \beta_t \sigma_{\psi_{\mu_t}}} \left( \frac{\mu_t - \mu_{t+1}}{\mu_t} \right)^2$$

$$+ \sum_{i=1}^m \frac{6c_{t+1}}{\eta_t \sigma_{\psi_{\mu_t}}} \left( \frac{\mu_t - \mu_{t+1}}{\mu_t^2} \right)^2 \left( \left( C_{\mathbf{v1}} \left\| \mathbf{v}^*_{\mu_{t+1}}(\mathbf{x}_{i,t+1}) \right\| + L_{f_{i\mathbf{y2}}} \right) \left\| \mathbf{y}^*_{\mu_{t+1}}(\mathbf{x}_{i,t+1}) \right\| \right.$$

$$\left. + C_{\mathbf{v2}} \left\| \nabla_{\mathbf{y}} f_i \left( \mathbf{x}_{i,t+1}, \mathbf{y}^*_{\mu_{t+1}}(\mathbf{x}_{i,t+1}) \right) \right\| \right)^2$$

$$+ d_{t+1} \left[ ((1 + l_1)\lambda^2 - 1) \|\mathbf{x}_t - \mathbf{1} \otimes \bar{\mathbf{x}}_t\|^2 + (1 + \frac{1}{l_1})\alpha_t^2 \|\mathbf{h}^t_{\mathbf{x}} - \mathbf{1} \otimes \bar{\mathbf{h}}^t_{\mathbf{x}}\|^2 \right]$$

$$+ d_{t+1}\alpha_{t+1}((1 + l_2)\lambda^2 - 1)\|\mathbf{h}^t_{\mathbf{x}} - \mathbf{1} \otimes \bar{\mathbf{h}}^t_{\mathbf{x}}\|^2$$

$$+ d_{t+1}\alpha_{t+1}(1 + \frac{1}{l_2})(L^2_{f_{i_1}} \|\mathbf{x}_{t+1} - \mathbf{x}_t\|^2 + L^2_{f_{i_2}} \beta_t^2 \|\mathbf{h}^t_{\mathbf{y}}\|^2).$$

Then, we obtain the following expression:

$$V_{t+1} - V_t$$
$$\leq - \frac{a_{t+1}\alpha_t}{2} \|\nabla\Phi_{\mu_t}(\bar{\mathbf{x}}_t)\|^2$$

$$- \left( a_{t+1} \left( \frac{\alpha_t}{2} - \frac{\alpha_t^2 L_{\Phi_{\mu_t}}}{2\sigma^2_{\psi_{\mu_t}}} \right) - 4\alpha_t^2 m \frac{6b_{t+1} L^2_{\psi_{\mathbf{y1}}}}{\beta_t \sigma^3_{\psi_{\mu_t}}} \right.$$

$$\left. - 4\alpha_t^2 m \frac{6c_{t+1} \left( L_{\mathbf{v1}} \left\| \mathbf{v}^*_{\mu_t}(\mathbf{x}_{i,t}) \right\| + L_{\mathbf{v2}} \right)^2}{\eta_t \sigma^5_{\psi_{\mu_t}}} \right) \|\bar{\mathbf{h}}^t_{\mathbf{x}}\|^2$$

$$+ \left( a_{t+1} \frac{\alpha_t L_{\Phi_{\mu_t}}}{r\sigma^2_{\psi_{\mu_t}} m} + 8 \left( \frac{6b_{t+1} L^2_{\psi_{\mathbf{y1}}}}{\beta_t \sigma^3_{\psi_{\mu_t}}} + \frac{6c_{t+1} \left( L_{\mathbf{v1}} \left\| \mathbf{v}^*_{\mu_t}(\mathbf{x}_{i,t}) \right\| + L_{\mathbf{v2}} \right)^2}{\eta_t \sigma^5_{\psi_{\mu_t}}} \right) \right)$$

$$\times \|\mathbf{x}_t - \mathbf{1} \otimes \bar{\mathbf{x}}_t\|^2$$

$$+ \left( a_{t+1}\alpha_t r \frac{1}{m} \left( L_{\psi_{\mathbf{xy2}}} \left\| \mathbf{v}^*_{\mu_t}(\mathbf{x}_{i,t}) \right\| + L_{f_{i\mathbf{x2}}} \right)^2 \right.$$

$$\left. - \left( \frac{b_{t+1}}{2} \beta_t \sigma_{\psi_{\mu_t}} - \frac{3c_{t+1}\eta_t}{\sigma_{\psi_{\mu_t}}} \left( L_{\psi_{\mathbf{yy2}}} \left\| \mathbf{v}^*_{\mu_t}(\mathbf{x}_{i,t}) \right\| + L_{f_{i\mathbf{y2}}} \right)^2 \right) \right) \times$$

$$\left\| \mathbf{y}_t - \mathbf{y}^*_{\mu_t}(\mathbf{x}_t) \right\|^2 - b_{t+1} \left( 1 + \frac{1}{2} \beta_t \sigma_{\psi_{\mu_t}} \right) \left( 2\beta_t \frac{1}{\sigma_{\psi_{\mu_t}} + L_{\psi_{\mu_t}}} - \beta_t^2 \right) \|\mathbf{h}^t_{\mathbf{y}}\|^2$$

$$+ \left( a_{t+1} \frac{\alpha_t r}{m} L^2_{\psi_{\mathbf{y}1}} - \frac{c_{t+1}}{2} \eta_t \sigma_{\psi_{\mu_t}} \right) \left\| \mathbf{v}_t - \mathbf{v}^*_{\mu_t}(\mathbf{x}_t) \right\|^2 + 2a_{t+1}\alpha_t r \left\| \frac{1}{m} \sum_{i=1}^{m} \mathbf{d}^{i,t}_{\mathbf{x}} - \bar{\mathbf{h}}^t_{\mathbf{x}} \right\|^2$$

$$+ 4\alpha_t^2 \left( \frac{6b_{t+1} L^2_{\psi_{\mathbf{y}1}}}{\beta_t \sigma^3_{\psi_{\mu_t}}} + \frac{6c_{t+1} \left( L_{\mathbf{v}1} \left\| \mathbf{v}^*_{\mu_t}(\mathbf{x}_{i,t}) \right\| + L_{\mathbf{v}2} \right)^2}{\eta_t \sigma^5_{\psi_{\mu_t}}} \right) \| \mathbf{h}^t_{\mathbf{x}} - \mathbf{1} \otimes \bar{\mathbf{h}}^t_{\mathbf{x}} \|^2$$

$$+ \frac{1}{m} \sum_{i=1}^{m} \frac{2a_{t+1} \left\| \nabla_{\mathbf{y}} f_i \left( \bar{\mathbf{x}}_{t+1}, \mathbf{y}^*_{\mu_{t+1}}(\bar{\mathbf{x}}_{t+1}) \right) \right\| \left\| \mathbf{y}^*_{\mu_{t+1}}(\mathbf{x}_{i,t+1}) \right\|}{\sigma_{h_i}} \left( \frac{\mu_t - \mu_{t+1}}{\mu_t} \right)$$

$$+ \frac{1}{m} \sum_{i=1}^{m} \frac{2a_{t+1} L_{f_{i\mathbf{y}2}} \| \mathbf{y}^*_{\mu_{t+1}}(\bar{\mathbf{x}}_{t+1}) \|^2}{\sigma^2_{h_i}} \left( \frac{\mu_t - \mu_{t+1}}{\mu_t} \right)^2$$

$$+ \sum_{i=1}^{m} \frac{24b_{t+1} \left\| \mathbf{y}^*_{\mu_{t+1}}(\mathbf{x}_{i,t+1}) \right\|^2}{\sigma^2_{h_i} \beta_t \sigma_{\psi_{\mu_t}}} \left( \frac{\mu_t - \mu_{t+1}}{\mu_t} \right)^2$$

$$+ \sum_{i=1}^{m} \frac{6c_{t+1}}{\eta_t \sigma_{\psi_{\mu_t}}} \left( \frac{\mu_t - \mu_{t+1}}{\mu_t^2} \right)^2 \left( \left( C_{\mathbf{v}1} \left\| \mathbf{v}^*_{\mu_{t+1}}(\mathbf{x}_{i,t+1}) \right\| + L_{f_{i\mathbf{y}2}} \right) \left\| \mathbf{y}^*_{\mu_{t+1}}(\mathbf{x}_{i,t+1}) \right\| \right.$$

$$\left. + C_{\mathbf{v}2} \left\| \nabla_{\mathbf{y}} f_i \left( \mathbf{x}_{i,t+1}, \mathbf{y}^*_{\mu_{t+1}}(\mathbf{x}_{i,t+1}) \right) \right\| \right)^2$$

$$+ d_{t+1} \left[ ((1+l_1)\lambda^2 - 1) \| \mathbf{x}_t - \mathbf{1} \otimes \bar{\mathbf{x}}_t \|^2 + (1 + \frac{1}{l_1})\alpha_t^2 \| \mathbf{h}^t_{\mathbf{x}} - \mathbf{1} \otimes \bar{\mathbf{h}}^t_{\mathbf{x}} \|^2 \right]$$

$$+ d_{t+1}\alpha_{t+1}((1+l_2)\lambda^2 - 1) \| \mathbf{h}^t_{\mathbf{x}} - \mathbf{1} \otimes \bar{\mathbf{h}}^t_{\mathbf{x}} \|^2$$

$$+ d_{t+1}\alpha_{t+1}(1 + \frac{1}{l_2})(L^2_{f_{i_1}} \| \mathbf{x}_{t+1} - \mathbf{x}_t \|^2 + L^2_{f_{i_2}} \beta_t^2 \| \mathbf{h}^t_{\mathbf{y}} \|^2).$$

For algorithm 1, with $\sum_{t=0}^{T} \left\| \frac{1}{m} \sum_{i=1}^{m} \mathbf{d}^{i,t}_{\mathbf{x}} - \bar{\mathbf{h}}^t_{\mathbf{x}} \right\|^2 = 0$, we have

$$V_{t+1} - V_t$$
$$\leq -\frac{a_{t+1}\alpha_t}{2} \| \nabla \Phi_{\mu_t}(\bar{\mathbf{x}}_t) \|^2$$

$$- \left( a_{t+1} \left( \frac{\alpha_t}{2} - \frac{\alpha_t^2 L_{\Phi_{\mu_t}}}{2\sigma^2_{\psi_{\mu_t}}} \right) - 4\alpha_t^2 m \frac{6b_{t+1} L^2_{\psi_{\mathbf{y}1}}}{\beta_t \sigma^3_{\psi_{\mu_t}}} - 4\alpha_t^2 m \frac{6c_{t+1} \left( L_{\mathbf{v}1} \left\| \mathbf{v}^*_{\mu_t}(\mathbf{x}_{i,t}) \right\| + L_{\mathbf{v}2} \right)^2}{\eta_t \sigma^5_{\psi_{\mu_t}}} \right.$$

$$\left. + d_{t+1}\alpha_{t+1} \left( 1 + \frac{1}{l_2} \right) L^2_{f_{i_1}} 4\alpha_t^2 m \right) \| \bar{\mathbf{h}}^t_{\mathbf{x}} \|^2$$

$$+ \left( a_{t+1} \frac{\alpha_t L_{\Phi_{\mu_t}}}{r\sigma^2_{\psi_{\mu_t}} m} + 8 \left( \frac{6b_{t+1} L^2_{\psi_{\mathbf{y}1}}}{\beta_t \sigma^3_{\psi_{\mu_t}}} + \frac{6c_{t+1} \left( L_{\mathbf{v}1} \left\| \mathbf{v}^*_{\mu_t}(\mathbf{x}_{i,t}) \right\| + L_{\mathbf{v}2} \right)^2}{\eta_t \sigma^5_{\psi_{\mu_t}}} \right) \right.$$

$$\left. + d_{t+1}((1+l_1)\lambda^2 - 1) + 8d_{t+1}\alpha_{t+1} \left( 1 + \frac{1}{l_2} \right) L^2_{f_{i_1}} \right) \| \mathbf{x}_t - \mathbf{1} \otimes \bar{\mathbf{x}}_t \|^2$$

$$+ \left( a_{t+1}\alpha_t r \frac{1}{m} \left( L_{\psi_{\mathbf{xy}2}} \left\| \mathbf{v}^*_{\mu_t}(\mathbf{x}_{i,t}) \right\| + L_{f_{i\mathbf{x}2}} \right)^2 \right.$$

$$\left. - \left( \frac{b_{t+1}}{2} \beta_t \sigma_{\psi_{\mu_t}} - \frac{3c_{t+1}\eta_t}{\sigma_{\psi_{\mu_t}}} \left( L_{\psi_{\mathbf{yy}2}} \left\| \mathbf{v}^*_{\mu_t}(\mathbf{x}_{i,t}) \right\| + L_{f_{i\mathbf{y}2}} \right)^2 \right) \right) \left\| \mathbf{y}_t - \mathbf{y}^*_{\mu_t}(\mathbf{x}_t) \right\|^2$$

$$+ \left( a_{t+1} \frac{\alpha_t r}{m} L^2_{\psi_{\mathbf{y}1}} - \frac{c_{t+1}}{2} \eta_t \sigma_{\psi_{\mu_t}} \right) \left\| \mathbf{v}_t - \mathbf{v}^*_{\mu_t}(\mathbf{x}_t) \right\|^2$$

$$+ \left( 4\alpha_t^2 \left( \frac{6b_{t+1} L^2_{\psi_{\mathbf{y}1}}}{\beta_t \sigma^3_{\psi_{\mu_t}}} + \frac{6c_{t+1} \left( L_{\mathbf{v}1} \left\| \mathbf{v}^*_{\mu_t}(\mathbf{x}_{i,t}) \right\| + L_{\mathbf{v}2} \right)^2}{\eta_t \sigma^5_{\psi_{\mu_t}}} \right) \right.$$

$$\left. + d_{t+1} \left( 1 + \frac{1}{l_1} \right) \alpha_t^2 + d_{t+1}\alpha_{t+1} \left( (1+l_2)\lambda^2 - 1 \right) \right.$$

$$+ d_{t+1}\alpha_{t+1}\left(1 + \frac{1}{l_2}\right)L_{f_{i_1}}^2 4\alpha_t^2\right)\|\mathbf{h}_{\mathbf{x}}^t - \mathbf{1}\otimes\bar{\mathbf{h}}_{\mathbf{x}}^t\|^2$$

$$+ \left(d_{t+1}\alpha_{t+1}\left(1 + \frac{1}{l_2}\right)L_{f_{i_2}}^2\beta_t^2 - b_{t+1}\left(1 + \frac{1}{2}\beta_t\sigma_{\psi_{\mu_t}}\right)\left(2\beta_t\frac{1}{\sigma_{\psi_{\mu_t}} + L_{\psi_{\mu_t}}} - \beta_t^2\right)\right)\|\mathbf{h}_{\mathbf{y}}^t\|^2$$

$$+ \frac{1}{m}\sum_{i=1}^{m}\frac{2a_{t+1}\left\|\nabla_{\mathbf{y}}f_i\left(\bar{\mathbf{x}}_{t+1}, \mathbf{y}_{\mu_{t+1}}^*(\bar{\mathbf{x}}_{t+1})\right)\right\|\left\|\mathbf{y}_{\mu_{t+1}}^*(\mathbf{x}_{i,t+1})\right\|}{\sigma_{h_i}}\left(\frac{\mu_t - \mu_{t+1}}{\mu_t}\right)$$

$$+ \frac{1}{m}\sum_{i=1}^{m}\frac{2a_{t+1}L_{f_{i\mathbf{y}2}}\|\mathbf{y}_{\mu_{t+1}}^*(\bar{\mathbf{x}}_{t+1})\|^2}{\sigma_{h_i}^2}\left(\frac{\mu_t - \mu_{t+1}}{\mu_t}\right)^2$$

$$+ \sum_{i=1}^{m}\frac{24b_{t+1}\left\|\mathbf{y}_{\mu_{t+1}}^*(\mathbf{x}_{i,t+1})\right\|^2}{\sigma_{h_i}^2\beta_t\sigma_{\psi_{\mu_t}}}\left(\frac{\mu_t - \mu_{t+1}}{\mu_t}\right)^2$$

$$+ \sum_{i=1}^{m}\frac{6c_{t+1}}{\eta_t\sigma_{\psi_{\mu_t}}}\left(\frac{\mu_t - \mu_{t+1}}{\mu_t^2}\right)^2\left(\left(C_{\mathbf{v}1}\left\|\mathbf{v}_{\mu_{t+1}}^*(\mathbf{x}_{i,t+1})\right\| + L_{f_{i\mathbf{y}2}}\right)\left\|\mathbf{y}_{\mu_{t+1}}^*(\mathbf{x}_{i,t+1})\right\|\right.$$

$$\left. + C_{\mathbf{v}2}\left\|\nabla_{\mathbf{y}}f_i\left(\mathbf{x}_{i,t+1}, \mathbf{y}_{\mu_{t+1}}^*(\mathbf{x}_{i,t+1})\right)\right\|\right)^2.$$

Then, it holds that

$$V_{t+1} - V_t$$

$$\leq -\frac{a_{t+1}\alpha_t}{2}\|\nabla\Phi_{\mu_t}(\bar{\mathbf{x}}_t)\|^2 + C_1\|\bar{\mathbf{h}}_{\mathbf{x}}^t\|^2 + C_2\|\mathbf{x}_t - \mathbf{1}\otimes\bar{\mathbf{x}}_t\|^2 + C_3\left\|\mathbf{y}_t - \mathbf{y}_{\mu_t}^*(\mathbf{x}_t)\right\|^2$$

$$+ C_4\left\|\mathbf{v}_t - \mathbf{v}_{\mu_t}^*(\mathbf{x}_t)\right\|^2 + C_5\|\mathbf{h}_{\mathbf{x}}^t - \mathbf{1}\otimes\bar{\mathbf{h}}_{\mathbf{x}}^t\|^2 + C_6\|\mathbf{h}_{\mathbf{y}}^{t-1}\|^2$$

$$+ \frac{1}{m}\sum_{i=1}^{m}\frac{2a_{t+1}\left\|\nabla_{\mathbf{y}}f_i\left(\bar{\mathbf{x}}_{t+1}, \mathbf{y}_{\mu_{t+1}}^*(\bar{\mathbf{x}}_{t+1})\right)\right\|\left\|\mathbf{y}_{\mu_{t+1}}^*(\mathbf{x}_{i,t+1})\right\|}{\sigma_{h_i}}\left(\frac{\mu_t - \mu_{t+1}}{\mu_t}\right)$$

$$+ \frac{1}{m}\sum_{i=1}^{m}\frac{2a_{t+1}L_{f_{i\mathbf{y}2}}\|\mathbf{y}_{\mu_{t+1}}^*(\bar{\mathbf{x}}_{t+1})\|^2}{\sigma_{h_i}^2}\left(\frac{\mu_t - \mu_{t+1}}{\mu_t}\right)^2$$

$$+ \sum_{i=1}^{m}\frac{24b_{t+1}\left\|\mathbf{y}_{\mu_{t+1}}^*(\mathbf{x}_{i,t+1})\right\|^2}{\sigma_{h_i}^2\beta_t\sigma_{\psi_{\mu_t}}}\left(\frac{\mu_t - \mu_{t+1}}{\mu_t}\right)^2$$

$$+ \sum_{i=1}^{m}\frac{6c_{t+1}}{\eta_t\sigma_{\psi_{\mu_t}}}\left(\frac{\mu_t - \mu_{t+1}}{\mu_t^2}\right)^2\left(\left(C_{\mathbf{v}1}\left\|\mathbf{v}_{\mu_{t+1}}^*(\mathbf{x}_{i,t+1})\right\| + L_{f_{i\mathbf{y}2}}\right)\left\|\mathbf{y}_{\mu_{t+1}}^*(\mathbf{x}_{i,t+1})\right\|\right.$$

$$\left. + C_{\mathbf{v}2}\left\|\nabla_{\mathbf{y}}f_i\left(\mathbf{x}_{i,t+1}, \mathbf{y}_{\mu_{t+1}}^*(\mathbf{x}_{i,t+1})\right)\right\|\right)^2,$$

where

$$C_1 = -a_{t+1}\left(\frac{\alpha_t}{2} - \frac{\alpha_t^2 L_{\Phi\mu_t}}{2\sigma_{\psi_{\mu_t}}^2}\right) + 4\alpha_t^2 m\frac{6b_{t+1}L_{\psi\mathbf{y}1}^2}{\beta_t\sigma_{\psi_{\mu_t}}^3} + 4\alpha_t^2 m\frac{6c_{t+1}\left(L_{\mathbf{v}1}\left\|\mathbf{v}_{\mu_t}^*(\mathbf{x}_{i,t})\right\| + L_{\mathbf{v}2}\right)^2}{\eta_t\sigma_{\psi_{\mu_t}}^5}$$

$$+ d_{t+1}\alpha_{t+1}\left(1 + \frac{1}{l_2}\right)L_{f_{i_1}}^2 4\alpha_t^2 m,$$

$$C_2 = a_{t+1}\frac{\alpha_t L_{\Phi\mu_t}}{r\sigma_{\psi_{\mu_t}}^2 m} + 8\left(\frac{6b_{t+1}L_{\psi\mathbf{y}1}^2}{\beta_t\sigma_{\psi_{\mu_t}}^3} + \frac{6c_{t+1}\left(L_{\mathbf{v}1}\left\|\mathbf{v}_{\mu_t}^*(\mathbf{x}_{i,t})\right\| + L_{\mathbf{v}2}\right)^2}{\eta_t\sigma_{\psi_{\mu_t}}^5}\right)$$

$$+ d_{t+1}\left((1 + l_1)\lambda^2 - 1\right) + 8d_{t+1}\alpha_{t+1}\left(1 + \frac{1}{l_2}\right)L_{f_{i_1}}^2,$$

$$C_3 = a_{t+1}\alpha_t r\frac{1}{m}\left(L_{\psi\mathbf{x}\mathbf{y}2}\left\|\mathbf{v}_{\mu_t}^*(\mathbf{x}_{i,t})\right\| + L_{f_{i\mathbf{x}2}}\right)^2$$

$$- \left(\frac{b_{t+1}}{2}\beta_t\sigma_{\psi_{\mu_t}} - \frac{3c_{t+1}\eta_t}{\sigma_{\psi_{\mu_t}}}\left(L_{\psi\mathbf{y}\mathbf{y}2}\left\|\mathbf{v}_{\mu_t}^*(\mathbf{x}_{i,t})\right\| + L_{f_{i\mathbf{y}2}}\right)^2\right),$$

$$C_4 = a_{t+1}\frac{\alpha_t r}{m}L_{\psi_{\mathbf{y1}}}^2 - \frac{c_{t+1}}{2}\eta_t\sigma_{\psi_{\mu_t}},$$

$$C_5 = 4\alpha_t^2\left(\frac{6b_{t+1}L_{\psi_{\mathbf{y1}}}^2}{\beta_t\sigma_{\psi_{\mu_t}}^3} + \frac{6c_{t+1}\left(L_{\mathbf{v1}}\left\|\mathbf{v}_{\mu_t}^*\left(\mathbf{x}_{i,t}\right)\right\| + L_{\mathbf{v2}}\right)^2}{\eta_t\sigma_{\psi_{\mu_t}}^5}\right)$$

$$+ d_{t+1}\left(1 + \frac{1}{l_1}\right)\alpha_t^2 + d_{t+1}\alpha_{t+1}\left((1 + l_2)\lambda^2 - 1\right) + d_{t+1}\alpha_{t+1}\left(1 + \frac{1}{l_2}\right)L_{f_{i_1}}^2 4\alpha_t^2,$$

$$C_6 = d_{t+1}\alpha_{t+1}\left(1 + \frac{1}{l_2}\right)L_{f_{i_2}}^2\beta_t^2 - b_{t+1}\left(1 + \frac{1}{2}\beta_t\sigma_{\psi_{\mu_t}}\right)\left(2\beta_t\frac{1}{\sigma_{\psi_{\mu_t}} + L_{\psi_{\mu_t}}} - \beta_t^2\right).$$

$\square$

### Step 5: Step-size calculations.

To ensure convergence, we choose step-sizes carefully by setting $l_1 = l_2 = \frac{1}{\lambda} - 1$, which simplifies several terms. Under this setting, we have $1 + \frac{1}{l_1} = 1 + \frac{1}{l_2} = \frac{1}{1-\lambda}$ and $(1 + l_1)\lambda^2 - 1 = \lambda - 1$. Using these values, we proceed to calculate each of the constants $C_1$ through $C_6$, which represent various components that contribute to the change in the Lyapunov function.

$$C_1 = -a_{t+1}\left(\frac{\alpha_t}{2} - \frac{\alpha_t^2 L_{\Phi\mu_t}}{2\sigma_{\psi_{\mu_t}}^2}\right) + 4\alpha_t^2 m\frac{6b_{t+1}L_{\psi_{\mathbf{y1}}}^2}{\beta_t\sigma_{\psi_{\mu_t}}^3} + 4\alpha_t^2 m\frac{6c_{t+1}\left(L_{\mathbf{v1}}\left\|\mathbf{v}_{\mu_t}^*\left(\mathbf{x}_{i,t}\right)\right\| + L_{\mathbf{v2}}\right)^2}{\eta_t\sigma_{\psi_{\mu_t}}^5}$$

$$+ \frac{1}{1-\lambda}d_{t+1}\alpha_{t+1}L_{f_{i_1}}^2 4\alpha_t^2 m,$$

$$C_2 = a_{t+1}\frac{\alpha_t L_{\Phi\mu_t}}{r\sigma_{\psi_{\mu_t}}^2 m} + 8\left(\frac{6b_{t+1}L_{\psi_{\mathbf{y1}}}^2}{\beta_t\sigma_{\psi_{\mu_t}}^3} + \frac{6c_{t+1}\left(L_{\mathbf{v1}}\left\|\mathbf{v}_{\mu_t}^*\left(\mathbf{x}_{i,t}\right)\right\| + L_{\mathbf{v2}}\right)^2}{\eta_t\sigma_{\psi_{\mu_t}}^5}\right)$$

$$+ d_{t+1}\left(\lambda - 1\right) + 8\frac{1}{1-\lambda}d_{t+1}\alpha_{t+1}L_{f_{i_1}}^2,$$

$$C_3 = a_{t+1}\alpha_t r\frac{1}{m}\left(L_{\psi_{\mathbf{xy2}}}\left\|\mathbf{v}_{\mu_t}^*\left(\mathbf{x}_{i,t}\right)\right\| + L_{f_{i_{\mathbf{x2}}}}\right)^2$$

$$- \left(\frac{b_{t+1}}{2}\beta_t\sigma_{\psi_{\mu_t}} - \frac{3c_{t+1}\eta_t}{\sigma_{\psi_{\mu_t}}}\left(L_{\psi_{\mathbf{yy2}}}\left\|\mathbf{v}_{\mu_t}^*\left(\mathbf{x}_{i,t}\right)\right\| + L_{f_{i_{\mathbf{y2}}}}\right)^2\right),$$

$$C_4 = a_{t+1}\frac{\alpha_t r}{m}L_{\psi_{\mathbf{y1}}}^2 - \frac{c_{t+1}}{2}\eta_t\sigma_{\psi_{\mu_t}},$$

$$C_5 = 4\alpha_t^2\left(\frac{6b_{t+1}L_{\psi_{\mathbf{y1}}}^2}{\beta_t\sigma_{\psi_{\mu_t}}^3} + \frac{6c_{t+1}\left(L_{\mathbf{v1}}\left\|\mathbf{v}_{\mu_t}^*\left(\mathbf{x}_{i,t}\right)\right\| + L_{\mathbf{v2}}\right)^2}{\eta_t\sigma_{\psi_{\mu_t}}^5}\right)$$

$$+ \frac{1}{1-\lambda}d_{t+1}\alpha_t^2 + d_{t+1}\alpha_{t+1}\left(\lambda - 1\right) + \frac{1}{1-\lambda}d_{t+1}\alpha_{t+1}L_{f_{i_1}}^2 4\alpha_t^2,$$

$$C_6 = d_{t+1}\alpha_{t+1}\frac{1}{1-\lambda}L_{f_{i_2}}^2\beta_t^2 - b_{t+1}\left(1 + \frac{1}{2}\beta_t\sigma_{\psi_{\mu_t}}\right)\left(2\beta_t\frac{1}{\sigma_{\psi_{\mu_t}} + L_{\psi_{\mu_t}}} - \beta_t^2\right).$$

Next, we set the unified form as follow:

$$\beta_t \in \left[\bar{\beta}(t+1)^{-\tau/4}, \frac{1}{L_{f_{i_{y2}}} + L_{g_{y2}}}\right], \eta_t = c_\eta\frac{b_{t+1}}{c_{t+1}}\beta_t\sigma_{\psi_{\mu_t}}^2, \alpha_t = c_\alpha\frac{a_{t+1}}{c_{t+1}}\eta_t\sigma_{\psi_{\mu_t}}^5,$$

where

$$c_\eta = \left(\frac{c_{t+1}}{b_{t+1}}\right)\left(\frac{\eta_t}{\beta_t}\right)\sigma_{\psi_{\mu_t}}^{-2}, \qquad c_\alpha = \left(\frac{c_{t+1}}{a_{t+1}}\right)\left(\frac{\alpha_t}{\eta_t}\right)\sigma_{\psi_{\mu_t}}^{-5}.$$

Also, we set

$$\eta_t = (t+1)^{-\tau/2}\beta_t, \qquad \alpha_t = (t+1)^{-3\tau/2}\beta_t\mu_t^3,$$

$$a_{t+1} = (t+1)^{-\tau}, \qquad b_{t+1} = (t+1)^{-\tau/2}\sigma_{\psi_{\mu_t}}^3,$$

$$c_{t+1} = (t+1)^{-\tau}\sigma_{\psi_{\mu_t}}^5, \qquad d_{t+1} = (t+1)^{-\tau/8},$$

$$c_\eta = (t+1)^{-\tau}, \qquad c_\alpha = (t+1)^{-\tau}\sigma^3_{\psi_{\mu_t}}.$$

For $C_1$, we have

$$C_1 = -a_{t+1}\left(\frac{\alpha_t}{2} - \frac{\alpha_t^2 L_{\Phi\mu_t}}{2\sigma^2_{\psi_{\mu_t}}}\right) + 4\alpha_t^2 m \frac{6b_{t+1}L^2_{\psi\mathbf{y}1}}{\beta_t\sigma^3_{\psi_{\mu_t}}}$$

$$+ 4\alpha_t^2 m \frac{6c_{t+1}\left(L_{\mathbf{v}1}\left\|\mathbf{v}^*_{\mu_t}(\mathbf{x}_{i,t})\right\| + L_{\mathbf{v}2}\right)^2}{\eta_t\sigma^5_{\psi_{\mu_t}}} + \frac{1}{1-\lambda}d_{t+1}\alpha_{t+1}L^2_{f_{i_1}}4\alpha_t^2 m$$

$$= -\frac{a_{t+1}\alpha_t}{2} + \frac{a_{t+1}\alpha_t^2 L_{\Phi\mu_t}}{2\sigma^2_{\psi_{\mu_t}}} + \frac{1}{1-\lambda}d_{t+1}\alpha_{t+1}L^2_{f_{i_1}}4\alpha_t^2 m$$

$$- 4\alpha_t^2 m \frac{a_{t+1}}{\alpha_t}c_\alpha\left[\frac{6b_{t+1}\alpha_t L^2_{\psi\mathbf{y}1}}{c_\alpha a_{t+1}\beta_t\sigma^3_{\psi_{\mu_t}}} + 6c_{t+1}\left(L_{\mathbf{v}1}\left\|\mathbf{v}^*_{\mu_t}(\mathbf{x}_{i,t})\right\| + L_{\mathbf{v}2}\right)^2\right].$$

By further simplification and applying our chosen step-size settings, we derive:

$$C_1 = -\frac{1}{2}(t+1)^{-5\tau/2}\beta_t\mu_t^3 + 24m(t+1)^{-7/2\tau}\beta_t\mu_t^6\left(L_{\psi\mathbf{y}1} + \left(L_{\mathbf{v}1}\left\|\mathbf{v}^*_{\mu_t}(\mathbf{x}_{i,t})\right\| + L_{\mathbf{v}2}\right)^2\right)$$

$$+ (t+1)^{-4\tau}\beta_t^2\mu_t^6\frac{L_{\Phi\mu_t}}{2\sigma^2_{\psi_{\mu_t}}} + 4m\frac{L^2_{f_{i_1}}}{1-\lambda}(t+1)^{-37\tau/8}\beta_t^3\mu_t^9.$$

As we can see, the dominant term for $C_1$ is $-(t+1)^{-5\tau/2}\beta_t\mu_t^3$, which is smaller than 0. This indicates that $C_1$ contributes to the overall decrease in the Lyapunov function.

Next, for $C_2$, we have

$$C_2 = a_{t+1}\frac{\alpha_t L_{\Phi\mu_t}}{r\sigma^2_{\psi_{\mu_t}}m} + 8\left(\frac{6b_{t+1}L^2_{\psi\mathbf{y}1}}{\beta_t\sigma^3_{\psi_{\mu_t}}} + \frac{6c_{t+1}\left(L_{\mathbf{v}1}\left\|\mathbf{v}^*_{\mu_t}(\mathbf{x}_{i,t})\right\| + L_{\mathbf{v}2}\right)^2}{\eta_t\sigma^5_{\psi_{\mu_t}}}\right)$$

$$+ d_{t+1}(\lambda - 1) + 8\frac{1}{1-\lambda}d_{t+1}\alpha_{t+1}L^2_{f_{i_1}}$$

$$= a_{t+1}\frac{\alpha_t L_{\Phi\mu_t}}{r\sigma^2_{\psi_{\mu_t}}m} + 8\frac{a_{t+1}}{\alpha_t}c_\alpha\left[\frac{6b_{t+1}\alpha_t L^2_{\psi\mathbf{y}1}}{c_\alpha a_{t+1}\beta_t\sigma^3_{\psi_{\mu_t}}} + 6\left(L_{\mathbf{v}1}\left\|\mathbf{v}^*_{\mu_t}(\mathbf{x}_{i,t})\right\| + L_{\mathbf{v}2}\right)^2\right]$$

$$+ d_{t+1}(\lambda - 1) + 8\frac{1}{1-\lambda}d_{t+1}\alpha_{t+1}L^2_{f_{i_1}}.$$

Simplifying further, we obtain:

$$C_2 = (t+1)^{-\tau/8}(\lambda - 1) + \frac{L_{\psi\mathbf{y}1}}{rm\sigma^2_{\psi_{\mu_t}}}(t+1)^{-5\tau/2}\mu_t^3 + \frac{8L^2_{f_{i_1}}}{1-\lambda}(t+1)^{-13\tau/8}\beta_t\mu_t^3$$

$$+ \frac{1}{\beta_t}(t+1)^{-\tau/2}\left(48 + 48\left(L_{\mathbf{v}1}\left\|\mathbf{v}^*_{\mu_t}(\mathbf{x}_{i,t})\right\| + L_{\mathbf{v}2}\right)^2\right).$$

The leading term is $-(t+1)^{-\tau/8}$, ensuring that $C_2$ is negative.

For $C_3$, we have

$$C_3 = a_{t+1}\alpha_t r\frac{1}{m}\left(L_{\psi\mathbf{xy}2}\left\|\mathbf{v}^*_{\mu_t}(\mathbf{x}_{i,t})\right\| + L_{f_{i\mathbf{x}2}}\right)^2$$

$$- \left(\frac{b_{t+1}}{2}\beta_t\sigma_{\psi_{\mu_t}} - \frac{3c_{t+1}\eta_t}{\sigma_{\psi_{\mu_t}}}\left(L_{\psi\mathbf{yy}2}\left\|\mathbf{v}^*_{\mu_t}(\mathbf{x}_{i,t})\right\| + L_{f_{i\mathbf{y}2}}\right)^2\right)$$

$$= -\frac{b_{t+1}}{2}\beta_t\sigma_{\psi_{\mu_t}} + \frac{3c_{t+1}\eta_t}{\sigma_{\psi_{\mu_t}}}\left(L_{\psi\mathbf{yy}2}\left\|\mathbf{v}^*_{\mu_t}(\mathbf{x}_{i,t})\right\| + L_{f_{i\mathbf{y}2}}\right)^2$$

$$+ a_{t+1}\alpha_t r\frac{1}{m}\left(L_{\psi\mathbf{xy}2}\left\|\mathbf{v}^*_{\mu_t}(\mathbf{x}_{i,t})\right\| + L_{f_{i\mathbf{x}2}}\right)^2$$

$$= -\left(1 - \frac{6c_{t+1}\eta_t}{b_{t+1}\beta_t\sigma^2_{\psi_{\mu_t}}}\right)\left(L_{\psi\mathbf{yy}2}\left\|\mathbf{v}^*_{\mu_t}(\mathbf{x}_{i,t})\right\| + L_{f_{i\mathbf{y}2}}\right)^2$$

$$-\frac{2a_{t+1}\alpha_t r}{b_{t+1}\beta_t \sigma_{\psi_{\mu_t}} m}\left(L_{\psi_{\mathbf{xy2}}}\left\|\mathbf{v}^*_{\mu_t}\left(\mathbf{x}_{i,t}\right)\right\|+L_{f_{i\mathbf{x2}}}\right)^2\right)\times\frac{b_{t+1}}{2}\beta_t\sigma_{\psi_{\mu_t}}$$

$$=-\left(1-c_\eta\left(\frac{6c_{t+1}\eta_t}{c_\eta b_{t+1}\beta_t\sigma^2_{\psi_{\mu_t}}}\left(L_{\psi_{\mathbf{yy2}}}\left\|\mathbf{v}^*_{\mu_t}\left(\mathbf{x}_{i,t}\right)\right\|+L_{f_{i\mathbf{y2}}}\right)^2\right.\right.$$

$$\left.\left.+\frac{2a_{t+1}\alpha_t r}{c_\eta b_{t+1}\beta_t\sigma_{\psi_{\mu_t}}m}\left(L_{\psi_{\mathbf{xy2}}}\left\|\mathbf{v}^*_{\mu_t}\left(\mathbf{x}_{i,t}\right)\right\|+L_{f_{i\mathbf{x2}}}\right)^2\right)\right)$$

$$\times\frac{b_{t+1}}{2}\beta_t\sigma_{\psi_{\mu_t}}.$$

After simplification, we have:

$$C_3=\frac{1}{2}(t+1)^{-\tau/2}\beta_t\sigma^4_{\psi_{\mu_t}}$$

$$\times\left(-1+(t+1)^{-\tau}\left(6\left(L_{\psi_{\mathbf{yy2}}}\left\|\mathbf{v}^*_{\mu_t}\left(\mathbf{x}_{i,t}\right)\right\|+L_{f_{i\mathbf{y2}}}\right)^2\right.\right.$$

$$\left.\left.+\frac{2r}{\sigma_{\psi^4_{\mu_t}}m}(t+1)^{-\tau}\mu^3_t\left(L_{\psi_{\mathbf{xy2}}}\left\|\mathbf{v}^*_{\mu_t}\left(\mathbf{x}_{i,t}\right)\right\|+L_{f_{i\mathbf{x2}}}\right)^2\right)\right),$$

which is negative as well, with the dominant term $-(t+1)^{-\tau/2}\beta_t$.

For $C_4$, the calculation proceeds similarly:

$$C_4=a_{t+1}\frac{\alpha_t r}{m}L^2_{\psi_{\mathbf{y1}}}-\frac{c_{t+1}}{2}\eta_t\sigma_{\psi_{\mu_t}}=-\left(1-\frac{2a_{t+1}\alpha_t L^2_{\psi_{\mathbf{y1}}}r}{c_{t+1}\eta_t\sigma_{\psi_{\mu_t}}m}\right)\frac{c_{t+1}}{2}\eta_t\sigma_{\psi_{\mu_t}}.$$

Then

$$C_4=-\frac{\sigma^5_{\psi_{\mu_t}}}{2}\left(1-(t+1)^{-\tau}\mu^{-3}_t\right)(t+1)^{-3\tau/2}\beta_t,$$

ensuring $C_4$ is negative with its dominant term being $-(t+1)^{-3\tau/2}\beta_t$.

For $C_5$, we have

$$C_5=4\alpha^2_t\left(\frac{6b_{t+1}L^2_{\psi_{\mathbf{y1}}}}{\beta_t\sigma^3_{\psi_{\mu_t}}}+\frac{6c_{t+1}\left(L_{\mathbf{v1}}\left\|\mathbf{v}^*_{\mu_t}\left(\mathbf{x}_{i,t}\right)\right\|+L_{\mathbf{v2}}\right)^2}{\eta_t\sigma^5_{\psi_{\mu_t}}}\right)$$

$$+\frac{1}{1-\lambda}d_{t+1}\alpha^2_t+d_{t+1}\alpha_{t+1}\left(\lambda-1\right)+\frac{1}{1-\lambda}d_{t+1}\alpha_{t+1}L^2_{f_{i_1}}4\alpha^2_t$$

$$=4\alpha^2_t\frac{a_{t+1}}{\alpha_t}c_\alpha\left[\frac{6b_{t+1}\alpha_t L^2_{\psi_{\mathbf{y1}}}}{c_\alpha a_{t+1}\beta_t\sigma^3_{\psi_{\mu_t}}}+6\left(L_{\mathbf{v1}}\left\|\mathbf{v}^*_{\mu_t}\left(\mathbf{x}_{i,t}\right)\right\|+L_{\mathbf{v2}}\right)^2\right]$$

$$+\frac{1}{1-\lambda}d_{t+1}\alpha^2_t+d_{t+1}\alpha_{t+1}\left(\lambda-1\right)+\frac{1}{1-\lambda}d_{t+1}\alpha_{t+1}L^2_{f_{i_1}}4\alpha^2_t.$$

Then we have

$$C_5=24(t+1)^{-7\tau/2}\beta_t\mu^6_t\left[L^2_{\psi_{\mathbf{y1}}}+\left(L_{\mathbf{v1}}\left\|\mathbf{v}^*_{\mu_t}\left(\mathbf{x}_{i,t}\right)\right\|+L_{\mathbf{v2}}\right)^2\right]$$

$$+\frac{1}{1-\lambda}(t+1)^{-25\tau/8}\beta^2_t\mu^6_t+(\lambda-1)(t+1)^{-13\tau/8}\beta_t\mu^3_t+\frac{4}{1-\lambda}L^2_{f_{i_1}}(t+1)^{-37\tau/8}\beta^3_t\mu^9_t.$$

As we can see, $C_5$ should be the same order as the order of $-(t+1)^{-13\tau/8}\beta_t\mu^3_t$, which is smaller than 0.

Finally, for $C_6$, we have

$$C_6=\frac{1}{1-\lambda}d_{t+1}\alpha_{t+1}L^2_{f_{i_2}}\beta^2_t-b_{t+1}\left(1+\frac{1}{2}\beta_t\sigma_{\psi_{\mu_t}}\right)\left(2\beta_t\frac{1}{\sigma_{\psi_{\mu_t}}+L_{\psi_{\mu_t}}}-\beta^2_t\right)$$

$$=\frac{1}{1-\lambda}L^2_{f_{i_2}}d_{t+1}\alpha_{t+1}\beta^2_t-2\beta_t\frac{b_{t+1}}{\sigma_{\psi_{\mu_t}}+L_{\psi_{\mu_t}}}+b_{t+1}\beta^2_t\frac{\sigma_{\psi_{\mu_t}}}{\sigma_{\psi_{\mu_t}}+L_{\psi_{\mu_t}}}+\frac{1}{2}b_{t+1}\beta^3_t\sigma_{\psi_{\mu_t}}.$$

Then we can get

$$C_6 = -2\beta_t \frac{(t+1)^{-\tau/2}\sigma_{\psi_{\mu_t}}^3}{\sigma_{\psi_{\mu_t}} + L_{\psi_{\mu_t}}} + \frac{1}{1-\lambda} L_{f_{i_2}}^2 (t+1)^{-25\tau/8}\beta_t^3\mu_t^3 + (t+1)^{-\tau/2}\frac{\sigma_{\psi_{\mu_t}}^4}{\sigma_{\psi_{\mu_t}} + L_{\psi_{\mu_t}}}\beta_t^2$$

$$+ \frac{1}{2}(t+1)^{-\tau/2}\beta_t^3\sigma_{\psi_{\mu_t}}^4.$$

As we can see, $C_6$ should be the same order as the order of $-(t+1)^{-\tau/2}\beta_t$, which is smaller than 0.

**Step 6: Proof of Theorem.**

It is helpful to find suitable coeffcients $a_{t+1}, b_{t+1}, c_{t+1}, d_{t+1}$ such that $C_1$ to $C_6 < 0$ and all of the sequences involving $\mu_t - \mu_{t+1}$ (denoted by $A_\mu^t, B_\mu^t, C_\mu^t$ respectively) are summable. Achieving this would allow us to conclude convergence using Lemma 10. First, we establish the following inequality for the change in the potential function $V_t$:

$$V_{t+1} - V_t \le A_\mu^t + B_\mu^t + C_\mu^t.$$

By the definition of potential function, we get $V_t \ge 0$ for all $t$. This implies that the potential function is bounded from below, and we can write:

$$V_T \le V_0 + \sum_{t=1}^{\infty} \left(A_\mu^t + B_\mu^t + C_\mu^t\right).$$

Since the sequences $A_\mu^t$, $B_\mu^t$, and $C_\mu^t$ are summable, we can conclude that:

$$V_T = O(1) \quad \text{as} \quad T \to \infty.$$

Next, applying Lemma 10, we have the following summation over the interval $t = T$ to $t = 2T + 1$:

$$\sum_{t=T}^{2T+1} \left( \frac{a_{t+1}\alpha_t}{2}\|\nabla\Phi_{\mu_t}(\bar{\mathbf{x}}_t)\|^2 - C_1\|\bar{\mathbf{h}}_{\mathbf{x}}^t\|^2 - C_2\|\mathbf{x}_t - \mathbf{1}\otimes\bar{\mathbf{x}}_t\|^2 - C_3\left\|\mathbf{y}_t - \mathbf{y}_{\mu_t}^*(\mathbf{x}_t)\right\|^2 \right.$$

$$\left. -C_4\left\|\mathbf{v}_t - \mathbf{v}_{\mu_t}^*(\mathbf{x}_t)\right\|^2 - C_5\|\mathbf{h}_{\mathbf{x}}^t - \mathbf{1}\otimes\bar{\mathbf{h}}_{\mathbf{x}}^t\|^2 - C_6\|\mathbf{h}_{\mathbf{y}}^t\|^2\right) = O(1).$$

Since the constants $C_1$ through $C_6$ are all strictly negative, the terms on the left-hand side are non-negative, and thus, the overall sum remains bounded. Note that $\sum_{t=T}^{2T+1}(t+1)^{-s} \ge \int_{T+1}^{2T+2} k^{-s}dk = \frac{2^{1-s}-1}{1-s}(T+1)^{1-s}$.

Now, by lower-bounding the coefficients, according to different step size strategies and the coefficients of Lyapunov function, we can get the estimates on

$$\min_{0\le t\le T} \left( \|\nabla\Phi_{\mu_t}(\bar{\mathbf{x}}_t)\|^2 + \|\bar{\mathbf{h}}_{\mathbf{x}}^t\|^2 + \|\mathbf{x}_t - \mathbf{1}\otimes\bar{\mathbf{x}}_t\|^2 + \left\|\mathbf{y}_t - \mathbf{y}_{\mu_t}^*(\mathbf{x}_t)\right\|^2 \right.$$

$$\left. + \left\|\mathbf{v}_t - \mathbf{v}_{\mu_t}^*(\mathbf{x}_t)\right\|^2 + \|\mathbf{h}_{\mathbf{x}}^t - \mathbf{1}\otimes\bar{\mathbf{h}}_{\mathbf{x}}^t\|^2 + \|\mathbf{h}_{\mathbf{y}}^t\|^2 \right).$$

By choosing suitable upper bounds for the parameters $p$ and $\tau$, we conclude that the desired convergence holds, completing the proof. To further lead to the convergence rates as measured by the KKT residual, due to the optimality of $\mathbf{y}_{\mu_t}^*(\mathbf{x}_t)$ and the definition of $\mathbf{v}_{\mu_t}^*(\mathbf{x}_t)$, there exists a positive constant $C$, independent of $t$, such that:

$$\text{KKT}(\mathbf{x}_t, \mathbf{y}_t, \mathbf{v}_t)$$

$$\le C\left(\|\nabla\Phi_{\mu_t}(\bar{\mathbf{x}}_t)\|^2 + \|\mathbf{x}_t - \mathbf{1}\otimes\bar{\mathbf{x}}_t\|^2 + \left\|\mathbf{y}_t - \mathbf{y}_{\mu_t}^*(\mathbf{x}_t)\right\|^2 + \left\|\mathbf{v}_t - \mathbf{v}_{\mu_t}^*(\mathbf{x}_t)\right\|^2 + \mu_t^2\right).$$

## D SUPPORTING LEMMAS

Now, we present the technical lemmas that would be useful throughout our main result.

To establish the convergence result of DUET, we first characterize the Lipschitz continuity of $\mathbf{y}_\mu^*(\mathbf{x}_i)$ in $\mathbf{x}_i$ and $\mathbf{v}_\mu^*(\mathbf{x}_i)$ in $\mathbf{x}_i$, without any boundedness assumption of $\nabla_{\mathbf{y}} f_i(\mathbf{x}_i, \mathbf{y}_i)$.

**Lemma 11.** *Suppose assumptions holds, the follow statement holds.*

*(a) For any $i \in [m]$, the function $\mathbf{y}_\mu^*(\mathbf{x}_i)$ is Lipschitz continuous in $\mathbf{x}_i$, i.e, for all $\mathbf{x}_i, \mathbf{x}_i'$,*

$$\|\mathbf{y}_\mu^*(\mathbf{x}_i) - \mathbf{y}_\mu^*(\mathbf{x}_i')\| \leq \frac{L_{\psi_{\mathbf{y}1}}}{\sigma_{\psi_\mu}} \|\mathbf{x}_i - \mathbf{x}_i'\|,$$

*where both $L_{\psi_{\mathbf{y}1}} = \mu L_{h_{i\mathbf{y}1}} + (1-\mu)L_{g_{i\mathbf{y}1}}$ and $\sigma_{\psi_\mu} = \mu\sigma_{h_i} + (1-\mu)\sigma_{g_i}$ are constants.*

*(b) For any $i \in [m]$, the function $\mathbf{v}_\mu^*(\mathbf{x}_i)$ satisfies*

$$\|\mathbf{v}_\mu^*(\mathbf{x}_i)\| = \left\|\left[\nabla_{\mathbf{yy}}^2 \psi_\mu\left(\mathbf{x}_i, \mathbf{y}_\mu^*(\mathbf{x}_i)\right)\right]^{-1} \nabla_{\mathbf{y}} f_i\left(\mathbf{x}_i, \mathbf{y}_\mu^*(\mathbf{x}_i)\right)\right\| \leq \frac{\left\|\nabla_{\mathbf{y}} f_i\left(\mathbf{x}_i, \mathbf{y}_\mu^*(\mathbf{x}_i)\right)\right\|}{\sigma_{\psi_\mu}},$$

*and for all $\mathbf{x}_i, \mathbf{x}_i'$,*

$$\left\|\mathbf{v}_\mu^*(\mathbf{x}_i) - \mathbf{v}_\mu^*(\mathbf{x}_i')\right\| \leq \frac{\left(L_{\mathbf{v}1}\|\mathbf{v}_\mu^*(\mathbf{x}_i)\| + L_{\mathbf{v}2}\right)}{\sigma_{\psi_\mu}^2} \|\mathbf{x}_i - \mathbf{x}_i'\|,$$

*where both $L_{\mathbf{v}1} := L_{\psi_{\mathbf{yy}2}} L_{\psi_{\mathbf{y}1}} + L_{\psi_{\mathbf{yy}1}}\sigma_{\psi_\mu}$ and $L_{\mathbf{v}2} := L_{f_{i\mathbf{y}2}} L_{\psi_{\mathbf{y}1}} + L_{f_{i\mathbf{y}1}}\sigma_{\psi_\mu}$ are constants.*

*Proof.* (a) By the optimality of $\mathbf{y}_{i,\mu}^*$ to LL problem, we have $\nabla_{\mathbf{y}} \psi_\mu(\mathbf{x}_i, \mathbf{y}_\mu^*(\mathbf{x}_i)) = 0$. Thus

$$\left(\nabla_{\mathbf{y}} \psi_\mu(\mathbf{x}_i, \mathbf{y}_\mu^*(\mathbf{x}_i)) - \nabla_{\mathbf{y}} \psi_\mu(\mathbf{x}_i, \mathbf{y}_\mu^*(\mathbf{x}_i'))\right)$$
$$+ \left(\nabla_{\mathbf{y}} \psi_\mu(\mathbf{x}_i, \mathbf{y}_\mu^*(\mathbf{x}_i')) - \nabla_{\mathbf{y}} \psi_\mu(\mathbf{x}_i', \mathbf{y}_\mu^*(\mathbf{x}_i'))\right) = 0.$$

Multiplying the above equation by $\mathbf{y}_\mu^*(\mathbf{x}_i) - \mathbf{y}_\mu^*(\mathbf{x}_i')$, by the $\sigma_{\psi_\mu}$- strongly convexity of $\psi_\mu(\mathbf{x}, \cdot)$ and $L_{\psi_{\mathbf{y}1}}$-smoothness of $\psi_\mu(\cdot, \mathbf{y})$, we have

$$\sigma_{\psi_\mu}\|\mathbf{y}_\mu^*(\mathbf{x}_i) - \mathbf{y}_\mu^*(\mathbf{x}_i')\|^2$$
$$\leq \|\nabla_{\mathbf{y}}\psi_\mu(\mathbf{x}_i, \mathbf{y}_\mu^*(\mathbf{x}_i')) - \nabla_{\mathbf{y}}\psi_\mu(\mathbf{x}_i', \mathbf{y}_\mu^*(\mathbf{x}_i'))\| \cdot \|\mathbf{y}_\mu^*(\mathbf{x}_i) - \mathbf{y}_\mu^*(\mathbf{x}_i')\|$$
$$\leq L_{\psi_{\mathbf{y}1}}\|\mathbf{x}_i - \mathbf{x}_i'\| \cdot \|\mathbf{y}_\mu^*(\mathbf{x}_i) - \mathbf{y}_\mu^*(\mathbf{x}_i')\|.$$

Then the conclusion follows immediately from the above inequality.
(b) First, by the $\sigma_{\psi_\mu}$- strongly convexity of $\psi_\mu(\mathbf{x}, \cdot)$ and Cauchy-Schwarz inequality, we have

$$\sigma_{\psi_\mu}\|\mathbf{v}_\mu^*(\mathbf{x}_i)\|^2 \leq \langle\mathbf{v}_\mu^*(\mathbf{x}_i), \nabla_{\mathbf{yy}}^2\psi_\mu\left(\mathbf{x}_i, \mathbf{y}_\mu^*(\mathbf{x}_i)\right)\mathbf{v}_\mu^*(\mathbf{x}_i)\rangle$$
$$= \langle\mathbf{v}_\mu^*(\mathbf{x}_i), \nabla_{\mathbf{y}} f_i\left(\mathbf{x}_i, \mathbf{y}_\mu^*(\mathbf{x}_i)\right)\rangle \leq \|\mathbf{v}_\mu^*(\mathbf{x}_i)\|\|\nabla_{\mathbf{y}} f_i\left(\mathbf{x}_i, \mathbf{y}_\mu^*(\mathbf{x}_i)\right)\|,$$

which implies the conclusion.
Second, the definition of $\mathbf{v}_\mu^*(\mathbf{x}_i)$ implies that $\left[\nabla_{\mathbf{yy}}^2\psi_\mu(\mathbf{x}_i, \mathbf{y}_\mu^*(\mathbf{x}_i'))\right]\mathbf{v}_\mu^*(\mathbf{x}_i) = \nabla_{\mathbf{y}} f_i\left(\mathbf{x}_i, \mathbf{y}_\mu^*(\mathbf{x}_i)\right)$. Thus

$$\left[\nabla_{\mathbf{yy}}^2\psi_\mu(\mathbf{x}_i', \mathbf{y}_\mu^*(\mathbf{x}_i'))\right]\left[\mathbf{v}_\mu^*(\mathbf{x}_i') - \mathbf{v}_\mu^*(\mathbf{x}_i)\right]$$
$$= \left[\nabla_{\mathbf{y}} f_i\left(\mathbf{x}_i', \mathbf{y}_\mu^*(\mathbf{x}_i')\right) - \nabla_{\mathbf{y}} f_i\left(\mathbf{x}_i, \mathbf{y}_\mu^*(\mathbf{x}_i)\right)\right]$$
$$+ \left[\nabla_{\mathbf{yy}}^2\psi_\mu(\mathbf{x}_i, \mathbf{y}_\mu^*(\mathbf{x}_i)) - \nabla_{\mathbf{yy}}^2\psi_\mu(\mathbf{x}_i', \mathbf{y}_\mu^*(\mathbf{x}_i'))\right]\mathbf{v}_\mu^*(\mathbf{x}_i).$$

By (a), the $\sigma_{\psi_\mu}$- strongly convexity of $\psi_\mu(\mathbf{x}_i, \cdot)$ and Lipschitz continuity of $\nabla_{\mathbf{y}}\psi_\mu$ and $\nabla_{\mathbf{y}} f_i$, we have

$$\sigma_{\psi_\mu}\|\mathbf{v}_\mu^*(\mathbf{x}_i) - \mathbf{v}_\mu^*(\mathbf{x}_i')\|$$
$$\leq \left(L_{f_{i\mathbf{y}1}}\|\mathbf{x}_i - \mathbf{x}_i'\| + L_{f_{i\mathbf{y}2}}\|\mathbf{y}_\mu^*(\mathbf{x}_i) - \mathbf{y}_\mu^*(\mathbf{x}_i')\|\right)$$
$$+ \|\mathbf{v}_\mu^*(\mathbf{x}_i)\|\left(L_{\psi_{\mathbf{yy}1}}\|\mathbf{x}_i - \mathbf{x}_i'\| + L_{\psi_{\mathbf{yy}2}}\|\mathbf{y}_\mu^*(\mathbf{x}_i) - \mathbf{y}_\mu^*(\mathbf{x}_i')\|\right)$$
$$\leq \frac{\|\mathbf{x}_i - \mathbf{x}_i'\|}{\sigma_{\psi_\mu}}\left(\|\mathbf{v}_\mu^*(\mathbf{x}_i)\|\left(L_{\psi_{\mathbf{yy}2}} L_{\psi_{\mathbf{y}1}} + \sigma_{\psi_\mu} L_{\psi_{\mathbf{yy}1}}\right) + L_{f_{i\mathbf{y}2}} L_{\psi_{\mathbf{y}1}} + \sigma_{\psi_\mu} L_{f_{i\mathbf{y}1}}\right),$$

which implies the desired result. $\square$

Next, we have a lemma that characterizes the smoothness of $\Phi_\mu^i(\mathbf{x}_i) = f_i\left(\mathbf{x}_i, \mathbf{y}_\mu^*(\mathbf{x}_i)\right)$ in $\mathbf{x}_i$, without any boundedness assumption of $\nabla_{\mathbf{y}} f_i(\mathbf{x}_i, \mathbf{y}_i)$, where $\nabla\Phi_\mu^i(\mathbf{x}_i) = \nabla_{\mathbf{x}} f_i\left(\mathbf{x}_i, \mathbf{y}_\mu^*(\mathbf{x}_i)\right) - \nabla_{\mathbf{xy}}^2 \psi_\mu^i\left(\mathbf{x}_i, \mathbf{y}_\mu^*(\mathbf{x}_i)\right) \mathbf{v}_\mu^*(\mathbf{x}_i)$.

**Lemma 12.** *Suppose all assumptions holds, for any $i \in [m]$ and $\mathbf{x}_i \in \mathbb{R}^{p_1}$, we have*

$$\left\|\nabla\Phi_\mu^i(\mathbf{x}_i) - \nabla\Phi_\mu^i\left(\mathbf{x}_i'\right)\right\| \leq \frac{\left(C_{\Phi 1}\left\|\mathbf{v}_\mu^*(\mathbf{x}_i)\right\| + C_{\Phi 2}\right)}{\sigma_{\psi_\mu}^2} \left\|\mathbf{x}_i - \mathbf{x}_i'\right\|,$$

*where both $C_{\Phi 1} := L_{\psi_{\mathbf{y}1}}^2 L_{\psi_{\mathbf{yy}2}} + L_{\psi_{\mathbf{y}1}}\left(L_{\psi_{\mathbf{yy}1}} + L_{\psi_{\mathbf{xy}1}}\right)\sigma_{\psi_\mu} + L_{\psi_{\mathbf{xy}1}}\sigma_{\psi_\mu}^2$ and $C_{\Phi 2} := L_{\psi_{\mathbf{y}1}}^2 L_{f_{i\mathbf{y}2}} + L_{\psi_{\mathbf{y}1}}\left(L_{f_{i\mathbf{y}1}} + L_{f_{i\mathbf{x}2}}\right)\sigma_{\psi_\mu} + L_{f_{i\mathbf{x}1}}\sigma_{\psi_\mu}^2$ are constants.*

*Proof.* Recall that $\mathbf{v}_\mu^*(\mathbf{x}_i) = \left[\nabla_{\mathbf{yy}}^2 \psi_\mu\left(\mathbf{x}_i, \mathbf{y}_\mu^*(\mathbf{x}_i)\right)\right]^{-1} \nabla_{\mathbf{y}} f_i\left(\mathbf{x}_i, \mathbf{y}_\mu^*(\mathbf{x}_i)\right)$ and $\nabla\Phi_\mu(\mathbf{x}_i) = \nabla_{\mathbf{x}} f_i\left(\mathbf{x}_i, \mathbf{y}_\mu^*(\mathbf{x}_i)\right) - \left[\nabla_{\mathbf{xy}}^2 \psi_\mu\left(\mathbf{x}_i, \mathbf{y}_\mu^*(\mathbf{x}_i)\right)\right]\mathbf{v}_\mu^*(\mathbf{x}_i)$. Thus

$$\begin{aligned}
\nabla\Phi_\mu^i(\mathbf{x}_i) - \nabla\Phi_\mu^i\left(\mathbf{x}_i'\right) =& \nabla_{\mathbf{x}} f_i\left(\mathbf{x}_i, \mathbf{y}_\mu^*(\mathbf{x}_i)\right) - \nabla_{\mathbf{x}} f_i\left(\mathbf{x}_i', \mathbf{y}_\mu^*(\mathbf{x}_i')\right) \\
&+ \left[\nabla_{\mathbf{xy}}^2 \psi_\mu\left(\mathbf{x}_i', \mathbf{y}_\mu^*(\mathbf{x}_i')\right)\right]\mathbf{v}_\mu^*(\mathbf{x}_i') - \left[\nabla_{\mathbf{xy}}^2 \psi_\mu\left(\mathbf{x}_i, \mathbf{y}_\mu^*(\mathbf{x}_i)\right)\right]\mathbf{v}_\mu^*(\mathbf{x}_i).
\end{aligned}$$

First, by Lipschitz continuity of $\nabla_{\mathbf{x}} f_i$, we have

$$\left\|\nabla_{\mathbf{x}} f_i\left(\mathbf{x}_i, \mathbf{y}_\mu^*(\mathbf{x}_i)\right) - \nabla_{\mathbf{x}} f_i\left(\mathbf{x}_i', \mathbf{y}_\mu^*(\mathbf{x}_i')\right)\right\| \leq L_{f_{i\mathbf{x}1}}\left\|\mathbf{x}_i - \mathbf{x}_i'\right\| + L_{f_{i\mathbf{x}2}}\left\|\mathbf{y}_\mu^*(\mathbf{x}_i) - \mathbf{y}_\mu^*\left(\mathbf{x}_i'\right)\right\|.$$

Second, note that

$$\begin{aligned}
&\left[\nabla_{\mathbf{xy}}^2 \psi_\mu\left(\mathbf{x}_i', \mathbf{y}_\mu^*(\mathbf{x}_i')\right)\right]\mathbf{v}_\mu^*\left(\mathbf{x}_i'\right) - \left[\nabla_{\mathbf{xy}}^2 \psi_\mu\left(\mathbf{x}_i, \mathbf{y}_\mu^*(\mathbf{x}_i)\right)\right]\mathbf{v}_\mu^*(\mathbf{x}_i) \\
=& \left[\nabla_{\mathbf{xy}}^2 \psi_\mu\left(\mathbf{x}_i', \mathbf{y}_\mu^*(\mathbf{x}_i')\right)\right]\left[\mathbf{v}_\mu^*\left(\mathbf{x}_i'\right) - \mathbf{v}_\mu^*(\mathbf{x}_i)\right] \\
&+ \left[\nabla_{\mathbf{xy}}^2 \psi_\mu\left(\mathbf{x}_i', \mathbf{y}_\mu^*(\mathbf{x}_i')\right) - \nabla_{\mathbf{xy}}^2 \psi_\mu\left(\mathbf{x}_i, \mathbf{y}_\mu^*(\mathbf{x}_i)\right)\right]\mathbf{v}_\mu^*(\mathbf{x}_i).
\end{aligned}$$

Thus, by Lipschitz continuity of $\nabla_{\mathbf{yy}}^2 \psi_\mu$ and Lemma 11, we have

$$\begin{aligned}
&\left\|\left[\nabla_{\mathbf{xy}}^2 \psi_\mu\left(\mathbf{x}_i', \mathbf{y}_\mu^*(\mathbf{x}_i')\right)\right]\mathbf{v}_\mu^*\left(\mathbf{x}_i'\right) - \left[\nabla_{\mathbf{xy}}^2 \psi_\mu\left(\mathbf{x}_i, \mathbf{y}_\mu^*(\mathbf{x}_i)\right)\right]\mathbf{v}_\mu^*(\mathbf{x}_i)\right\| \\
\leq& L_{\psi_{\mathbf{y}1}}\left\|\mathbf{v}_\mu^*\left(\mathbf{x}_i'\right) - \mathbf{v}_\mu^*(\mathbf{x}_i)\right\| + \left\|\mathbf{v}_\mu^*(\mathbf{x}_i)\right\|\left(L_{\psi_{\mathbf{xy}1}}\left\|\mathbf{x}_i' - \mathbf{x}_i\right\| + L_{\psi_{\mathbf{xy}2}}\left\|\mathbf{y}_\mu^*(\mathbf{x}_i) - \mathbf{y}_\mu^*\left(\mathbf{x}_i'\right)\right\|\right) \\
\leq& \frac{L_{\psi_{\mathbf{y}1}}\left(L_{\mathbf{v}1}\left\|\mathbf{v}_\mu^*(\mathbf{x}_i)\right\| + L_{\mathbf{v}2}\right)}{\sigma_{\psi_\mu}^2}\left\|\mathbf{x}_i - \mathbf{x}_i'\right\| \\
&+ \left\|\mathbf{v}_\mu^*(\mathbf{x}_i)\right\|\left(L_{\psi_{\mathbf{xy}1}}\left\|\mathbf{x}_i' - \mathbf{x}_i\right\| + L_{\psi_{\mathbf{xy}}}\left\|\mathbf{y}_\mu^*(\mathbf{x}_i) - \mathbf{y}_\mu^*\left(\mathbf{x}_i'\right)\right\|\right).
\end{aligned}$$

where $L_{\psi_{\mathbf{y}1}}$ is constant given by $L_{\psi_{\mathbf{y}1}} = \mu L_{h_{i\mathbf{y}1}} + (1-\mu)L_{g_{i\mathbf{y}1}}$.

It implies the desired result since

$$L_{\psi_{\mathbf{y}1}} L_{\mathbf{v}1} + L_{\psi_{\mathbf{xy}2}} L_{\psi_{\mathbf{y}1}}\sigma_{\psi_\mu} + L_{\psi_{\mathbf{xy}1}}\sigma_{\psi_\mu}^2 = L_{\psi_{\mathbf{y}1}}^2 L_{\psi_{\mathbf{yy}2}} + L_{\psi_{\mathbf{y}1}}\left(L_{\psi_{\mathbf{yy}1}} + L_{\psi_{\mathbf{xy}2}}\right)\sigma_{\psi_\mu} + L_{\psi_{\mathbf{xy}1}}\sigma_{\psi_\mu}^2$$

and

$$L_{\psi_{\mathbf{y}1}} L_{\mathbf{v}2} + L_{f_{i\mathbf{x}2}} L_{\psi_{\mathbf{y}1}}\sigma_{\psi_\mu} + L_{f_{i\mathbf{x}1}}\sigma_{\psi_\mu}^2 = L_{\psi_{\mathbf{y}1}}^2 L_{f_{i\mathbf{y}2}} + L_{\psi_{\mathbf{y}1}}\left(L_{f_{i\mathbf{y}1}} + L_{f_{i\mathbf{x}2}}\right)\sigma_{\psi_\mu} + L_{f_{i\mathbf{x}1}}\sigma_{\psi_\mu}^2.$$

$\square$

Next lemma characterize the smoothness of $\mathbf{y}_\mu^*(\mathbf{x}_i)$ and $\mathbf{v}_\mu^*(\mathbf{x}_i)$ in $\mu$ when the LL problem has multiple minimizers.

**Lemma 13.** *Suppose assumptions holds, let $0 \leq \mu \leq 1/2$ and $\mu' \leq 2\mu$, the follow statements hold.*

*(a) For any $i \in [m]$,*

$$\left\|\mathbf{y}_\mu^*(\mathbf{x}_i) - \mathbf{y}_{\mu'}^*(\mathbf{x}_i)\right\| \leq \frac{2\left\|\nabla_{\mathbf{y}} h_i\left(\mathbf{x}_i, \mathbf{y}_\mu^*(\mathbf{x}_i)\right)\right\|}{\sigma_{h_i}} \cdot \frac{|\mu - \mu'|}{\mu'} = \frac{2\left\|\mathbf{y}_\mu^*(\mathbf{x}_i)\right\|}{\sigma_{h_i}} \cdot \frac{|\mu - \mu'|}{\mu'}.$$

*(b) For any $i \in [m]$,*

$$\left\|\mathbf{v}_\mu^*(\mathbf{x}_i) - \mathbf{v}_{\mu'}^*(\mathbf{x}_i)\right\| \leq \left(C_{\mathbf{v}1}\|\mathbf{v}_\mu^*(\mathbf{x}_i)\| + L_{f_{i\mathbf{y}2}}\right)\|\mathbf{y}_\mu^*(\mathbf{x}_i)\| \cdot \frac{|\mu - \mu'|}{(\mu')^2}$$

$$+ C_{\mathbf{v}2}\|\nabla_{\mathbf{y}}f_i\left(\mathbf{x}_i, \mathbf{y}_\mu^*(\mathbf{x}_i)\right)\| \cdot \frac{|\mu - \mu'|}{(\mu')^2},$$

*where both $C_{\mathbf{v}1} := 2\left(L_{h_{i\mathbf{yy}2}} + L_{g_{i\mathbf{yy}2}}\right)/\sigma_{h_i}^2$ and $C_{\mathbf{v}2} := 2\left(L_{f_{i\mathbf{y}2}} + L_{g_{i\mathbf{y}2}}\right)/\sigma_{h_i}^2$ are constants.*

*Proof.* (a) Recall that $\mathbf{y}_{i,\mu}^* = \mathbf{y}_\mu^*(\mathbf{x}_i)$ satisfies $\mu\nabla_{\mathbf{y}}h_i\left(\mathbf{x}_i, \mathbf{y}_{i,\mu}^*\right) + (1-\mu)\nabla_{\mathbf{y}}g_i\left(\mathbf{x}_i, \mathbf{y}_{i,\mu}^*\right) = 0$. Thus

$$(\mu - \mu')\nabla_{\mathbf{y}}h_i\left(\mathbf{x}_i, \mathbf{y}_{i,\mu}^*\right) + \mu'\left[\nabla_{\mathbf{y}}h_i\left(\mathbf{x}_i, \mathbf{y}_{i,\mu}^*\right) - \nabla_{\mathbf{y}}h_i\left(\mathbf{x}_i, \mathbf{y}_{i,\mu'}^*\right)\right]$$

$$+ (\mu' - \mu)\nabla_{\mathbf{y}}g_i\left(\mathbf{x}_i, \mathbf{y}_{i,\mu}^*\right) + (1-\mu')\left[\nabla_{\mathbf{y}}g_i\left(\mathbf{x}_i, \mathbf{y}_{i,\mu}^*\right) - \nabla_{\mathbf{y}}g_i\left(\mathbf{x}_i, \mathbf{y}_{i,\mu'}^*\right)\right] = 0,$$

which imply that

$$\mu'\left[\nabla_{\mathbf{y}}h_i\left(\mathbf{x}_i, \mathbf{y}_{i,\mu}^*\right) - \nabla_{\mathbf{y}}h_i\left(\mathbf{x}_i, \mathbf{y}_{i,\mu'}^*\right)\right] + (1-\mu')\left[\nabla_{\mathbf{y}}g_i\left(\mathbf{x}_i, \mathbf{y}_{i,\mu}^*\right) - \nabla_{\mathbf{y}}g_i\left(\mathbf{x}_i, \mathbf{y}_{i,\mu'}^*\right)\right]$$

$$= (\mu' - \mu)\nabla_{\mathbf{y}}h_i\left(\mathbf{x}_i, \mathbf{y}_{i,\mu}^*\right) + (\mu' - \mu)\frac{\mu}{1-\mu}\nabla_{\mathbf{y}}h_i\left(\mathbf{x}_i, \mathbf{y}_{i,\mu}^*\right) = (\mu' - \mu)\frac{\nabla_{\mathbf{y}}h_i\left(\mathbf{x}_i, \mathbf{y}_{i,\mu}^*\right)}{1-\mu}.$$

Since $h_i(\mathbf{x}_i, \cdot)$ is $\sigma_{h_i}$- strongly convex, we have

$$\langle\nabla_{\mathbf{y}}h_i\left(\mathbf{x}_i, \mathbf{y}_{i,\mu}^*\right) - \nabla_{\mathbf{y}}h_i\left(\mathbf{x}_i, \mathbf{y}_{i,\mu'}^*\right), \mathbf{y}_{i,\mu}^* - \mathbf{y}_{i,\mu'}^*\rangle \geq \sigma_{h_i}\|\mathbf{y}_{i,\mu}^* - \mathbf{y}_{i,\mu'}^*\|^2.$$

Similarly, since $g_i(\mathbf{x}_i, \cdot)$ is convex, we get

$$\langle\nabla_{\mathbf{y}}g_i\left(\mathbf{x}_i, \mathbf{y}_{i,\mu}^*\right) - \nabla_{\mathbf{y}}g_i\left(\mathbf{x}_i, \mathbf{y}_{i,\mu'}^*\right), \mathbf{y}_{i,\mu}^* - \mathbf{y}_{i,\mu'}^*\rangle \geq 0.$$

Combining the above inequalities, we have

$$0 \leq \mu'\sigma_{h_i}\|\mathbf{y}_{i,\mu}^* - \mathbf{y}_{i,\mu'}^*\|^2 \leq \left(\frac{\mu' - \mu}{1-\mu}\right)\langle\nabla_{\mathbf{y}}h_i\left(\mathbf{x}_i, \mathbf{y}_{i,\mu}^*\right), \mathbf{y}_{i,\mu}^* - \mathbf{y}_{i,\mu'}^*\rangle.$$

Since $0 \leq \mu \leq 1/2$, we get $1/(1-\mu) \leq 2$ and then

$$\mu'\sigma_{h_i}\|\mathbf{y}_{i,\mu}^* - \mathbf{y}_{i,\mu'}^*\|^2 \leq 2\|\nabla_{\mathbf{y}}h_i\left(\mathbf{x}_i, \mathbf{y}_{i,\mu}^*\right)\| \cdot \|\mu - \mu'\| \cdot \|\mathbf{y}_{i,\mu}^* - \mathbf{y}_{i,\mu'}^*\|$$

$$= 2\|\mathbf{y}_{i,\mu}^*\| \cdot \|\mu - \mu'\| \cdot \|\mathbf{y}_{i,\mu}^* - \mathbf{y}_{i,\mu'}^*\|$$

which implies the desired result.
(b)The definition of $\mathbf{v}_{i,\mu}^* = \mathbf{v}_\mu^*(\mathbf{x}_i)$ says that $\left[\nabla_{\mathbf{yy}}\psi_\mu(\mathbf{x}_i, \mathbf{y}_\mu^*(\mathbf{x}_i'))\right]\mathbf{v}_\mu^*(\mathbf{x}_i) = \nabla_{\mathbf{y}}f_i\left(\mathbf{x}_i, \mathbf{y}_\mu^*(\mathbf{x}_i)\right)$. Thus we have

$$\left[\nabla_{\mathbf{yy}}^2\psi_{\mu'}(\mathbf{x}_i, \mathbf{y}_{\mu'}^*)\right]\left(\mathbf{v}_{i,\mu}^* - \mathbf{v}_{i,\mu'}^*\right) + \left[\nabla_{\mathbf{yy}}^2\psi_\mu(\mathbf{x}_i, \mathbf{y}_\mu^*) - \nabla_{\mathbf{yy}}^2\psi_{\mu'}(\mathbf{x}_i, \mathbf{y}_{\mu'}^*)\right]\mathbf{v}_{i,\mu}^*$$

$$= \nabla_{\mathbf{y}}f_i\left(\mathbf{x}_i, \mathbf{y}_{i,\mu}^*\right) - \nabla_{\mathbf{y}}f_i\left(\mathbf{x}_i, \mathbf{y}_{i,\mu'}^*\right).$$

Multiplying the above equation by $\mathbf{v}_{i,\mu}^* - \mathbf{v}_{i,\mu'}^*$, by the $\sigma_{\psi_\mu}$- strongly convexity of $\psi_{\mu_t}(\mathbf{x}_i, \cdot)$, we get

$$\sigma_{\psi_{\mu'}}\|\mathbf{v}_{i,\mu}^* - \mathbf{v}_{i,\mu'}^*\|^2 \leq \|\nabla_{\mathbf{yy}}^2\psi_\mu(\mathbf{x}_i, \mathbf{y}_\mu^*) - \nabla_{\mathbf{yy}}^2\psi_{\mu'}(\mathbf{x}_i, \mathbf{y}_{\mu'}^*)\| \cdot \|\mathbf{v}_{i,\mu}^*\| \cdot \|\mathbf{v}_{i,\mu}^* - \mathbf{v}_{i,\mu'}^*\|$$

$$+ \|\nabla_{\mathbf{y}}f_i\left(\mathbf{x}_i, \mathbf{y}_{i,\mu}^*\right) - \nabla_{\mathbf{y}}f_i\left(\mathbf{x}_i, \mathbf{y}_{i,\mu'}^*\right)\| \cdot \|\mathbf{v}_{i,\mu}^* - \mathbf{v}_{i,\mu'}^*\|.$$

Note that by the definition of $\psi_\mu$ we have

$$\nabla_{\mathbf{yy}}^2\psi_\mu(\mathbf{x}_i, \mathbf{y}_{i,\mu}^*) - \nabla_{\mathbf{yy}}^2\psi_{\mu'}(\mathbf{x}_i, \mathbf{y}_{i,\mu'}^*)$$

$$= (\mu - \mu')\nabla_{\mathbf{yy}}^2h_i\left(\mathbf{x}_i, \mathbf{y}_{i,\mu}^*\right) + \mu'\left[\nabla_{\mathbf{yy}}^2h_i\left(\mathbf{x}_i, \mathbf{y}_{i,\mu}^*\right) - \nabla_{\mathbf{yy}}^2h_i\left(\mathbf{x}_i, \mathbf{y}_{i,\mu'}^*\right)\right]$$

$$+ (\mu' - \mu)\nabla_{\mathbf{yy}}^2g_i\left(\mathbf{x}_i, \mathbf{y}_{i,\mu}^*\right) + (1-\mu')\left[\nabla_{\mathbf{yy}}^2g_i\left(\mathbf{x}_i, \mathbf{y}_{i,\mu}^*\right) - \nabla_{\mathbf{yy}}^2g_i\left(\mathbf{x}_i, \mathbf{y}_{i,\mu'}^*\right)\right].$$

Since $\mu' \leq 2\mu \leq 1$, we get

$$\sigma_{\psi_{\mu'}}\|\mathbf{v}_{i,\mu}^* - \mathbf{v}_{i,\mu'}^*\|$$

$$\leq \left[\left(L_{h_{i\mathbf{y}2}} + L_{g_{i\mathbf{y}2}}\right)\|\mu - \mu'\| + \left(L_{h_{i\mathbf{yy}2}} + L_{g_{i\mathbf{yy}2}}\right)\|\mathbf{y}_{i,\mu}^* - \mathbf{y}_{i,\mu'}^*\|\right]\|\mathbf{v}_{i,\mu}^*\| + L_{f_{i\mathbf{y}2}}\|\mathbf{y}_{i,\mu}^* - \mathbf{y}_{i,\mu'}^*\|$$

$$\leq \left[\left(L_{h_{i\mathbf{yy}2}} + L_{g_{i\mathbf{yy}2}}\right)\|\mathbf{v}_{i,\mu}^*\| + L_{f_{i\mathbf{y}2}}\right]\|\mathbf{y}_{i,\mu}^* - \mathbf{y}_{i,\mu'}^*\| + \left(L_{h_{i\mathbf{y}2}} + L_{g_{i\mathbf{y}2}}\right)\|\mathbf{v}_{i,\mu}^*\| \cdot \|\mu - \mu'\|.$$

By (a), since $\sigma_{\psi_{\mu'}} = \mu'\sigma_{h_i}$ when $\sigma_{g_i} = 0$, we have

$$\|\mathbf{v}_{i,\mu}^* - \mathbf{v}_{i,\mu'}^*\| \leq \left[\left(L_{h_{i\mathbf{yy}2}} + L_{g_{i\mathbf{yy}2}}\right)\|\mathbf{v}_{i,\mu}^*\| + L_{f_{i\mathbf{y}2}}\right]\frac{2\left\|\mathbf{y}_{\mu}^*(\mathbf{x}_i)\right\|}{\sigma_{h_i}^2} \cdot \frac{|\mu - \mu'|}{(\mu')^2}$$

$$+ \frac{\left(L_{h_{i\mathbf{y}2}} + L_{g_{i\mathbf{y}2}}\right)}{\sigma_{h_i}}\|\mathbf{v}_{i,\mu}^*\| \cdot \frac{|\mu - \mu'|}{\mu'}.$$

Since $\mu' \leq 2\mu$, Lemma 11 implies that

$$\mu'\|\mathbf{v}_{i,\mu}^*\| \leq \mu'\frac{\left\|\nabla_{\mathbf{y}}f_i\left(\mathbf{x}_i, \mathbf{y}_{\mu}^*(\mathbf{x}_i)\right)\right\|}{\mu\sigma_{h_i}} \leq \frac{2\left\|\nabla_{\mathbf{y}}f_i\left(\mathbf{x}_i, \mathbf{y}_{\mu}^*(\mathbf{x}_i)\right)\right\|}{\sigma_{h_i}}.$$

Hence we get the desired result. $\qquad\square$

Finally, we present the lemma that characterizes the Lipschitz properties of the approximate gradient by using $\|\nabla_{\mathbf{y}}f_i(\mathbf{x}_i, \mathbf{y}_i)\| \leq L_{f_{i_0}}$ from Assumption 1.

**Lemma 14.** *For any $i \in [m]$,*

$$\|\nabla f_i(\mathbf{x}_i, \mathbf{y}_i) - \nabla f_i(\mathbf{x}_i', \mathbf{y}_i')\|^2 \leq L_{f_{i_1}}^2\|\mathbf{x}_i - \mathbf{x}_i'\|^2 + L_{f_{i_2}}^2\|\mathbf{y}_i - \mathbf{y}_i'\|^2,$$

*where $L_{f_{i_1}}^2 = 4L_{f_{i\mathbf{x}1}}^2 + 4L_{\psi_{\mathbf{y}1}}^2\frac{1}{\sigma_{\psi_\mu}^2}L_{f_{i\mathbf{y}1}}^2 + 4\frac{1}{\sigma_{\psi_\mu}^2}L_{f_{i_0}}^2L_{\psi_{\mathbf{xy}1}}^2 + 4L_{\psi_{\mathbf{y}1}}^2L_{f_{i_0}}^2\frac{1}{\sigma_{\psi_\mu}^4}L_{\psi_{\mathbf{yy}1}}^2$, and $L_{f_{i_2}}^2 = 4L_{f_{i\mathbf{x}2}}^2 + 4L_{\psi_{\mathbf{y}1}}^2\frac{1}{\sigma_{\psi_\mu}^2}L_{f_{i\mathbf{y}2}}^2 + 4\frac{1}{\sigma_{\psi_\mu}^2}L_{f_{i_0}}^2L_{\psi_{\mathbf{xy}2}}^2 + 4L_{\psi_{\mathbf{y}1}}^2L_{f_{i_0}}^2\frac{1}{\sigma_{\psi_\mu}^4}L_{\psi_{\mathbf{yy}2}}^2$.*

*Proof.* Recall that

$$\nabla f_i(\mathbf{x}_i, \mathbf{y}_i) = \nabla_{\mathbf{x}}f_i\left(\mathbf{x}_i, \mathbf{y}_i\right) - \nabla_{\mathbf{xy}}^2\psi_\mu^i\left(\mathbf{x}_i, \mathbf{y}_i\right)\mathbf{v}_i,$$

where $\mathbf{v}_i \in \mathbb{R}^{p_2}$ is define as:

$$\mathbf{v}_i := [\nabla_{\mathbf{yy}}^2\psi_\mu^i(\mathbf{x}_i, \mathbf{y}_i)]^{-1}\nabla_{\mathbf{y}}f_i(\mathbf{x}_i, \mathbf{y}_i).$$

Then, we have

$$\|\nabla f_i(\mathbf{x}_i, \mathbf{y}_i) - \nabla f_i(\mathbf{x}_i', \mathbf{y}_i')\|^2$$
$$\leq 2\|\nabla_{\mathbf{x}}f_i(\mathbf{x}_i, \mathbf{y}_i) - \nabla_{\mathbf{x}}f_i(\mathbf{x}_i', \mathbf{y}_i')\|^2$$
$$+ 2\|\nabla_{\mathbf{xy}}^2\psi_\mu^i\left(\mathbf{x}_i, \mathbf{y}_i\right)[\nabla_{\mathbf{yy}}^2\psi_\mu^i(\mathbf{x}_i, \mathbf{y}_i)]^{-1}\nabla_{\mathbf{y}}f_i(\mathbf{x}_i, \mathbf{y}_i)$$
$$- \nabla_{\mathbf{xy}}^2\psi_\mu^i\left(\mathbf{x}_i', \mathbf{y}_i'\right)[\nabla_{\mathbf{yy}}^2\psi_\mu^i(\mathbf{x}_i', \mathbf{y}_i')]^{-1}\nabla_{\mathbf{y}}f_i(\mathbf{x}_i', \mathbf{y}_i')\|^2$$
$$\leq 2\|\nabla_{\mathbf{x}}f_i(\mathbf{x}_i, \mathbf{y}_i) - \nabla_{\mathbf{x}}f_i(\mathbf{x}_i', \mathbf{y}_i')\|^2$$
$$+ 2\|\nabla_{\mathbf{xy}}^2\psi_\mu^i\left(\mathbf{x}_i, \mathbf{y}_i\right)[\nabla_{\mathbf{yy}}^2\psi_\mu^i(\mathbf{x}_i, \mathbf{y}_i)]^{-1}(\nabla_{\mathbf{y}}f_i(\mathbf{x}_i, \mathbf{y}_i) - \nabla_{\mathbf{y}}f_i(\mathbf{x}_i', \mathbf{y}_i'))\|^2$$
$$+ 2\|(\nabla_{\mathbf{xy}}^2\psi_\mu^i\left(\mathbf{x}_i, \mathbf{y}_i\right) - \nabla_{\mathbf{xy}}^2\psi_\mu^i\left(\mathbf{x}_i', \mathbf{y}_i'\right))[\nabla_{\mathbf{yy}}^2\psi_\mu^i(\mathbf{x}_i, \mathbf{y}_i)]^{-1}\nabla_{\mathbf{y}}f_i(\mathbf{x}_i, \mathbf{y}_i)\|^2$$
$$+ 2\|\nabla_{\mathbf{xy}}^2\psi_\mu^i\left(\mathbf{x}_i', \mathbf{y}_i'\right)([\nabla_{\mathbf{yy}}^2\psi_\mu^i(\mathbf{x}_i, \mathbf{y}_i)]^{-1}([\nabla_{\mathbf{yy}}^2\psi_\mu^i(\mathbf{x}_i, \mathbf{y}_i)]$$
$$- [\nabla_{\mathbf{yy}}^2\psi_\mu^i(\mathbf{x}_i', \mathbf{y}_i')])[\nabla_{\mathbf{yy}}^2\psi_\mu^i(\mathbf{x}_i', \mathbf{y}_i')]^{-1})\nabla_{\mathbf{y}}f_i(\mathbf{x}_i', \mathbf{y}_i')\|^2.$$

Then we have the following:

$$\|\nabla_{\mathbf{x}}f_i(\mathbf{x}_i, \mathbf{y}_i) - \nabla_{\mathbf{x}}f_i(\mathbf{x}_i', \mathbf{y}_i')\| \leq L_{f_{i\mathbf{x}1}}\|\mathbf{x}_i - \mathbf{x}_i'\|^2 + L_{f_{i\mathbf{x}2}}\|\mathbf{y}_i - \mathbf{y}_i'\|.$$

$$\|\nabla_{\mathbf{y}}f_i(\mathbf{x}_i, \mathbf{y}_i) - \nabla_{\mathbf{y}}f_i(\mathbf{x}_i', \mathbf{y}_i')\| \leq L_{f_{i\mathbf{y}1}}\|\mathbf{x}_i - \mathbf{x}_i'\| + L_{f_{i\mathbf{y}2}}\|\mathbf{y}_i - \mathbf{y}_i'\|.$$

Furthermore, we have

$$\|\nabla_{\mathbf{xy}}^2\psi_\mu^i\left(\mathbf{x}_i, \mathbf{y}_i\right) - \nabla_{\mathbf{xy}}^2\psi_\mu^i\left(\mathbf{x}_i', \mathbf{y}_i'\right)\| \leq L_{\psi_{\mathbf{xy}1}}\|\mathbf{x}_i - \mathbf{x}_i'\| + L_{\psi_{\mathbf{xy}2}}\|\mathbf{y}_i - \mathbf{y}_i'\|.$$

$$\|\nabla^2_{\mathbf{yy}}\psi^i_\mu(\mathbf{x}_i,\mathbf{y}_i) - \nabla^2_{\mathbf{yy}}\psi^i_\mu(\mathbf{x}'_i,\mathbf{y}'_i)\| \leq L_{\psi_{\mathbf{yy1}}}\|\mathbf{x}_i - \mathbf{x}'_i\| + L_{\psi_{\mathbf{yy2}}}\|\mathbf{y}_i - \mathbf{y}'_i\|.$$

Assuming that $\|\nabla^2_{\mathbf{xy}}\psi^i_\mu(\mathbf{x}_i,\mathbf{y}_i)\| \leq L_{\psi_{\mathbf{y1}}}$ and $\|\nabla_{\mathbf{y}}f_i(\mathbf{x}_i,\mathbf{y}_i)\| \leq L_{f_{i_0}}$, then we have

$$\|\nabla f_i(\mathbf{x}_i,\mathbf{y}_i) - \nabla f_i(\mathbf{x}'_i,\mathbf{y}'_i)\|^2$$
$$\leq 4L^2_{f_{i\mathbf{x1}}}\|\mathbf{x}_i - \mathbf{x}'_i\|^2 + 4L^2_{f_{i\mathbf{x2}}}\|\mathbf{y}_i - \mathbf{y}'_i\|^2$$
$$+ 4L^2_{\psi_{\mathbf{y1}}}\frac{1}{\sigma^2_{\psi_\mu}}L^2_{f_{i\mathbf{y1}}}\|\mathbf{x}_i - \mathbf{x}'_i\|^2 + 4L^2_{\psi_{\mathbf{y1}}}\frac{1}{\sigma^2_{\psi_\mu}}L^2_{f_{i\mathbf{y2}}}\|\mathbf{y}_i - \mathbf{y}'_i\|^2$$
$$+ 4\frac{1}{\sigma^2_{\psi_\mu}}L^2_{f_{i_0}}L^2_{\psi_{\mathbf{xy1}}}\|\mathbf{x}_i - \mathbf{x}'_i\|^2 + 4\frac{1}{\sigma^2_{\psi_\mu}}L^2_{f_{i_0}}L^2_{\psi_{\mathbf{xy2}}}\|\mathbf{y}_i - \mathbf{y}'_i\|$$
$$+ 4L^2_{\psi_{\mathbf{y1}}}L^2_{f_{i_0}}\frac{1}{\sigma^4_{\psi_\mu}}L^2_{\psi_{\mathbf{yy1}}}\|\mathbf{x}_i - \mathbf{x}'_i\| + 4L^2_{\psi_{\mathbf{y1}}}L^2_{f_{i_0}}\frac{1}{\sigma^4_{\psi_\mu}}L^2_{\psi_{\mathbf{yy2}}}\|\mathbf{y}_i - \mathbf{y}'_i\|.$$

Thus we have

$$\|\nabla f_i(\mathbf{x}_i,\mathbf{y}_i) - \nabla f_i(\mathbf{x}'_i,\mathbf{y}'_i)\|^2$$
$$\leq (4L^2_{f_{i\mathbf{x1}}} + 4L^2_{\psi_{\mathbf{y1}}}\frac{1}{\sigma^2_{\psi_\mu}}L^2_{f_{i\mathbf{y1}}} + 4\frac{1}{\sigma^2_{\psi_\mu}}L^2_{f_{i_0}}L^2_{\psi_{\mathbf{xy1}}} + 4L^2_{\psi_{\mathbf{y1}}}L^2_{f_{i_0}}\frac{1}{\sigma^4_{\psi_\mu}}L^2_{\psi_{\mathbf{yy1}}})\|\mathbf{x}_i - \mathbf{x}'_i\|^2$$
$$+ (4L^2_{f_{i\mathbf{x2}}} + 4L^2_{\psi_{\mathbf{y1}}}\frac{1}{\sigma^2_{\psi_\mu}}L^2_{f_{i\mathbf{y2}}} + 4\frac{1}{\sigma^2_{\psi_\mu}}L^2_{f_{i_0}}L^2_{\psi_{\mathbf{xy2}}} + 4L^2_{\psi_{\mathbf{y1}}}L^2_{f_{i_0}}\frac{1}{\sigma^4_{\psi_\mu}}L^2_{\psi_{\mathbf{yy2}}})\|\mathbf{y}_i - \mathbf{y}'_i\|^2$$
$$\leq L^2_{f_{i_1}}\|\mathbf{x}_i - \mathbf{x}'_i\|^2 + L^2_{f_{i_2}}\|\mathbf{y}_i - \mathbf{y}'_i\|^2,$$

where

$$L^2_{f_{i_1}} = 4L^2_{f_{i\mathbf{x1}}} + 4L^2_{\psi_{\mathbf{y1}}}\frac{1}{\sigma^2_{\psi_\mu}}L^2_{f_{i\mathbf{y1}}} + 4\frac{1}{\sigma^2_{\psi_\mu}}L^2_{f_{i_0}}L^2_{\psi_{\mathbf{xy1}}} + 4L^2_{\psi_{\mathbf{y1}}}L^2_{f_{i_0}}\frac{1}{\sigma^4_{\psi_\mu}}L^2_{\psi_{\mathbf{yy1}}},$$

$$L^2_{f_{i_2}} = 4L^2_{f_{i\mathbf{x2}}} + 4L^2_{\psi_{\mathbf{y1}}}\frac{1}{\sigma^2_{\psi_\mu}}L^2_{f_{i\mathbf{y2}}} + 4\frac{1}{\sigma^2_{\psi_\mu}}L^2_{f_{i_0}}L^2_{\psi_{\mathbf{xy2}}} + 4L^2_{\psi_{\mathbf{y1}}}L^2_{f_{i_0}}\frac{1}{\sigma^4_{\psi_\mu}}L^2_{\psi_{\mathbf{yy2}}}.$$

$\square$

# E  ADDITIONAL EXPERIMENTAL DETAILS AND RESULTS

All numerical experiments were conducted on a MacBook Pro equipped with an Apple M3 Pro chip, featuring a 12-core CPU with 6 performance cores and 6 efficiency cores, and 36GB of memory.

## E.1  A PEDAGOGICAL EXAMPLE

In this section, we present the results of a pedagogical example using the DUET and DSGT algorithms. For the DUET algorithm, we set the LL learning rate to 0.005, with $p = \frac{1}{10}$ and $\tau = \frac{1}{40}$. For the DSGT algorithm, the LL learning rate is set to 0.001, with $p = \frac{1}{15}$ and $\tau = \frac{1}{40}$.

The following figures illustrate the norm of the variables $\mathbf{x}$ and $\mathbf{y}$ during the optimization process.

## E.2  DECENTRALIZED META-LEARNING PROBLEMS WITH REAL-WORLD DATA:

In this section, we present more details for decentralized meta-learning problems with Real-World Data. At each node, we construct a neural network classifier comprising an input layer, a single hidden layer with 32 neurons using a sigmoid activation function, and a linear layer as final output layer parameterized by $\theta$. The hidden layer parameters $\mathbf{x}$ are shared across all nodes to ensure global consensus, while the output layer parameters $\theta$ are fine-tuned locally to adapt to the specific data available at each node. The objective functions $f_i$ and $g_i$ are: $f_i(\mathbf{x},\theta) = \sum_{(s_{ij},b_{ij})\in D_i} b_{ij}\ln(y_j(\mathbf{x},\theta;s_{ij})) + \frac{\gamma}{2}\|\theta\|^2$, and $g_i(\mathbf{x},\theta) = \sum_{(s_{ij},b_{ij})\in D_i} b_{ij}\ln(y_j(x,\theta;s_{ij}))$, where $(s_{ij},b_{ij})$ represents the $j$-th sample at node $i$, with $s_{ij}$ as the feature vector and $b_{ij}$ as the corresponding label. Here, $\gamma = 0.1$ denotes the regularization coefficient, and $y_j(\mathbf{x},\theta;s_{ij})$ denotes the output of the neural network.

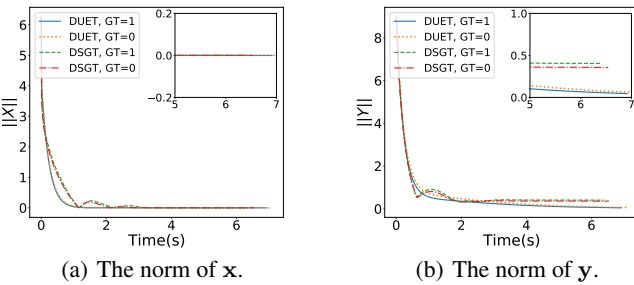

(a) The norm of **x**.    (b) The norm of **y**.

Figure 4: The norms of **x** and **y**.

The decentralized stochastic gradient descent (DSGD) approach is used as baseline for i.i.d. case that updates $\theta$ first by gradient descent and then uses the updated $\theta$ to calculate the gradient of **x**, subsequently updating **x** via SGD.

We compare the performance of DUET and DSGT in both i.i.d. and non-i.i.d. settings. Figures 5 and 6 illustrate the train loss and accuracy results for both settings.

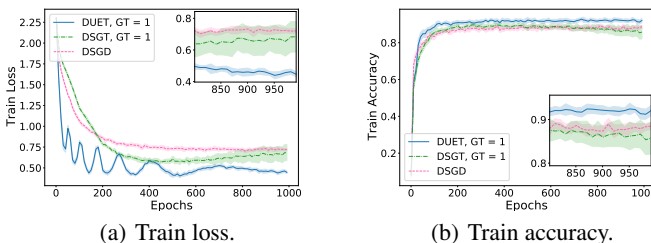

(a) Train loss.    (b) Train accuracy.

Figure 5: Comparisons between DUET and DSGT in the i.i.d. data scenario on the meta-learning problem with a 5-agent network on MNIST.

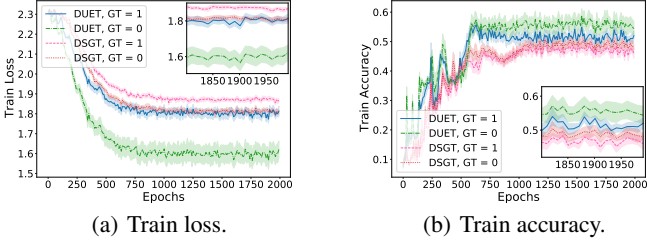

(a) Train loss.    (b) Train accuracy.

Figure 6: Comparisons between DUET and DSGT in the non-i.i.d. data scenario on the meta-learning problem with a 5-agent network on MNIST.

In Figure 5, the DUET algorithm demonstrates superior performance by achieving the highest training accuracy, along with fast convergence. For the non-i.i.d. case (Figure 6), the algorithms with gradient tracking handle the heterogeneity of the data well, with better performance.

Figure 7 shows the label distributions of data heterogeneity across different nodes, highlighting the strong non-i.i.d. nature of the data used in our experiments. This visual representation of non-i.i.d. data distribution provides a clear understanding of the varying degrees of heterogeneity.

For a 10-agent network (Figure 8) and a 50-agent network (Figure 9) in the i.i.d case, DUET continues to perform effectively and demonstrates the best performance among the tested methods DSGT and

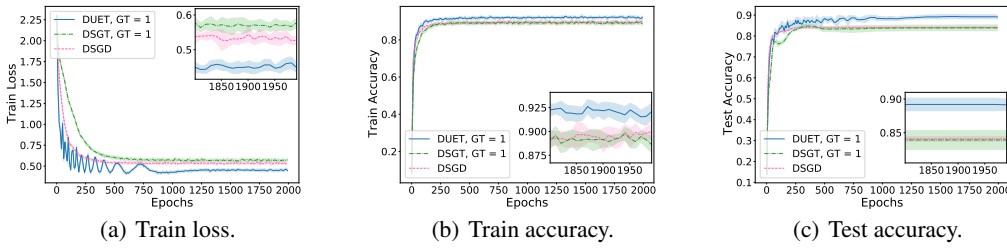

Figure 7: Label distributions of data heterogeneity across nodes for non-iid case on the meta-learning problem with a 5-agent network on MNIST.

baseline DSGD, maintaining robust convergence and accuracy even in larger, more complex network settings.

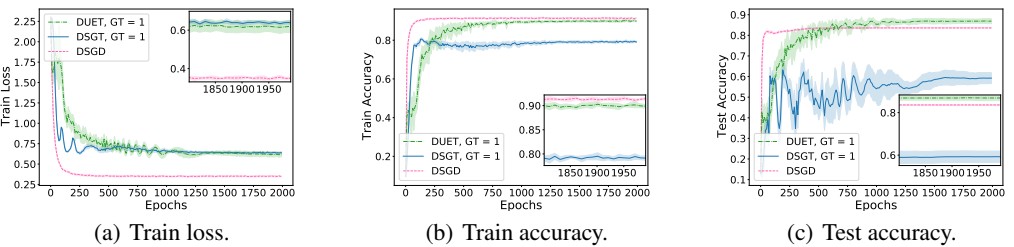

(a) Train loss.          (b) Train accuracy.          (c) Test accuracy.

Figure 8: Comparisons between DUET and DSGT in the i.i.d. data scenario on the meta-learning problem with a 10-agent network on MNIST.

(a) Train loss.          (b) Train accuracy.          (c) Test accuracy.

Figure 9: Comparisons between DUET and DSGT in the i.i.d. data scenario on the meta-learning problem with a 50-agent network on MNIST.

The following table summarizes the parameter settings for the DUET , DSGT , and DSGD algorithms under both i.i.d. and non-i.i.d. cases, illustrating the diverse configurations tested in our experiments.

E.3   DECENTRALIZED HYPERPARAMETER OPTIMIZATION WITH REAL-WORLD DATA:

In this section, we explore the hyperparameter optimization problem, which is formulated as a bilevel optimization task. Specifically, we employ softmax regression (with parameters $\mathbf{y}_i$) as the classifier and introduce hyperparameters $\mathbf{x}_i$ to weight samples for training.

Let $D_{tr,i}$ and $D_{val,i}$ as the training and validation sets for agent $i$. We define $\ell(\mathbf{y}_i; \mathbf{u}_i, \mathbf{v}_i)$ as the cross-entropy loss function, where $\mathbf{y}_i$ denotes the classification parameters and $(\mathbf{u}_i, \mathbf{v}_i)$ are the data pairs. The LL problem minimizes the softmax regression loss over the training dataset, and the LL objective function is formulated as follows: $g_i(\mathbf{x}_i, \mathbf{y}_i) = \sum_{(\mathbf{u}_i, \mathbf{v}_i) \in D_{tr,i}} [\sigma(\mathbf{x}_i)] \ell(\mathbf{y}_i; \mathbf{u}_i, \mathbf{v}_i)$, where $\mathbf{x}_i$ is the hyperparameter that penalizes the objective for different training sample.

Simultaneously, the UL problem aims to improve the performance of the regression model on the validation dataset by fine-tuning the hyperparameters. The UL objective is defined as:

| Setting | Algorithm | UL Learning Rate | LL Learning Rate | Parameters $(\mu, p)$ |
|---------|-----------|------------------|------------------|------------------------|
| IID | DUET | 10 | 0.01 | $(0.1, \frac{1}{5})$ |
| | DSGT | 0.5 | 0.00001 | $(0.5, \frac{1}{5})$ |
| | DSGD | 1 | 0.01 | - |
| non-IID | DUET | 0.1 | 0.0001 | $(0.1, \frac{1}{5})$ |
| | DSGT | 0.001 | 0.001 | $(0.5, \frac{1}{5})$ |

Table 3: Parameter settings for DUET, DSGT, and DSGD under iid and non-iid conditions on the meta-learning problem on MNIST.

$f_i(\mathbf{x}_i, \mathbf{y}_i) = \sum_{(\mathbf{u}_i, \mathbf{v}_i) \in D_{val,i}} \ell(\mathbf{y}(\mathbf{x}_i); \mathbf{u}_i, \mathbf{v}_i) + \frac{1}{d} \sum_{s=1}^{d} \frac{\rho(\mathbf{y}_s(\mathbf{x}_i))^2}{1 + \rho(\mathbf{y}_s(\mathbf{x}_i))^2}$, where $\rho = 10^{-4}$ is the regularization parameter. This regularizer is non-convex and is applied in a distributed manner. To evaluate the proposed method, we use the FashionMNIST dataset, which consists of images of clothing categories and serves as an alternative to the classic MNIST dataset. For each agent, the dataset is split into training, validation, and testing sets, each containing 5000 samples. We compare the performance of DUET and DSGT algorithms in terms of test accuracy and F1 score, utilizing a 10-agent communication network with a connection probability $p_c = 0.5$.

As show in Figure 10, the analysis of F1 score and test accuracy reveals similar trends across the three algorithms, indicating consistency between the metrics in evaluating performance. DUET achieves the best results in both F1 score and test accuracy, with the fastest convergence and highest stability across epochs. DSGT follows closely, showing competitive performance but slightly behind DUET. In contrast, DSGD, which lacks gradient tracking, exhibits slower convergence, lower overall performance in both metrics. These results highlight the effectiveness of gradient tracking in DUET and DSGT, with DUET emerging as the most robust and generalizable approach. The similar trends in F1 score and test accuracy further validate the reliability of these algorithms' performance in decentralized optimization.

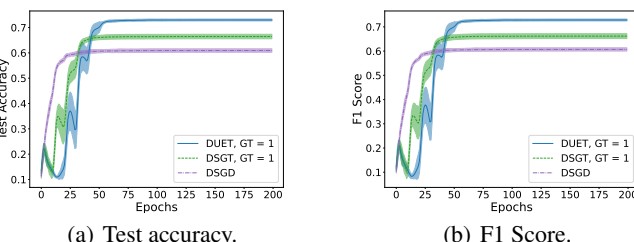

(a) Test accuracy.       (b) F1 Score.

Figure 10: Comparisons between DUET and DSGT on the hyperparameter optimization problem with 10-agent network on FashinMNIST.

The following table summarizes the parameter settings for the DUET, DSGT, and DSGD algorithms on the hyperparameter optimization problem with 10-agent network on FashinMNIST.

| Algorithm | UL Learning Rate | LL Learning Rate | Parameters $(\mu, p)$ |
|-----------|------------------|------------------|------------------------|
| DUET | 0.001 | 0.1 | $(0.1, \frac{1}{5})$ |
| DSGT | 0.1 | 0.01 | $(0.9, \frac{1}{5})$ |
| DSGD | 0.01 | 0.01 | - |

Table 4: Parameter settings for DUET, DSGT, and DSGD on the hyperparameter optimization problem with 10-agent network on FashinMNIST.

