# OpenReview forum: "DUET: Decentralized Bilevel Optimization without Lower-Level Strong Convexity"
_ICLR.cc/2025/Conference — ICLR 2025 Poster_

### Official Review · Reviewer_7zfE · 2024-10-27

**Soundness:** 2
**Presentation:** 2
**Contribution:** 2
**Rating:** 6
**Confidence:** 5

**Summary:**

This paper considers decentralized bilevel optimization (DBO) without using lower-level strong convexity (LLSC).

To tackle this problem, the authors first propose a novel convergence metric called LLSC-less convergence. On top of this, they propose DUET that uses local updates and gradient tracking based communication updates to solve the DBO problem. The authors further provide a rigorous theoretical analysis of DUET, and obtain a finite-time convergence analysis which reveals the number of communication rounds to find an eps-stationary point is O(eps^{-3.5}) and O(eps^{-2.5}) under different settings. Besides, the authors also provide an extension of their algorithm -- DSGT, in which an additional assumption, upper-level strong convexity, is imposed. Similar results are obtained for DSGT.

Some numerical experiments are conducted to showcase the efficiency of the proposed algorithms over existing ones.

**Strengths:**

1. (Weaker assumptions) The assumptions used in the paper are standard and do not impose strong convexity in the lower-level problem.

2. (Novel metric) The authors propose a novel metric to characterize the convergence of DBO problems without LLSC.

**Weaknesses:**

1. (Novelty of the theory) Although the proposed metric and algorithm are novel, the analysis used to obtain the final convergence is quite standard in distributed optimization and bilevel optimization literature.

2. (Simple experiments with incomplete context) The experimental settings are relatively simple -- A Pedagogical Example and MNIST dataset are so simple that they may not well represent the non-convex nature of lower-level non-convexity in many bilevel optimization problems. Some important details about the experiments setup are missing -- for example it would be better if the architecture of the neural network in the meta learning experiments is reported, otherwise the readers need to go through the code to figure it out.

3. (Unclear Table 1) Most of the algorithms in Table 1 use the eps-stationary convergence metric while this paper uses LLSC-less. However, this is not directly visible in this table. Meanwhile, the centralized settings may be moved to the appendix, as this paper mainly focuses on the distributed setting.

4. (Misleading name) The name of the extended version of DUET, DSGT, might be a little misleading -- as it refers to distributed stochastic optimization with gradient tracking in the distributed optimization literature. It would be better if the authors could use another name for the extended version of their algorithm.

**Questions:**

1. Can the authors provide some insights on the lower bound of this problem? For example the current O(eps^{-2.5}) seems not to be optimal, and it would be interesting to study the lower bound of DBO under the LLSC-less metric.

2. Is it correct to claim the complexity results are O(eps^{-3.5}) and O(eps^{-2.5}) in Corollary 2 and 4? Technically speaking \tau cannot be 0 and thus the logic "\tau --> 0 leads to -3.5 and -2.5" complexity seems a little bit problematic.

---

> ### Author Response · Authors · 2024-11-22
>
> > **Comment 1:** (Novelty of the theory) Although the proposed metric and algorithm are novel, the analysis used to obtain the final convergence is quite standard in distributed optimization and bilevel optimization literature.
>
> **Our Response:** Thanks for your comments. We note that our theoretical analysis is actually quite non-standard due to the absence of LLSC in decentralized bilevel optimization. These challenges necessitate novel metrics, proof techniques, and significant modifications to the existing proof fameworks. In what follows, we will elaborate the novelty of our theoretical analysis:
>
> 1. **General Challenges Without LLSC:** Without LLSC, the LL solution $\mathbf{y}^*$ may not be unique, and the hyper-gradient is ill-defined. Therefore, we have to redefine the stationarity convergence metric using the KKT stationarity through a logically equivalent reformulated problem. These signficant changes necessitate new algorithmic designs on how to track and bound the variations in both the LL solution and dual variables in the KKT system across iterations. Note that the existing methods based on the LLSC setting do not need to handle these complications, since strong convexity guarantees a unique LL solution and simplifies convergence analysis. Hence, the theoretical analyses of the existing methods are much simpler compared to ours.
>
> 2. **New Proof Techniques:** Due to the new challenges in the non-LLSC setting stated above, we have to propose several new proof techniques as follows:
>
> * In **Lemma 4**, we establish new bounds on the deviations of the current LL solution $\mathbf{y} _ {i,t+1}$ and dual multiplier $\mathbf{v} _ {i,t+1}$, i.e., $\left\\|\mathbf{y} _ {i,t+1}-\mathbf{y} _ {\mu_t}^*\left(\mathbf{x} _ {i,t}\right)\right\\|^2$ and $\left\\|\mathbf{v} _ {i,t+1}-\mathbf{v} _ {\mu_t}^*\left(x_{i,t}\right)\right\\|^2$.
>
> * In **Lemma 5**, we further analyze and bound the variations of both the primal LL solution and the dual multiplier as the regularization parameter $\mu_t$ diminishes. Specifically, we handle terms such as: $\left\\|\mathbf{y} _ {\mu_t}^*\left(x _ {i,t}\right) - \mathbf{y} _ {\mu_{t+1}}^*\left(\mathbf{x} _ {i,t+1}\right)\right\\|^2$ and $\left\\|\mathbf{v} _ {\mu_t}^*\left(\mathbf{x} _ {i,t}\right) - \mathbf{v} _ {\mu_{t+1}}^*\left(\mathbf{x} _ {i,t+1}\right)\right\\|^2.$ These terms capture the evolution of the optimal LL solution and its associated dual multipliers, which are influenced by the diminishing regularization parameter $\mu_t$ and decentralized updates.
> Unlike in settings with LLSC, where strong convexity simplifies the analysis, we cannot directly bound terms like $\left\\|\mathbf{y} _ {i,t+1}-\mathbf{y} _ {t+1}^*\left(\mathbf{x} _ {i,t+1}\right)\right\\|^2-\left\\|\mathbf{y} _ {i,t}-\mathbf{y} _ {t+1}^*\left(\mathbf{x} _ {i,t}\right)\right\\|^2$ and turn out to be quite challenging to analyze. In our theortical analysis, based on Lemma 4 and Lemma 5, our proof must carefully account for the interplay between diminishing regularization $\left\\|\mathbf{y} _ {i,t+1}-\mathbf{y} _ {\mu_{t+1}}^*\left(\mathbf{x} _ {i,t+1}\right)\right\\|^2-\left\\|\mathbf{y} _ {i,t}-\mathbf{y} _ {\mu_t}^*\left(\mathbf{x} _ {i,t}\right)\right\\|^2$ and $\left\\|\mathbf{v} _ {i,t+1}-\mathbf{v} _ {\mu_{t+1}}^*\left(\mathbf{x} _ {i,t+1}\right)\right\\|^2-\left\\|\mathbf{v} _ {i,t}-\mathbf{v} _ {\mu_t}^*\left(\mathbf{x} _ {i,t}\right)\right\\|^2$.  This is very **different** from the counterpart in the LLSC setting, where one can easily obtain $\left\\|\mathbf{y} _ {i,t+1}-\mathbf{y} _ {t+1}^*\left(\mathbf{x} _ {i,t+1}\right)\right\\|^2-\left\\|\mathbf{y} _ {i,t}-\mathbf{y} _ {t+1}^*\left(\mathbf{x} _ {i,t}\right)\right\\|^2$, which are fundamentally **different** from the complicated terms appearing in proofs in the non-LLSC setting.
>
> 3. **Impacts on Aggregation Function:** The use of the diminishing regularization parameter $\mu_t$ also implies that care must be taken in the aggregation function design. Unlike standard decentralized methods, our aggregation function incorporates a time-varying of $\mu_t$, which requires that both the primal and dual variables remain synchronized as $\mu_t$ evolves. This adjustment is essential to maintain convergence in the non-LLSC setting and our theoretical analysis in this aspect is new.

---

> ### Author Response · Authors · 2024-11-22
>
> > **Comment 2:** (Simple experiments with incomplete context) The experimental settings are relatively simple -- A Pedagogical Example and MNIST dataset are so simple that they may not well represent the non-convex nature of lower-level non-convexity in many bilevel optimization problems. Some important details about the experiments setup are missing -- for example it would be better if the architecture of the neural network in the meta learning experiments is reported, otherwise the readers need to go through the code to figure it out.
>
> **Our Response:** Thanks for your comments. The details about the experiment setup are as follows:
>
> 1. **Choice of Problems:** Since our work addresses the case in DBO where the LL problem is convex but not strongly convex, and the UL problem is non-convex, the problems we chose (the pedagogical example and the decentralized meta-learning) align with this focus and reflect the theoretical assumptions of our proposed algorithm.
>
> 2. **Choice of Datasets:** The use of simpler datasets such as MNIST and the Pedagogical Example is to clearly illustrate the core concepts and validate the effectiveness of our proposed algorithm. These datasets provide a controlled environment to demonstrate the scalability and performance of the method.
>
> 3. **Expanded Experiments:** We are currently working on additional experiment results. We will provide the additional experiment results in both the revised manuscript as well as the response in here as soon as we finish the additional experiments.

---

> > ### Author Response · Authors · 2024-11-25
> >
> > Regarding your comment on limited experimental results, in this rebuttal period, we have conducted two more sets of experiments to demonstrate the efficacy of our proposed DUET method:
> >
> > 1) **Expanded Meta-Learning with 50 Agents:** To evaluate the scalability of our method, we have added experiments on a meta-learning task with 50 agents. These results further validate the robustness of our approach in larger-scale distributed settings.
> > 2) **Hyperparameter Optimization on FashionMNIST:** We have added experiments on hyperparameter optimization using the FashionMNIST dataset to illustrate the applicability of our method to a wider class of bilevel optimization problems.

---

> ### Author Response · Authors · 2024-11-22
>
> > **Comment 3:** (Unclear Table 1) Most of the algorithms in Table 1 use the eps-stationary convergence metric while this paper uses LLSC-less. However, this is not directly visible in this table. Meanwhile, the centralized settings may be moved to the appendix, as this paper mainly focuses on the distributed setting.
>
> **Our Response:** Thanks for your comments. We would like to clarify a misunderstanding regarding the stationary measures and the purpose of the table:
>
> 1. **Stationarity Measures in the Convex Setting:** For convex settings, it is not true that most algorithms in Table 1 use the same $\epsilon$-stationary convergence metric. In fact, these algorithms have different stationarity measures, since each is tailored to the specific method for solveing the problems. Our work introduces a new stationary measure to address the unique challenges of decentralized bilevel optimization in the non-LLSC setting.
>
> 2. **Purpose of Table 1:** The primary goal of Table 1 is not to compare stationary measures but to highlight the differences in problem settings of the listed algorithms. This includes the distinction between centralized and decentralized settings, the presence or absence of LLSC, and the types of objectives considered (e.g., convex, strongly convex, or non-convex). The table serves to provide readers with a comprehensive understanding of how our work fits within the broader literature.
>
> 3. **Placement of Centralized Settings:** While our paper focuses on the decentralized setting, centralized settings are included in Table 1 to provide additional context and to better highlight our contributions. Moving these to the appendix may obscure the broader comparison and reduce the table’s function to showcase differences in problem settings.
>
>
> > **Comment 4:** (Misleading name) The name of the extended version of DUET, DSGT, might be a little misleading -- as it refers to distributed stochastic optimization with gradient tracking in the distributed optimization literature. It would be better if the authors could use another name for the extended version of their algorithm.
>
> **Our Response:**
> Thanks for your comments. We clarify that DSGT is **not** an extension of our algorithm DUET. Rahter, it is an extension of the centralized SL-BAMM method, which is also done by us and used as a baseline for performance comparison. Regarding the DSGT naming, we understand that the name DSGT might cause confusion, as it is associated with distributed stochastic optimization with gradient tracking. To avoid such misunderstanding, we will rename DSGT in our paper to a less used term (e.g., SL-BAMM+) in the revised manuscript.
>
> > **Comment 5:** Can the authors provide some insights on the lower bound of this problem? For example the current O(eps^{-2.5}) seems not to be optimal, and it would be interesting to study the lower bound of DBO under the LLSC-less metric.
>
> **Our Response:** Thanks for your question. To our knowledge, there is no existing research on the lower bounds for decentralized bilevel optimization in the non-LLSC setting, which is a fundamental and challenging problem itself and deserves a dedicated paper. In fact, even our convergence rate upper bound $O(\epsilon^{-2.5})$ is the first for DBO problems without LLSC, let alone the corresponding lower bound. This gap highlights a lack of fundamental understanding regarding the complexity of DBO problems in the non-LLSC setting. We appreciate the reviewer for pointing out this important research direction, which we will further investigate in our future studies.
>
>
> > **Comment 6:** Is it correct to claim the complexity results are O(eps^{-3.5}) and O(eps^{-2.5}) in Corollary 2 and 4? Technically speaking \tau cannot be 0 and thus the logic "\tau --> 0 leads to -3.5 and -2.5" complexity seems a little bit problematic.
>
> **Our Respinse:** Thanks for your comments. You are correct that $\tau$ cannot exactly be 0, and the complexities $O(\epsilon^{-3.5})$ and $O(\epsilon^{-2.5})$ represent asymptotic behavior as $\tau \to 0$. These results should be interpreted as sample complexity bounds that our method approaches in the limit. However, since $\tau$ can be made arbitrarily close to 0, one can always achieve a sample complexity bound that is ever so slightly larger than $O(\epsilon^{-3.5})$ and $O(\epsilon^{-2.5})$.

---

### Official Review · Reviewer_D8W2 · 2024-10-29

**Soundness:** 3
**Presentation:** 2
**Contribution:** 3
**Rating:** 6
**Confidence:** 3

**Summary:**

To address the limitations of most existing decentralized bilevel optimization (DBO) methods, which predominantly rely on lower-level strong convexity (LLSC), this paper proposes a single-loop DBO algorithm named Diminishing Quadratically-Regularized Bilevel Decentralized Optimization (DUET). This algorithm circumvents the necessity for LLSC by incorporating a diminishing quadratic regularization into the lower-level objective. DUET achieves an iteration complexity of $\mathcal{O}(1/T^{1-5p-\frac{11}{4}\tau})$ for approximating KKT-stationary point convergence under more lenient assumptions, where $p$ and $\tau$ serve as control parameters for the lower-level learning rate and averaging, respectively. Additionally, DUET integrates gradient tracking to effectively tackle data heterogeneity, a significant hurdle in DBO scenarios.

**Strengths:**

1. This paper proposes DUET, a single-loop algorithm that integrates gradient tracking and consensus updates to avoid the computational complexity of conventional double-loop structure in bilevel optimization while ensuring convergence in decentralized settings with data heterogeneity, which is the first algorithm with provable convergence for DBO without LLSC.

2. Without gradient heterogeneity assumptions, the proposed algorithm overcomes the challenges of consensus errors in DBO with data heterogeneity, ensuring agents can synchronize their updates effectively without relying on LLSC.

**Weaknesses:**

1. My primary concern is the presentation of this paper. For instance:

1-1. It is imperative to delineate the feasible region for $p$ and $\tau$ in line 81;

1-2. The definitions of communication costs and communication complexity must be elucidated in lines 105 and 117;

1-3. In line 138, it is noted that several methods exist which offer enhanced complexity solutions for simple bilevel problems, as evidenced by references [1, 2, 3, 4];

1-4. The formulation of the decentralized bilevel optimization problem should be introduced earlier in the paper, ideally in the Section introduction;

1-5. A precise definition of $\nabla \mathcal{L}$ is essential and should be provided in line 231. Furthermore, given that $\nabla \mathcal{L}$ is a matrix (since $x$ may be a matrix), it is crucial to specify the type of norm to be used (F-norm or 2-norm); moreover, similarly, the definition (vector or matrix) of $\nabla \Phi$ in line 347 is also important.

1-6. The definitions of $x_{i,t}$ and $y_{i,t}$ remain ambiguous in line 242. Here, I assume that these represent the vectors $x_{i}$ and $y_{i}$ at the $t$-th iteration;

1-7. The explanation of the solution $v_{i,\mu_t}^*(x_{i,t})$ to the linear system and the multiplier $v_t$ could potentially lead to confusion among readers.

1-8. The citation format in the References section is incorrect. For example, the second reference and the reference on line 557 lack the publication location of the article, while the reference on line 574 omits the article's page numbers.

Typo:

1. line 402, "for all $k$" should be "for all $t$".

[1] Merchav, Roey, and Shoham Sabach. "Convex Bi-level Optimization Problems with Nonsmooth Outer Objective Function." SIAM Journal on Optimization 33.4 (2023): 3114-3142.

[2] Chen, Pengyu, et al. "Penalty-based Methods for Simple Bilevel Optimization under H\"{o} lderian Error Bounds." arXiv preprint arXiv:2402.02155 (2024).

[3] Cao, Jincheng, et al. "An Accelerated Gradient Method for Simple Bilevel Optimization with Convex Lower-level Problem." arXiv preprint arXiv:2402.08097 (2024).

[4] Samadi, Sepideh, Daniel Burbano, and Farzad Yousefian. "Achieving optimal complexity guarantees for a class of bilevel convex optimization problems." 2024 American Control Conference (ACC). IEEE, 2024.

**Questions:**

1. The update mode of Algorithm 1 is similar to the algorithm of [5], so I want to know the distinctions between these two algorithms. Furthermore, I am curious about whether the step of updating the optimal multiplier mirrors the step of solving the linear system within the hypergradient, given that the descent direction for the multiplier $v$, as proposed in this paper, aligns closely with the gradient descent step used for solving the linear system in the hypergradient. Considering this, I question whether the assumption of boundedness concerning the multiplier $v$ and the resolution of the linear system presented in Theorem 3 can be consolidated into a singular assumption.

2. The rationale behind the utilization of the projection parameters $r_v^t$ and $r_y^t$ remains ambiguous (line 301).

3. The definitions of $\otimes$ and $\bar{\beta}, \bar{\mu}$ are not clearly articulated (lines 347 and 362).

4. To enhance the understanding of Theorem 5, a comparative analysis with other algorithms is recommended.

5. For other questions, please see the weaknesses above.


[5] Liu, Risheng, et al. "Towards extremely fast bilevel optimization with self-governed convergence guarantees." arxiv preprint arxiv:2205.10054 (2022).

---

> ### Author Response · Authors · 2024-11-22
>
> > **Comment 1:** My primary concern is the presentation of this paper. For instance:
> 1-1. It is imperative to delineate the feasible region for $p$ and $\tau$ in line 81;
>
> **Our Response:** Thanks for your suggesiton. We have added the feasible region for $p$ and $\tau$ in the revised manuscript.
>
> >1-2. The definitions of communication costs and communication complexity must be elucidated in lines 105 and 117;
>
> **Our Response:** Thanks for your suggestion. Here, both "communication costs" and "communication complexity" refer to the total rounds of communications required by an algorithm to achieve an $\epsilon$-stationary point. We have explicitly defined the terms "communication costs" and "communication complexity" to avoid ambiguity and ensure consistent usage throughout the paper.
>
>
> >1-3. In line 138, it is noted that several methods exist which offer enhanced complexity solutions for simple bilevel problems, as evidenced by references [1, 2, 3, 4];
>
> **Our Response:** Thanks for providng pointers to the related works.
>
> [1] addressed convex bilevel optimization problems with nonsmooth outer objectives, introducing a generalized sub-gradient method to the bilevel setting and achieves sub-linear convergence rates for outer and inner objectives under mild assumptions. They additionally improves to a linear rate for strongly convex outer objectives while allowing for nonsmoothness. However, our work specifically deals with smooth UL objectives.
>
> [2] proposed a penalty-based algorithm that leverages Hölderian error bounds to achieve convergence rates of $O(1/\epsilon^2)$, assuming a convex UL objective.
>
> [3] introduced an accelerated gradient method specifically designed for bilevel problems with a convex lower-level problem, achieving convergence rates of $O(1/\sqrt{\epsilon})$ for smooth convex objectives.
>
> [4] achieved optimal complexity guarantees for a class of convex bilevel problems using advanced regularization techniques and strong convexity assumptions.
>
> While their work ([2, 3, 4]) assume UL convexity, our work focuses on non-convex UL objectives, broadening the applicability to more general problem settings.
>
> We will enrich the references in the revised manuscript to include all necessary publication ([1, 2, 3, 4]).
>
> >1-4. The formulation of the decentralized bilevel optimization problem should be introduced earlier in the paper, ideally in the Section introduction;
>
> **Our Response:** Thanks for the suggestion. We will move the problem formulation to the first paragraph of the introduction to provide early context for readers.
>
> >1-5. A precise definition of $\nabla \mathcal{L}$ is essential and should be provided in line 231. Furthermore, given that $\nabla \mathcal{L}$ is a matrix (since may be a matrix), it is crucial to specify the type of norm to be used (F-norm or 2-norm); moreover, similarly, the definition (vector or matrix) of $\nabla \Phi$ in line 347 is also important.
>
> **Our Response:** Thanks for your comment. However, we would like to clarify that $\nabla \mathcal{L}$ is a vector, not a matrix. This is because the Lagrangian function in Line 221 is scalar-valued. Also, the UL variable $\mathbf{x}$ is a vector in $\mathbb{R}^{p_1}$ (cf. the definition of $\mathbf{x}$ in Line 190. Note that, even if a learning model contains multiple layers of matrices, one can always construct such a vector $\mathbf{x}$ by stacking the columns of these matrices without loss of generality). Hence, the gradient of $\mathcal{L}$ with respect to $\mathbf{x}$ is a vector. The norm we use in Line 231 is the $\ell_2$ norm. Hence, we use the $\|\cdot\|$ norm notation following the convention. To make the paper more reader-friendly, we have added a notation table in the revised manuscript.
>
> >1-6. The definitions of $x_{i,t}$ and $y_{i,t}$ remain ambiguous in line 242. Here, I assume that these represent the vectors $x_{i}$ and $y_{i}$ at the
> $t$-th iteration;
>
> **Our Response:** Thanks for the comment. We would like to further clarify that $\mathbf{x} _ {i,t}$ and $\mathbf{y} _ {i,t}$ represent the values of UL and LL variables at agent $i$ in iteration $t$, respectively. To this ambiguity, we have added the definition of $\mathbf{x} _ {i,t}$ and $\mathbf{y} _ {i,t}$ in the revised manuscript.

---

> > ### Author Response · Authors · 2024-11-22
> >
> > >1-7. The explanation of the solution $v^{*} _ {i,\mu_t}({x}_{i,t})$ to the linear system and the multiplier could potentially lead to confusion among readers.
> >
> > **Our Response:** Thanks for your comments. However, we don't quite fully understand what the reviewer's confusion is. We would highly appreciate if the reviewer could further elaborate. But if our guess is right, then it appears that there are some misunderstandings regarding the use and meaning of the notation $\mathbf{v}^{*} _ {i,\mu_t}({x}_{i,t})$ here, and we would like to further clarify as follows:
> >
> > * The notation $\mathbf{v}^{*} _ {i,\mu_t}({x}_{i,t})$ is only used for representing the solution of the linear system in Line 255. We haven't used it to denote the dual multiplier anywhere else.
> > * The notation of the dual multiplier value of node $i$ in each iteration $t$ is $\mathbf{v} _ {i,t}$ and the optimal dual solution that the series $\\{ \mathbf{v} _ {i,t} \\}_{t=1}^{\infty}$ converge to is denoted as $\mathbf{v}_i^*$.
> >
> > In other words, even though they all share the letter $\mathbf{v}$, they are different notations with differnt meanings. Please let us know whether our interpretation of your comment is correct or not, and we are always happy to further clarify if the confusion remains.
> >
> > >1-8. The citation format in the References section is incorrect. For example, the second reference and the reference on line 557 lack the publication location of the article, while the reference on line 574 omits the article's page numbers.
> >
> > **Our Reponse:** Thanks for catching these erros. In this revision, we have made the following corrections:
> > * The second reference now includes the missing publication location.
> > * The reference in Line 557 has been updated with the correct publication details.
> > * The reference in Line 574 now includes the missing page numbers.
> >
> >
> > >Typo: line 402, "for all $k$" should be "for all $t$".
> >
> > **Our Response:** Thanks for catching the typo. We have fixed the typo and changed it to "for all t."

---

> > > ### Comment · Reviewer_D8W2 · 2024-11-23
> > > **For 1-7 and 1-8**
> > >
> > > 1-7: Thanks for the authors' response. I know that these two notations have different meanings. The problem is that authors can replace the notation $\mathbf{v}$ in $\mathbf{v}_{i,\mu_t}^*$ with $\mathbf{s}$ or another notation, to prevent ambiguity and ensure clarity in the notation. This adjustment is significant because the main theorems (e.g., Theorem 3 and Lemma 1) involve both notations. Additionally, the authors use $\\|\mathbf{v}\_{i,t} - \mathbf{v}\_{{\mu}\_t}^*(\mathbf{x}\_{i,t})\\|$ to represent the approximation error of the dual parameter, which makes this part difficult to follow.
> > >
> > > 1-8: The references contain some other errors that require correction. However, this is not a significant drawback for me.

---

> > > > ### Author Response · Authors · 2024-11-25
> > > >
> > > > **Our Response:** Thanks for your suggestion. We agree with the reviewer's suggestion that changing the notation $\mathbf{v}$ to $\mathbf{s}$ will help avoid ambiguity and ensure clarity. In the revised version, we will make the notation according to the reviewer's suggstion. Regarding the errors in the references, we will again make sure to correct those errors.

---

> ### Author Response · Authors · 2024-11-22
>
> > **Comment 2:** The update mode of Algorithm 1 is similar to the algorithm of [5], so I want to know the distinctions between these two algorithms.
>
>
> **Our Response:** Thanks for your comments. We would like to further clarify the differences between our algorithm and the approach in [5]:
>
> 1. **Different Assumptions on the UL Objective Functions:** Ref. [5] assumes the UL objective $F(x,y)$ is strongly convex (cf. Assumption 3.1 in Ref. [5]), so that they can aggregate the UL and LL objective functions to recover LLSC in the LL variable $\mathbf{y}$. This strongly convex assumption of the UL objective function significantly limits the applicability of their proposed approach, since many real-world bilevel optimization problems involve non-convex UL objectives. In stark contrast, our work does **not** assume the UL objective $F(x,y)$ to be strongly convex. Instead, we address the non-LLSC challenge through the proposed **diminishing quadratic regularization** approach, which temporarily enforces a LLSC property in each iteration that is gradually diminishing. As the regularization paremter $\mu_t$ shrinks over the iterations, our DUET algorithm guarantees the convergence to a KKT-stationary solution of the original DBO problem.
>
> 2. **Decentralized Problem Setting:** Our DUET algorithm is designed for decentralized bilevel optimization (DBO), where agents operate in a peer-to-peer network and share information through local communication. This requires a **consensus-based aggregation** mechanism to synchronize the UL updates across the agents, while accounting for communication constraints, consensus errors, and heterogeneity among agents simultaneously. All these algorithm designs are **absent** in Ref. [5], which is focused on *centralized* bilevel optimization.

---

> > ### Author Response · Authors · 2024-11-22
> >
> > >Furthermore, I am curious about whether the step of updating the optimal multiplier mirrors the step of solving the linear system within the hypergradient, given that the descent direction for the multiplier $v$, as proposed in this paper, aligns closely with the gradient descent step used for solving the linear system in the hypergradient. Considering this, I question whether the assumption of boundedness concerning the multiplier $v$ and the resolution of the linear system presented in Theorem 3 can be consolidated into a singular assumption.
> >
> > **Our Response:** Thanks for your question. Again, we suspect that this question may be arising due to the same confusion in your Comment 1-7 regarding the notation $\mathbf{v}^{*} _ {i,\mu_t}({x}_{i,t})$, and we would like to further clarify.
> >
> > First of all, the answer to your question "...wether the step of updating the optimal multiplier mirrors the steps of solving the linear system within the hyper-gradient..." is *no*. Hence, the boundedness assumption concerning the multiplier $\mathbf{v}$ and the solution of the linear system in Theorem 3 cannot be consolidated into a singluar assumption. To see this, we will explain the process of dual multiplier updates and solving the linear system within the hyper-gradient separately as follows:
> >
> > 1) **The Updates of Dual Multipliers:** From Eq. (6) in our paper, the dual multiplier update direction can be written as:  $\mathbf{d} _ {\mathbf{v}}^{i,t}=\nabla _ {\mathbf{y}} f_i\left(\mathbf{x} _ {i,t}, \mathbf{y} _ {i,t}\right)-\nabla _ {\mathbf{y} \mathbf{y}}^2 \psi^i _ {\mu_{t}}\left(\mathbf{x} _ {i,t}, \mathbf{y} _ {i,t}\right) \mathbf{v} _ {i,t},$ which is due to the recovered LLSC upon augmenting the LL objective function with Eq. (4). This update is motivated by the SOBA approach in [1] and can be interpreted as taking the gradient of a qudratic programming (QP) problem $\mathbf{v}^{\top} \nabla _ {\mathbf{y} \mathbf{y}}^2 \psi^i _ {\mu_{t}}\left(\mathbf{x} _ {i,t}, \mathbf{y} _ {i,t}\right) \mathbf{v} + \nabla_{\mathbf{y}} f_i\left(\mathbf{x} _ {i,t}, \mathbf{y} _ {i,t}\right) \mathbf{v} = 0$, whose solution is the implicit gradient of the $\mu_t$-regularized bilevel optimization problem. This QP problem's solution is also the solution of the linear system $\nabla _ {\mathbf{y} \mathbf{y}}^2 \psi^i _ {\mu_{t}}\left(\mathbf{x} _ {i,t}, \mathbf{y} _ {i,t}\right) \mathbf{v} = \nabla _ {\mathbf{y}} f _ i\left(\mathbf{x} _ {i,t}, \mathbf{y} _ {i,t}\right)$. Since the $\mu_t$-regularized bilevel optimization problem has LLSC, this QP's solution is also the optimal dual solution in the KKT system of the reformulated problem with $\mu_t$-regularization according to Ref. [2]. Combining all the above, the dual updates for $\mathbf{v}_{i,t}$ in Eqs. (5) and (6) can be interpreted as taking a **single gradient step** on the QP problem associated with inputs $\mathbf{x} _ {i,t}$, $\mathbf{y} _ {i,t}$, $\mathbf{v} _ {i,t}$, and $\mu_t$.

---

> > > ### Author Response · Authors · 2024-11-22
> > >
> > > 2) **The Steps of Solving the Linear System within the Hyper-gradient:** First of all, we want to point out that our algorithm does *not* require multiple steps to explicitly solve the linear equation system to obtain $\mathbf{v}^* _ {i,\mu_t}(\mathbf{x} _ {i,t}):= [\nabla^2 _ {\mathbf{y}\mathbf{y}}\psi _ {\mu_t}^i(\mathbf{x} _ {i,t}, \mathbf{y}^* _ {i,\mu_t}(\mathbf{x} _ {i,t}))]^{-1} \nabla _ {\mathbf{y}}f _ i(\mathbf{x} _ {i,t}, \mathbf{y}^* _ {i,\mu_t}(\mathbf{x} _ {i,t}))$. Even if we need to solve for $\mathbf{v}^* _ {i,\mu_t}(\mathbf{x} _ {i,t})$ by, say, using a series of gradient descent updates and compare them with the trajactory of $\mathbf{v} _ {i,t}$, the steps could still be quite different because they are the solutions to different linear equation systems. To see this, note that the steps for solving $\mathbf{v} _ {i,\mu_t}^*(\mathbf{x} _ {i,t})$ is associated with the linear equation system $\nabla^2 _ {\mathbf{y}\mathbf{y}}\psi _ {\mu_t}^i(\mathbf{x} _ {i,t}, \mathbf{y}^* _ {i,\mu_t}(\mathbf{x} _ {i,t})) \mathbf{v} = \nabla _ {\mathbf{y}}f _ i(\mathbf{x} _ {i,t}, \mathbf{y}^* _ {i,\mu_t}(\mathbf{x} _ {i,t}))$ with $\mathbf{x} _ {i,t}$, $\mathbf{y} _ {i,\mu_t}^*(\mathbf{x} _ {i,t})$, and $\mu_t$ as inputs. As mentioned earlier, the steps for solving $\mathbf{v} _ {i,t}$ is corresponding to solving the linear system $\nabla _ {\mathbf{y} \mathbf{y}}^2 \psi^i _ {\mu_{t}}\left(\mathbf{x} _ {i,t}, \mathbf{y} _ {i,t}\right) \mathbf{v} = \nabla _ {\mathbf{y}} f _ i\left(\mathbf{x} _ {i,t}, \mathbf{y} _ {i,t}\right)$ with $\mathbf{x} _ {i,t}$, $\mathbf{y} _ {i,t}$, $\mathbf{v} _ {i,t}$, and $\mu_t$ as inputs. Note that $\mathbf{y} _ {i,t}$ is **not** necessarily close to $\mathbf{y} _ {i,\mu_t}^*(\mathbf{x} _ {i,t})$ since $\mathbf{y} _ {i,t}$ is only a single-gradient-step update approximation of $\mathbf{y} _ {i,\mu_t}^*(\mathbf{x} _ {i,t})$. Thus, there is **no** guarantee that the statement *"...the step of updating the optimal multiplier mirrors the steps of solving the linear system within the hyper-gradient..."* would hold in general.
> > >
> > > [1] M. Dagréou, P. Ablin, S. Vaiter, and T. Moreau. A framework for bilevel optimization that enables stochastic and global variance reduction algorithms. In Advances in Neural Information Processing Systems, volume 35, pp. 26698–26710. Curran Associates, Inc., 2022.
> > >
> > > [2] Risheng Liu, Yaohua Liu, Wei Yao, Shangzhi Zeng, and Jin Zhang. Averaged method of multipliers for bi-level optimization without lower-level strong convexity. In Proceedings of International Conference on Machine Learning, pp. 21839–21866, 2023a.

---

> > > > ### Comment · Reviewer_D8W2 · 2024-11-23
> > > > **For Question 1**
> > > >
> > > > Thanks for the authors' response. In practice, we solve the linear system approximately, which eliminates the need for $\mathbf{y}\_{i,\mu_t}^*(\mathbf{x}\_{i,t})$. In my view, both the update of the dual multiplier $\mathbf{v}$ and the solution $\mathbf{v}\_{i,\mu_t}$ to the linear system can be interpreted as approximate gradient descent steps for the linear system. Could the authors revisit this point to determine whether the boundedness assumption for the multiplier and the resolution of the linear system, as stated in Theorem 3, can be consolidated into a single assumption?
> > > >
> > > > I'm sorry if I misunderstood the response.

---

> > > > > ### Author Response · Authors · 2024-11-25
> > > > >
> > > > > Thank you for your comment. We appreciate the opportunity to further clarify this point. If we understand your question correctly, then the boundedness assumption concerning the multiplier $\mathbf{v} _ {i,t}$ and the solution of the linear system $\left\\|\mathbf{v} _ {\mu _t}^*\left(\mathbf{x} _ {i,t}\right)\right\\|$ in Theorem 3 **cannot** be consolidated into a single assumption  due to their distinct roles in the analysis:
> > > > >
> > > > > 1. Boundedness of $\mathbf{v} _ {i,t}$: As stated in Theorem 3, the boundedness of $\mathbf{v} _ {i,t}$ is important for linking the convergence rates measured by $\min_{0 \leq t \leq T} {\Pi (\mathbf{x_t}, \mathbf{y_t}, \mathbf{v_t})}$ to the convergence rates as measured by the KKT residual. Specificly, with the boundedness of $\mathbf{v}_{i,t}$, there exists a positive constant $C$, such that $\mathrm{KKT}(\mathbf{x_t}, \mathbf{y_t}, \mathbf{v_t})\leq C ( \Pi (\mathbf{x_t}, \mathbf{y_t}, \mathbf{v_t})+\mu_t^2)$.
> > > > >
> > > > > 2. Boundedness of $\left\\|\mathbf{v} _ {\mu_t}^*\left(\mathbf{x} _ {i,t}\right)\right\\|$: As stated in Lemma 2, the boundedness of $\left\\|\mathbf{v} _ {\mu  _ t}^*\left(\mathbf{x} _ {i,t}\right)\right\\|$ is essential for controlling the contraction codfficients of key terms in the the descent lemma of the dencetralzied implied UL objective, such as $\left\lVert\bar{\mathbf{h}} _ {{\mathbf{x}}}^t \right\rVert^2$, $\\| \bar{\mathbf{x}} _ {t}- \mathbf{x} _ {i,t}\\|^2$ and $\\|\mathbf{y} _ {i,t}-\mathbf{y} _ {\mu_t}^*(\mathbf{x} _ {i,t})\\|^2$. Since $\left\\|\mathbf{v} _ {\mu _ t}^*\left(\mathbf{x} _ {i,t}\right)\right\\|$ is derived from the augmented LL objective with strongly convexity, it closed form allows us to explicitly bound $\left\\|\mathbf{v} _ {\mu _t}^*\left(\mathbf{x} _ {i,t}\right)\right\\|$ (Lemma 3).
> > > > >
> > > > > Again, we hope we have understood your question correctly and our responses above have clarified your questions. We are happy to further clarify if any question remains. Thanks!

---

> ### Author Response · Authors · 2024-11-22
>
> > **Comment 3:** The rationale behind the utilization of the projection parameters $r_v^t$ and $r_y^t$ remains ambiguous (line 301).
>
> **Our Response:** Thanks for comments. We would like to clarify that the roles of radii $r_y^t$ and $r_v^t$ in the projection steps for $\mathbf{y} _ {i,t}$ and $\mathbf{v} _ {i,t}$ are to ensure that the sequences $\mathbf{y} _ {i,t}$ and $\mathbf{v} _ {i,t}$ remain bounded with radii $r_y^t$ and $r_v^t$, respectively. These boundedness results are important for establishing convergence due to the following reasons:
>
> 1) **Inducing Boundedness of $\mathbf{x} _ {i,t}$:** In Lemma 8, we show that the boundedness of $\mathbf{y} _ {i,t}$ and $\mathbf{v} _ {i,t}$, ensured by the above projection steps, directly results in the boundedness of the UL variables $\mathbf{x} _ {i,t}$. This interplay is important for maintaining feasible updates across iterations.
>
> 2) **Convergence via Bounded $\mathbf{x} _ {i,t}$:** In Lemma 3, the boundedness of $\mathbf{x} _ {i,t}$ is a key condition for establishing the overall convergence of the algorithm. By controlling the growth of $r_y^t$ and $r_v^t$, the projection steps ensure that $\mathbf{x} _ {i,t}$ remains bounded, thereby establishing convergence guarantee.
>
> We will revise the manuscript to clarify the role of these projection parameters $r_v^t$ and $r_y^t$, and their connection to Lemma 8 and Lemma 3.
>
> > **Comment 4:** The definitions of $\otimes$ and $\bar\beta$, $\bar \mu$ are not clearly articulated (lines 347 and 362).
>
> **Our Response:** Thanks for your comments. The $\otimes$ symbol represents the Kronecker product in linear algebra. $\bar\beta$ and $\bar\mu$ are the initial LL learning rate and averaging control parameter, respectively. Both $\bar{\beta}$ and $\bar{\mu}$ are constants. We have defined and clarified these terms in the revised manuscript.
>
> > **Comment 5:** To enhance the understanding of Theorem 5, a comparative analysis with other algorithms is recommended.
>
> **Our Response:** Thanks for your suggestion. However, since our work is the first in the areas of decentralized bilevel optimization (DBO) without LLSC, there is a lack of existing algorithms as comparison baselines in the current literature. Due to this reason, in our numeical studies, we have to develop another DBO method called DSGT, which can handle the non-LLSC setting by adapting a related centralized bilevel optimization algorithm to the decentralized setting. Unfortunately, adapting centralized methods to the decentralized setting is highly non-trivial and may not be always possible due to the unique challenges in the decentralized setting, such as communication constraints, consensus errors, and heterogeneous data distributions. As a first step toward a theoretical and algorithmic foundation for solivng DBO problems without LLSC, the significance of Theorem 5 is that it shows it is possible to design algorithms that solve non-LLSC DBO problems without provable convergence rate guarantees.

---

> ### Comment · Reviewer_D8W2 · 2024-11-25
>
> Thanks for the author's responses. As the authors demonstrate, if it is true that the algorithm proposed in this paper is the first attempt at DBO, I am willing to increase my score to 6. However, I may decrease my confidence from 3 to 2 if the authors cannot confirm that this paper is the first attempt at DBO. Can the authors confirm this point again?

---

> > ### Author Response · Authors · 2024-11-27
> >
> > **Response:** Thanks so much for raising your score! Yes, we can **confirm** that, to our knowledge, our paper is the first work on decentralized bilevel optimization (DBO) that does **not** assume lower-level strong convexity (LLSC). Meanwhile, the existing works in the DBO literature so far **all** rely on the LLSC assumption (see the following list of works on DBO to date that we are aware of upon a thorough literature survey):
> >
> > [1] Xuxing Chen, Minhui Huang, and Shiqian Ma. Decentralized bilevel optimization. Optimization Letters, pages 1–65, 2024.
> >
> > [2] Xuxing Chen, Minhui Huang, Shiqian Ma, and Krishna Balasubramanian. Decentralized stochastic bilevel optimization with improved per-iteration complexity. In International Conference on Machine Learning, pages 4641–4671. PMLR, 2023.
> >
> > [3] Youran Dong, Shiqian Ma, Junfeng Yang, and Chao Yin. A single-loop algorithm for decentralized bilevel optimization. arXiv preprint arXiv:2311.08945, 2023.
> >
> > [4] Boao Kong, Shuchen Zhu, Songtao Lu, Xinmeng Huang, and Kun Yuan. Decentralized bilevel optimization over graphs: Loopless algorithmic update and transient iteration complexity. arXiv preprint arXiv:2402.03167,2024.
> >
> > [5] Zhuqing Liu, Xin Zhang, Prashant Khanduri, Songtao Lu, and Jia Liu.
> > Interact: Achieving low sample and communication complexities in decentralized bilevel learning over networks. In Proceedings of the Twenty-Third International Symposium on Theory, Algorithmic Foundations, and Protocol Design for Mobile Networks and Mobile Computing, pages 61–70, 2022.
> >
> > [6] Zhuqing Liu, Xin Zhang, Prashant Khanduri, Songtao Lu, and Jia Liu. Prometheus: taming sample and communication complexities in constrained decentralized stochastic bilevel learning. In International Conference on Machine Learning, pages 22420–22453. PMLR, 2023.
> >
> > [7] Songtao Lu, Xiaodong Cui, Mark S Squillante, Brian Kingsbury, and Lior Horesh. Decentralized bilevel optimization for personalized client learning. In ICASSP 2022-2022 IEEE International Conference on Acoustics, Speech and Signal Processing (ICASSP), pages 5543–5547. IEEE, 2022.
> >
> > [8] Songtao Lu, Siliang Zeng, Xiaodong Cui, Mark Squillante, Lior Horesh, Brian Kingsbury, Jia Liu, and Mingyi Hong. A stochastic linearized augmented lagrangian method for decentralized bilevel optimization. Advances in Neural Information Processing Systems, 35:30638–30650, 2022.
> >
> > [9] Peiwen Qiu, Yining Li, Zhuqing Liu, Prashant Khanduri, Jia Liu, Ness B Shroff, Elizabeth Serena Bentley, and Kurt Turck. Diamond: Taming sample and communication complexities in decentralized bilevel optimization. In IEEE INFOCOM 2023-IEEE Conference on Computer Communications, pages 1–10. IEEE, 2023.
> >
> > [10] Xiaoyu Wang, Xuxing Chen, Shiqian Ma, and Tong Zhang. Fully first-order methods for decentralized bilevel optimization. arXiv preprint arXiv:2410.19319, 2024.
> >
> > [11] Shuoguang Yang, Xuezhou Zhang, and Mengdi Wang. Decentralized gossip-based stochastic bilevel optimization over communication networks. Advances in neural information processing systems, 35:238–252, 2022.
> >
> > [12] Yihan Zhang, My T Thai, Jie Wu, and Hongchang Gao. On the communication complexity of decentralized bilevel optimization. arXiv preprint arXiv:2311.11342, 2023.
> >
> > [13] Shuchen Zhu, Boao Kong, Songtao Lu, Xinmeng Huang, and Kun Yuan. Sparkle: A unified single-loop primal-dual framework for decentralized bilevel optimization. arXiv preprint arXiv:2411.14166, 2024.
> >
> > [14] Ziqin Chen and Yongqiang Wang. Locally differentially private decentralized stochastic bilevel optimization with guaranteed convergence accuracy. In Forty-first International Conference on Machine Learning, 2024.
> >
> > [15] Hongchang Gao, Bin Gu, and My T Thai. On the convergence of distributed stochastic bilevel optimization algorithms over a network. In International Conference on Artificial Intelligence and Statistics, pages 9238–9281. PMLR, 2023.

---

> > > ### Comment · Reviewer_D8W2 · 2024-11-29
> > >
> > > Thanks for the author's claims, I keep my confidence at 3.

---

### Official Review · Reviewer_KZ3e · 2024-11-01

**Soundness:** 2
**Presentation:** 1
**Contribution:** 1
**Rating:** 3
**Confidence:** 4

**Summary:**

This work presents a decentralized bilevel optimization (DBO) algorithm, termed DUET, designed to solve local bilevel tasks in multi-agent systems without requiring a central server. Unlike traditional DBO methods that rely on the lower-level strong convexity (LLSC) assumption to ensure unique solutions and a well-defined hypergradient, DUET eliminates the need for LLSC by introducing diminishing quadratic regularization to the lower-level objective.

**Strengths:**

1. A single-loop algorithm for LLSC-less DBO is proposed in this work.

2. The proposed algorithm can be used to handle the challenges of consensus errors in DBO with data heterogeneity.

**Weaknesses:**

After reading this work, I have some major concerns.

1. What is the need for this algorithm in the ML community? This question arises from two main points: 1) Although this work avoids the strong convexity assumption, it still relies on the convexity assumption, whereas objectives in ML are often non-convex. Could you provide additional theoretical analyses that do not depend on convexity assumptions? 2) This work emphasizes the deterministic setting of the algorithm. While the deterministic setting is indeed important in optimization, I am curious about the challenges in analyzing convergence rates under a stochastic setting in the proposed algorithm. In ML, stochastic gradient descent is far more commonly used than deterministic approaches.

2. The relaxed stationarity (KKT-stationarity) is employed in this work without adequate justification. It is essential to provide a clear rationale for this relaxation, as it may result in significantly different solutions compared to the original problem. The stationarity measures are somewhat different from the traditional bilevel optimization works.

3. In Corollary 2, in addition to discussing the number of communication rounds required for the proposed algorithm to converge, it is also essential to analyze the number of bits needed during this process. Can you provide some comparisons about the communication complexity between the proposed method with the existing distributed bilevel optimization methods in the experiments?

4. The experimental results are not sufficiently convincing, as only a limited set of results is provided. Additionally, the experiments are conducted on a small-scale distributed system with only a few clients. I have concerns about the scalability of the proposed method, especially given its deterministic setting. Specifically: 1) Could the authors provide additional experimental results on a larger-scale distributed system? 2) The tasks addressed in the experiments are quite limited. I am curious if the proposed algorithm can be applied to more complex tasks, such as bilevel NAS (e.g., DARTS).

**Questions:**

Please refer to the Weaknesses. Additionally, the readability of this work is quite poor, with an excessive number of notations. Could you improve the manuscript’s readability and overall presentation?

---

> ### Author Response · Authors · 2024-11-22
>
> > **Comment 1:** What is the need for this algorithm in the ML community? This question arises from two main points: 1) Although this work avoids the strong convexity assumption, it still relies on the convexity assumption, whereas objectives in ML are often non-convex. Could you provide additional theoretical analyses that do not depend on convexity assumptions? 2) This work emphasizes the deterministic setting of the algorithm. While the deterministic setting is indeed important in optimization, I am curious about the challenges in analyzing convergence rates under a stochastic setting in the proposed algorithm. In ML, stochastic gradient descent is far more commonly used than deterministic approaches.
>
> **Our Response:** Thanks for your comments. We will organize our responses corresponding to your two subquestions, respectively:
>
> * **1) Convexity of the LL subproblem:** Although bilevel optimization (BLO) problems arise in a large number of ML paradigms, BLO problems are highly challenging and their algorithmic designs as well as the associated theoretical performance understandings remain very limited. As a starting point, most researchers in the ML community focused on a subset of *tractable* BLO problems, where the LL subproblems are strongly convex (see [1] for an excellent recent survey on BLO). So far, even relaxing BLO problems to cases with merely convex LL subproblems turns out to the highly non-trivial from a theoretical perspective. Our work not only relaxes the LL subproblem strong convexity assumption in BLO but also extends the studies to decentralized settings that have important applications in distributed ML training, marking a significant theoretical advancement. No prior work has addressed this important setting in the literature. Thus, our work fills an important gap in the BLO literature. On the other hand, even though the reviewer is correct that many ML tasks have non-convex objectives, BLO with convex or strongly convex LL subproblems have found quite a few interesting applications in ML (e.g., adversarial robust training [2], model pruning with sparsity constraint [3], invariant risk minimization [4])
>
> * **2) The Deterministic Setting:** Just as in most subfields in optimization algorithms design for machine learning, we also follow the "determistic-->stochastic" development order in this new decentralized BLO area (see the recent BLO survey in [1] to see the development history of determistic BLO algorithms before their stochastic counterpart). Specifically, we start with the relatively more tractable determistic setting to obtain a full understanding on algorithmic designs for decentralized BLO without LLSC, without which the development for stochastic cases would be difficult if not impossible. Furthermore, deterministic methods serve as a theoretical foundation for stochastic extensions. Moreover, we note that the determistic setting is also important in its own right in ML scenarios with moderate dataset sizes (e.g., LLMs supervised finetuning with small and high-quality datasets), where full gradient evaluations are possible.
>
> [1] Yihua Zhang, Prashant Khanduri, Ioannis Tsaknakis, Yuguang Yao, Mingyi Hong, and Sijia Liu, "An Introduction to Bi-level Optimization: Foundations and Applications in Signal Processing and Machine Learning," arXiv:2308.00788.
>
> [2] Yihua Zhang, Guanhua Zhang, Prashant Khanduri, Mingyi Hong, Shiyu Chang, and Sijia Liu, “Revisiting and advancing fast adversarial training through the lens of bi-level optimization,” in International Conference on Machine Learning, 2022, pp. 26693–26712.
>
> [3] Yihua Zhang, Yuguang Yao, Parikshit Ram, Pu Zhao, Tianlong Chen, Mingyi Hong, Yanzhi Wang, and Sijia Liu, “Advancing model pruning via bi-level optimization,” in Advances in Neural Information Processing
> Systems, 2022.
>
> [4] Martin Arjovsky, Leon Bottou, Ishaan Gulrajani, and David Lopez-Paz, “Invariant risk minimization,” arXiv preprint arXiv:1907.02893, 2019.

---

> ### Author Response · Authors · 2024-11-22
>
> > **Comment 2:** The relaxed stationarity (KKT-stationarity) is employed in this work without adequate justification. It is essential to provide a clear rationale for this relaxation, as it may result in significantly different solutions compared to the original problem. The stationarity measures are somewhat different from the traditional bilevel optimization works.
>
> **Our Response:** Thanks for your comments. We would like to further clarify and explain the rationale behind the use of KKT-stationarity. As stated in the paper, our work addresses decentralized bilevel optimization (DBO) without the LLSC assumption. In this setting, the hypergradient is **not** well-defined due to the lack of invertibility of the Hessian of the LL objective. To see this, note that the computation of the hyper-gradient (if exists) requires Hessian inverse computation (cf. Eq. (5) in Ref. [1] for the expression of the hyper-gradient). However, without LLSC, there is *no* guarantee that the Hessian matrix is full-rank and it is only positive semidefnite when the LL objective function is just convex (i.e., the Hessian matrix could be singular).
>
> Precisely because of the fundamental challenge of not having a well-defined hyper-gradient, one can **no longer** use the norm of the hyper-gradient in conventional bilevel optimization works as the stationarity measure. As a result, the stationary measure in bilevel optimization problems without LLSC has to be redefined and hence **cannot be the same as conventional bilevel optimization works**. To overcome this challenge, we reformulate the problem as a logically equivalent **single-level constrained optimization problem** in Eq. (3). Thanks to the logical equivalence, the optimal solution to this new single-level constrained optimization problem (characterized by the new problem's KKT system) implies the solution to the original bilevel optimization without LLSC. Further, the stationarity condition in the KKT system (one of the four KKT conditions) provides a natural **stationarity metric** for our subsequent optimization algorithm design.

---

> ### Author Response · Authors · 2024-11-22
>
> > **Comment 3:** In Corollary 2, in addition to discussing the number of communication rounds required for the proposed algorithm to converge, it is also essential to analyze the number of bits needed during this process. Can you provide some comparisons about the communication complexity between the proposed method with the existing distributed bilevel optimization methods in the experiments?
>
> **Our Response:** Thanks for your comments. Here, we first discuss per-round communication complexity, which will further shed light on overall communication complexity "in terms of the number of bits".
> 1. **Per-Round Communication Volume:** In DUET, the per-round communication cost (or volume) of each agent includes transmitting the UL variables ($\mathbf{x}_i$) and the tracked gradients ($\mathbf{h}_i$). Hence, the per-round communication cost in terms of "bits" at each node is $2d$, where $d$ is the model size. We note that this communication structure is *identical* to several existing decentralized gradient-tracking-based bilevel optimization methods, such as INTERACT [1] and DIAMOND [2] (i.e., having the same need to exchange these variables). As a result, the amount of information exchanged per round in DUET is the same as these existing methods in decentralized bilevel optimization with LLSC.
>
> 2. **Overall Communication Volume of DUET:** Multiplying the $2d$ per-round communication cost above with the communication complexity results in Corollaries 2 and 4 yields the overall communication costs of our DUET method in terms of "bits."
>
>
> [1] Zhuqing Liu, Xin Zhang, Prashant Khanduri, Songtao Lu, and Jia Liu. Interact: Achieving low sample and communication complexities in decentralized bilevel learning over networks. In Proceedings of the Twenty-Third International Symposium on Theory, Algorithmic Foundations, and Protocol Design for Mobile Networks and Mobile Computing, pp. 61–70, 2022.
>
> [2] P. Qiu, Y. Li, Z. Liu, P. Khanduri, J. Liu, N.B. Shroff, E.S. Bentley, and K. Turck. Diamond: Taming sample and communication complexities in decentralized bilevel optimization. In IEEE INFOCOM 2023 - IEEE Conference on Computer Communications, pp. 1–10, 2023.

---

> ### Author Response · Authors · 2024-11-22
>
> > **Comment 4:** The experimental results are not sufficiently convincing, as only a limited set of results is provided. Additionally, the experiments are conducted on a small-scale distributed system with only a few clients. I have concerns about the scalability of the proposed method, especially given its deterministic setting. Specifically: 1) Could the authors provide additional experimental results on a larger-scale distributed system? 2) The tasks addressed in the experiments are quite limited. I am curious if the proposed algorithm can be applied to more complex tasks, such as bilevel NAS (e.g., DARTS).
>
> **Our Response:** Thanks for your comments. We are currently working on additional experiment results. We will provide the additional experiment results in both the revised manuscript as well as the response in here as soon as we finish the additional experiments.

---

> ### Author Response · Authors · 2024-11-22
>
> > **Comment 5:** Please refer to the Weaknesses. Additionally, the readability of this work is quite poor, with an excessive number of notations. Could you improve the manuscript’s readability and overall presentation?
>
> **Our Response:** Thanks for your feedback regarding the readability and presentation of the manuscript. We will attempt to improve. We note that, due to the problem nature of in decentralized bilevel optimization without LLSC, the problem statement and algorithmic design inherently require complex formulations and careful theoretical treatment. Hence, having a large number of mathematical notations is somewhat inevitable. We note that the number of notations used in our paper is not to excessive in the sense that the number of notations is quite comparable with other works on decentralization bilevel optimization in this field. To enhance the readability and clarity as suggested by the reviewer, we have added a notation table in the appendix of the revised manuscript. This table will serve as a quick reference guide, summarizing all key notations and their definitions to make the content more accessible.

---

> ### Comment · Reviewer_KZ3e · 2024-11-23
> **Thanks for your rebuttals**
>
> Thank you for the responses. Unfortunately, my concerns remain unaddressed. The responses are not convincing enough, I still believe the proposed method offers limited contributions to the machine learning community due to its reliance on the convex assumption and deterministic setting. This work might be more suited for mathematical conferences or journals. Furthermore, decentralized bilevel optimization has already been studied for over two years [1], and replacing the strongly convex assumption with the convex assumption does not pique my interest, as both are strictly assumptions in the ML community, significantly limiting the practical applicability of the proposed method. The paper largely builds upon the ideas and analysis of [2], extending them to the context of decentralized bilevel optimization (DBO). In addition, as claimed by the author, the proposed method can be applied to adversarial robust training and model pruning with a sparsity constraint, but I cannot find any experimental results regarding these two applications, I only find limited experimental results which are not sufficiently convincing. Furthermore, after reading the revised paper, my concerns about the readability of this paper have not been adequately addressed. After reading all the reviews and rebuttals, combined with the aforementioned reasons as well as the current state of this work, I regret to conclude that I am unable to recommend acceptance to ICLR.
>
> [1] Chen, Xuxing, Minhui Huang, and Shiqian Ma. "Decentralized Bilevel Optimization." arXiv preprint arXiv:2206.05670 (2022).
>
> [2] Liu, Risheng, et al. "Averaged method of multipliers for bi-level optimization without lower-level strong convexity." International Conference on Machine Learning. PMLR, 2023.

---

> > ### Author Response · Authors · 2024-11-25
> >
> > >**Your Comment:** I still believe the proposed method offers limited contributions to the machine learning community due to its reliance on the convex assumption and deterministic setting. This work might be more suited for mathematical conferences or journals.
> >
> > **Our Response:** Thank you for your feedback. While we appreciate your review, we respectfully disagree with several of your remarks. Accordingly, the following are our responses to your remarks, which are organized in *three* key parts:
> >
> > **1) Our paper is well-suited for ML conferences including ICLR, as bilevel optimization (BLO) is a foundational problem of interest to a large audience in the machine learning community.**
> >
> > Specifically, BLO is a fundamental problem underlying the foundations of many machine learning paradigms, and has become a focal research area in machine learning, as evidenced by a large number of papers published in ICLR, ICML, and NeurIPS, i.e., the "Big Three" ML venues. For example, see below for a partial list of significant bilevel optimization papers published in ICLR, ICML, and NeurIPS, as well as their problem settings (**Note:** LLSC = "Lower-Level Strongly Convex"; LLC="Lower-Level Convex"):
> >
> > **ICLR:**
> >
> > [1] Wei Yao, Chengming Yu, Shangzhi Zeng, and Jin Zhang, “Constrained Bi-Level Optimization: Proximal Lagrangian Value Function Approach and Hessian-Free Algorithm,” Proc. ICLR 2024. (**LLSC**)
> >
> > [2]	Jie Hao, Xiaochuan Gong, and Mingrui Liu, “Bilevel Optimization under Unbounded Smoothness: A New Algorithm and Convergence Analysis,” Proc. ICLR 2024. (**LLSC**)
> >
> > [3]	Shi Fu, Fengxiang He, Xinmei Tian, and Dacheng Tao, “Convergence of Bayesian Bilevel Optimization,” Proc. ICLR 2024 (**LLC**)
> >
> > **ICML:**
> >
> > [4]	“Blockwise Stochastic Variance-Reduced Methods with Parallel Speedup for Multi-Block Bilevel Optimization”
> > Quanqi Hu, Zi-Hao Qiu, Zhishuai Guo, Lijun Zhang, Tianbao Yang. Proc. ICML 2023. (**LLSC**)
> >
> > [5]	Jaeho Kwon, Donghwan Lee, Stephen Wright, et al., “A Fully First-Order Method for Stochastic Bilevel Optimization,” Proc. ICML 2023. (**LLSC**)
> >
> > [6]	Prashant Khanduri, Ioannis Tsaknakis, Yihua Zhang, Jia Liu, Sijia Liu, Jiawei Zhang, and Mingyi Hong, “Linearly Constrained Bilevel Optimization: A Smoothed Implicit Gradient Approach,” Proc. ICML 2023 (**LLSC**)
> >
> > [7]	Kaiyi Ji, Junjie Yang, and Yingbin Liang, "Bilevel Optimization: Convergence Analysis and Enhanced Design," Proc. ICML 2021 (**LLSC**)
> >
> > [8]	Risheng Liu, Yaohua Liu, Wei Yao, Shangzhi Zeng, and Jin Zhang, “Averaged Method of Multipliers for Bi-Level Optimization without Lower-Level Strong Convexity,” Proc. ICML 2023. (**LLC**)
> >
> > [9] Zhuqing Liu, Xin Zhang, Prashant Khanduri, Songtao Lu, and Jia Liu, “Prometheus: Taming Sample and Communication Complexities in Constrained Decentralized Stochastic Bilevel Learning,” Proc. ICML 2023. (**LLSC**)
> >
> > [10] Xuxing Chen, Minhui Huang, Shiqian Ma, Krishnakumar Balasubramanian, “Decentralized Stochastic Bilevel Optimization with Improved Per-Iteration Complexity,” Proc. ICML 2023 (**LLSC**)
> >
> > **NeurIPS:**
> >
> > [11] Junjie Yang, Kaiyi Ji, and Yingbin Liang, “Provably Faster Algorithms for Bilevel Optimization,” Proc. NeurIPS 2021. (**LLSC**）
> >
> > **2) The Lower-level Convexity (LLC) assumption is NOT a limitation of our work. Rather, it is a fundamentally challeging setting in bilevel optimization (BLO). So far, how to develop efficeint solution apporach for the BLO-LLC setting remains NOT well-understood to the entire ML research community. Any breakthrough in the BLO-LLC problem that advances the state of the art should be appreciated.**
> >
> > In the list of important papers in the "Big Three" ML venues above, it can be clearly seen that **most of** these papers ([1,2,4,5,6,7,9,10,11]) relies on the LLSC assumption to simplify theoretical analysis and ensure convergence. In contrast, only a handful of existing works have attempted to address the LLC setting ([3,8]) and their convergence rates are far from satisctory. Moreover, **none of them considered BLO-LLC in the decentralized BLO setting**, for which our work is the first attempt in the literature. To clearly highlight this point, in this revision, we have made the following changes:
> >
> > 1) **Centralized BLO-LLC Setting:** For studies on centralized BLO-LLC problems, we have carefully discussed the literature in the related work section. We have also added several related work pointed out by Reviewer D8W2  to further enhance the literatrue review.
> >
> > 2) **Decentralized BLO-LLC Setting:** As shown in Table 1 and discussed in the related work section, there are currently **no** existing works in the literature that studied decentralized BLO-LLC problems. To fill this gap, **our work is the first to address this challenging setting decentralized BLO-LLC setting, shedding new theoretical insights and offering low-complexity algorithmic designs in practice.**

---

> > > ### Author Response · Authors · 2024-11-25
> > >
> > > **3) The determistic setting is important and interesting in its own right, since it helps obtain a fundamental understanding of the optimization problem and pave the way for stochastic algorithms design.**
> > >
> > > Again, we would like to emphasize that the determistic setting is always an important first step in many  optimization algorithms, including decentralized BLO problems. To convince the reviewers, in what follows, we list several recent BLO papers that specifically study deterministic BLO, **all of which are published in "Big Three" ML venues**. Moreover, even papers focusing on stochastic BLO typically first propose and analyze the deterministic version before extending it to the stochastic setting.
> > >
> > > [12] Luca Franceschi, Paolo Frasconi, Saverio Salzo, Riccardo Grazzi, and Massimiliano Pontil, “Bilevel programming for hyperparameter optimization and meta-learning,” Proc. ICML 2018, pp. 1568–1577.
> > >
> > > [13] Kaiyi Ji, Junjie Yang, and Yingbin Liang, “Bilevel optimization: Convergence analysis and enhanced design.” Proc. ICML 2021. (Same as [7])
> > >
> > > [14] Risheng Liu, Xuan Liu, Xiaoming Yuan, Shangzhi Zeng, and Jin Zhang. "A value-function-based interior-point method for non-convex bi-level optimization." Proc. ICML 2021.

---

> > ### Author Response · Authors · 2024-11-25
> >
> > >**Your Comment:** Furthermore, Decentralized Bilevel Optimization has already been studied for over two years [1], and the topic does not pique my interest.
> >
> >
> > **Our Response:** Thanks for your comment. We respect that different persons may have varying interests. However, **we believe that an "Accept" or "Reject" decision in a high-profile conference like ICLR should NOT be made solely based on a reviewer's subjective and personal interest. Rather, such an important decsion should be made based on the intellectual merits and technical contributions of the paper.**
> >
> > Also, while decentralized bilevel optimization has been studied for over two years, many important problems in this area are still wide open. Particulary, existing works in this area all assumed LLSC. Our work is the first attemtp to consider the general LLC setting, which requires new proof techniques and metrics to ensure convergence and practical applicability.
> >
> > Furthermore, **just because a problem has been studied for two years does not necessarily mean that the problem is uninteresting**. In fact, many mathematical research areas, a prolem or a conjecture could be open for decads or even hundreds of years. For example, in the areas of deep learning theory, deep artificial neural networks have been studied for more than 10 years. Yet, there is still a lack of theoretical foundation to fully explain why deep learning is so powerful. Similarly, BLO problems have been studied for several decades. But how to develop efficient algorithms for the non-LLSC setting remains an active research topic.
> >
> >
> >
> >
> > >**Your Comment:** The paper largely builds upon the ideas and analysis of [2], extending them to the context of decentralized bilevel optimization (DBO).
> >
> >
> > **Our Response:** We respectfully disagree with the assertion that our work is a simple extension of [2]. In our paper, we have thoroughly discussed the relationship and key differences between our work and [2] in the related work section. To further clarify, we highlight the following distinctions again in here:
> >
> > **1) Upper-Level (UL) Assumptions:** Our work assumes that the UL objective can be non-convex, providing a more general framework. **In contrast, [2] requires the UL objective to be strongly convex, limiting its applicability**.
> >
> > **2) Lower-Level (LL) Convexity Handling:** We address LL convexity by introducing a diminishing $\mu_t$-Quadratic Regularization to reformulating it into an augmented LL objective. This approach preserves the original problem as $\mu_t \to 0$. On the other hand, [2] combines the UL and LL objectives into a new LL objective via averaging, which relies on UL strongly convexity to recover strong convexity at the lower level.
> >
> > **3) Bounding via Projection Updates:** We use the projection updates for $\mathbf{y}$ and $\mathbf{v}$ ensuring boundedness of $\mathbf{x}$ during optimization. In contrast, [2] directly assumes that $(\mathbf{x},\mathbf{y})$ is bounded, without additional mechanisms to enforce this.
> >
> > **4) Decentralized Setting:** As you mentioned, our approch could deal with decentralzied setting. In contrast, [2] cannot. Extending bilevel optimization to decentralized settings is highly non-trivial, as the decentralized nature introduces additional complexities, including consensus errors, heterogeneous data, and communication constraints. Addressing these challenges required the development of a **novel convergence metric**, specifically designed to account for multiple error components: stationary error, LL error, dual multiplier error, and **consensus error**. This metric is critical for analyzing decentralized bilevel optimization without LLSC and ensuring convergence, making our work fundamentally distinct from [2].

---

> > ### Author Response · Authors · 2024-11-25
> >
> > >**Your Comment:** In addition, as claimed by the author, the proposed method can be applied to adversarial robust training and model pruning with a sparsity constraint, but I cannot find any experimental results regarding these two applications, I only find limited experimental results which are not sufficiently convincing.
> >
> > **Our Response:** Thanks for your feedback. While we understand the reviewer's desire for additional applications such as adversarial training and model pruning, we would like to clarify that adversarial robust training and model pruning with a sparsity constraint are classic applications of centralized bilevel optimization without LLSC (as demonstrated in [2, 3]). However, adversarial training and model pruning are not applicable in the context of **decentralized bilevel optimization (DBO)**, which is the primary focus of our work. This is the reason that we did not conduct experiments using these two applications. We provide two applications only in response to your question about the need for the non-LLSC setting.
> >
> > On the other hand, we chose to evaluate our proposed method using **meta-learning tasks** because meta-learning naturally involves multiple tasks distributed across agents, making it a good fit and meaningful application for decentralized multi-agent bilevel optimization without LLSC.
> >
> > Regarding your comment on limited experimental results, in this rebuttal period, we have conducted two more sets of experiments to demonstrate the efficacy of our proposed DUET method:
> >
> > 1) **Expanded Meta-Learning with 50 Agents:** To evaluate the scalability of our method, we have added experiments on a meta-learning task with 50 agents. These results further validate the robustness of our approach in larger-scale distributed settings.
> > 2) **Hyperparameter Optimization on FashionMNIST:** We have added experiments on hyperparameter optimization using the FashionMNIST dataset to illustrate the applicability of our method to a wider class of bilevel optimization problems.

---

> > > ### Comment · Reviewer_KZ3e · 2024-12-02
> > > **Thank you for your responses**
> > >
> > > Thank you for your responses. I truly appreciate it. However, after reviewing the second-round responses, I regret to say that my concerns remain unresolved. For instance:
> > >
> > > 1. I find it difficult to appreciate the limited contributions of this work. Specifically, it only relaxes a strict condition—strongly convex assumption—to a relatively strict one, the convex assumption, within a niche field. In fact, I hesitate to characterize it as a relaxation at all, given that this work is restricted to the deterministic setting (notably, the stochastic setting in decentralized bilevel optimization has already been studied in [1]). Consequently, while one condition is relaxed, another is tightened. Furthermore, this work lacks a comparison of sample complexity with [1], and I could not locate any convergence rate comparisons in the experimental results. Additionally, I still have concerns regarding the relaxed stationarity condition.
> > >
> > > 2. I also have some concerns about the potential application scenarios of this work. It appears that it may not be suitable for most machine learning settings due to its deterministic setting and convex assumptions. The experimental section lacks sufficient experiments to support the applicability of the proposed algorithm. Moreover, the authors mentioned in the first round of rebuttals that this work could be applied to adversarial robust training and model pruning. However, I did not see any experimental results to substantiate the applicability of the proposed algorithm in these areas since I was unable to verify all the demonstration steps outlined in the manuscript given the limited time. As such, I was hoping to see experimental evidence to confirm the correctness and practical applicability of the algorithm.
> > >
> > > Thus, I find it difficult to fully appreciate this work for several reasons, including its limited contributions and immature results. Therefore, I apologize to the authors for being unable to adjust my score.
> > >
> > > [1] Yang, Shuoguang, Xuezhou Zhang, and Mengdi Wang. "Decentralized gossip-based stochastic bilevel optimization over communication networks." Advances in neural information processing systems 35 (2022): 238-252.

---

> ### Author Response · Authors · 2024-12-03
>
> **Our Response for 1:** Thanks for your comment. We respectfully disagree with the assessment of limited contributions. Bilevel optimization is a key area in ML, with important applications in meta-learning and hyperparameter optimization. Its growing importance is evidenced in the long list of publications in the "Big Three" ML conferences as we have previously noted.
>
> Moreover, it has been widely recognized in the research community that the **watershed between easy and hard bilevel optimization problems is whether the lower-level problem is strongly convexity (LLSC) or just convex, regardless of whether the problem is deterministic or stochastic**. Without LLSC, the ill-defined implict gradient problem renders traditional bilevel optimization methods under LLSC assumption inapplicable. More specifically, due to the ill-defined hyper-gradient, **all existing bilevel optimization methods (e.g.[5-7]) that rely on hyper-gradient updates** breakdown and their convergence performance analysis do **NOT** even have a proper convergence metric in the first place. Excerbating the problem is the lack of a guaranteed unique lower-level solution in general, which fundamentall changes the problem formulation and necessites the optimization of LL variables in the UL problem (cf. our problem formulations and other related DBO work with LLSC [1-4]).
>
> Addressing decentralized bilevel problems without LLSC requires fundamentally new techniques for measuring stationarity and analyzing convergence. Our work directly tackles this challenge by introducing a **novel stationarity metric** and a **new approach** that not only handles LL convexity but also effectively addresses consensus errors, filling a clear gap in the literature and advancing the field meaningfully.
>
> While the stochastic setting has been explored in works such as [1] mentioned by the reviewer and [2-4], they **all rely on the LLSC assumption** to to simplify their analysis using traditional stationarity measures and approaches. Specifically, as highlighted in Assumption 3.4 of [1], their work is focused on decentralized bilevel optimization under LLSC. Due to this fundamental difference in setting, we do not compare our sample complexity with [1]. **Since Ref. [1] and other related DBO related work [2-4] cannot be used in non-LLSC setting, comparing their theoretical sample complexities and empirical convergence performances is infeasible.**
>
> [1] Yang, Shuoguang, Xuezhou Zhang, and Mengdi Wang. "Decentralized gossip-based stochastic bilevel optimization over communication networks." Advances in neural information processing systems 35 (2022): 238-252.
>
> [2] Zhuqing Liu, Xin Zhang, Prashant Khanduri, Songtao Lu, and Jia Liu. Interact: Achieving low sample and communication complexities in decentralized bilevel learning over networks. In Proceedings of the Twenty-Third International Symposium on Theory, Algorithmic Foundations, and Protocol Design for Mobile Networks and Mobile Computing, pp. 61–70, 2022.
>
> [3] Peiwen Qiu, Yining Li, Zhuqing Liu, Prashant Khanduri, Jia Liu, Ness B Shroff, Elizabeth Serena Bentley, and Kurt Turck. Diamond: Taming sample and communication complexities in decentralized bilevel optimization. In IEEE INFOCOM 2023-IEEE Conference on Computer Communications, pp. 1–10. IEEE, 2023.
>
> [4] Zhuqing Liu, Xin Zhang, Prashant Khanduri, Songtao Lu, and Jia Liu. Prometheus: Taming sample and communication complexities in constrained decentralized stochastic bilevel learning. In Proceedings of the 40th International Conference on Machine Learning, pp. 22420–22453, 2023b.
>
> [5] Junyi Li, Bin Gu, and Heng Huang. A fully single loop algorithm for bilevel optimization without hessian inverse. In Proceedings of the AAAI Conference on Artificial Intelligence, volume 36, pages 7426–7434, 2022.
>
> [6] Kaiyi Ji and Yingbin Liang. Lower bounds and accelerated algorithms for bilevel optimization. arXiv preprint arXiv:2102.03926, 2021.
>
> [7] Kaiyi Ji, Junjie Yang, and Yingbin Liang. Bilevel optimization: Convergence analysis and enhanced design. In ICML, 2021.

---

> > ### Author Response · Authors · 2024-12-03
> >
> > **Our Response for 2:** As stated in our second-round response, adversarial robust training and model pruning are applications of **centralized** bilevel optimization, but **not** of decentralized bilevel optimization (DBO) -- the focus of our work. Note that we mentioned these two applications **only in our response to the reviewer's first-round question on bilevel optimization's applications in general**. We didn't list them as application examples in our paper. In this work, we selected experiments that naturally align with decentralized multi-agent settings to demonstrate the applicability of our proposed algorithm.
> >
> > To validate its practicality, we conducted experiments on meta-learning tasks, which inherently involve distributed agents and are well-suited for DBO. **Next, regarding the reviewer's question on "applicability of the proposed method," we provide a proof here to sustantiate the applicability of the proposed method (i.e., the decentralized meta-learning problem does not have lower-level strong convexity, and only has lower-level convexity)**. To see this, we will show that the lower-level (LL) objectives $g_i$ in this experiment based on the cross-entropy loss are convex but not strongly convex with respect to the task-specific parameters $\mathbf{y} _ i$. Toward this end, the LL objective is
> > $$
> > g_i(\mathbf{x} _ i, \mathbf{y} _ i) = \frac{1}{|D_{\text{train}, i}|} \sum_{(\mathbf{u} _ {ij}, \mathbf{v} _ {ij}) \in D_{\text{train}, i}} \ell(\mathbf{y} _ i; \mathbf{u} _ {ij}, \mathbf{v} _ {ij}),
> > $$ where $D_{\text{train}, i}$ is the training dataset for agent $i$, consisting of feature-label pairs $(\mathbf{u} _ {ij}, \mathbf{v} _ {ij})$, $\ell(\mathbf{y} _ i; \mathbf{u} _ {ij}, \mathbf{v} _ {ij})$ is the cross-entropy loss given by $\ell(\mathbf{y} _ i; \mathbf{u} _ {ij}, \mathbf{v} _ {ij}) = -\sum_{c=1}^C \mathbf{v} _ {ij}[c] \ln \sigma(\mathbf{h} _ {ij}^\top \mathbf{y} _ i)[c].$ Here, $\mathbf{h} _ {ij}$ represents the transformed feature vector produced by the hidden layers, which is then passed to the final linear layer with task-specific parameters $\mathbf{y} _ i$. The term $\sigma(\mathbf{h} _ {ij}^\top \mathbf{y} _ i)$ is the softmax function applied to the logits $\mathbf{h} _ {ij}^\top \mathbf{y} _ i$, defined as: $\sigma(\mathbf{h} _ {ij}^\top \mathbf{y} _ i)[c] = \frac{\exp((\mathbf{h} _ {ij}^\top \mathbf{y} _ i)[c])}{\sum_{c'=1}^C \exp((\mathbf{h} _ {ij}^\top \mathbf{y} _ i)[c'])}$, where $C$ is the number of classes.
> >
> > Next, note that the cross-entropy loss $\ell(\mathbf{y} _ i; \mathbf{u} _ {ij}, \mathbf{v} _ {ij})$ is convex with respect to $\mathbf{y} _ i$. This can be readily verified by examining the Hessian of $\ell$ with respect to $\mathbf{y} _ i$, which can be computed as:
> > $$
> > \frac{\partial^2 \ell}{\partial \mathbf{y} _ i^2} = \mathbf{h} _ {ij} \mathbf{h} _ {ij}^\top \left( \text{diag}(\sigma) - \sigma \sigma^\top \right),
> > $$
> > where $\text{diag}(\sigma)$ is a diagonal matrix with $\sigma_c = \sigma(\mathbf{h} _ {ij}^\top \mathbf{y} _ i)[c]$ on the diagonal and $\sigma \sigma^\top$ is the outer product of the softmax probabilities. The term $\text{diag}(\sigma) - \sigma \sigma^\top$ is the covariance matrix of the softmax probabilities, which is positive semi-definite since it is diagonally dominant. Since $\mathbf{h} _ {ij} \mathbf{h} _ {ij}^\top$ is a positive semi-definite matrix, the entire Hessian $\frac{\partial^2 \ell}{\partial \mathbf{y} _ i^2}$ is positive semi-definite. Thus, the cross-entropy loss $\ell(\mathbf{y} _ i; \mathbf{u} _ {ij}, \mathbf{v} _ {ij})$ is convex but **not** strongly convex with respect to $\mathbf{y} _ i$. Lastly, since the summation operation in $g_i(\mathbf{x} _ i,\mathbf{y} _ i)$ preserves convexity, it follows that the LL objective $g _ i(\mathbf{x} _ i,\mathbf{y} _ i)$ is also convex but **not** strongly convex with respect to $\mathbf{y} _ i$.
> >
> > Additionally, we included new experiments on decentralized hyperparameter optimization using the FashionMNIST dataset. Together, these results provide strong evidence supporting the correctness and applicability of our algorithm in realistic DBO scenarios.

---

### Official Review · Reviewer_Psor · 2024-11-04

**Soundness:** 3
**Presentation:** 3
**Contribution:** 3
**Rating:** 6
**Confidence:** 2

**Summary:**

This paper addresses decentralized bilevel optimization (DBO) in multi-agent systems without relying on lower-level strong convexity (LLSC), a common but limiting assumption. The authors propose DUET, a novel single-loop algorithm that introduces diminishing quadratic regularization to the lower-level objective, thus bypassing the need for LLSC. DUET achieves convergence to an approximate KKT-stationary point with a defined iteration complexity and incorporates gradient tracking to handle data heterogeneity across agents. This work is the first to address DBO without LLSC in decentralized settings with heterogeneous data, with both theoretical analysis and numerical experiments supporting DUET’s effectiveness.

**Strengths:**

1 The paper is well-written, providing clear explanations in the methodology section where each step addresses specific challenges, along with comparisons to existing problems and methods.

2 The authors compare the convergence rate and other results, clearly demonstrating improvements over current methods.

3 They propose using approximate KKT stationarity as the convergence measure for DBO solution quality, offering a detailed analysis of the convergence rate based on this new measure, which will aid in further research on this topic.

**Weaknesses:**

1 The real-world data experiments are limited to decentralized meta-learning problems on the MNIST dataset, which I believe may not be sufficiently comprehensive.

**Questions:**

Could you provide experimental results on other problems or datasets? This would help further demonstrate the effectiveness of the proposed method.

---

> ### Author Response · Authors · 2024-11-22
>
> Thanks for your comments. We are currently working on additional experiment results. We will provide the additional experiment results in both the revised manuscript as well as the response in here as soon as we finish the additional experiments.

---

> > ### Author Response · Authors · 2024-11-25
> >
> > Regarding your comment on limited experimental results, in this rebuttal period, we have conducted two more sets of experiments to demonstrate the efficacy of our proposed DUET method:
> >
> > 1) **Expanded Meta-Learning with 50 Agents:** To evaluate the scalability of our method, we have added experiments on a meta-learning task with 50 agents. These results further validate the robustness of our approach in larger-scale distributed settings.
> > 2) **Hyperparameter Optimization on FashionMNIST:** We have added experiments on hyperparameter optimization using the FashionMNIST dataset to illustrate the applicability of our method to a wider class of bilevel optimization problems.

---

> > > ### Comment · Reviewer_Psor · 2024-11-27
> > >
> > > Thank you for the additional experiments. I will maintain my score.

---

### Meta-Review · Area_Chair_JHng · 2024-12-14

**Metareview:**

This paper studies decentralized bilevel optimization (DBO) without lower-level strong convexity. In particular, the authors proposed a new single-loop algorithm called DUET to handle this class of problems. Iteration complexity for obtaining an approximation KKT-stationary point is shown. Existing works on DBO requires strong convexity for lower-level problem, and this work weakened it by only requiring convexity for lower-level problem. So the contribution is clear. The authors are advised to incorporate various suggestion from the reviewers to improve the presentation and the numerical experiments in the final version.

**Additional Comments On Reviewer Discussion:**

Added numerical experiments and discussed novelty.

---

### Decision · Program_Chairs · 2025-01-22

Accept (Poster)